# Circadian control of brain glymphatic and lymphatic fluid flow

Lauren M. Hablitz 1✉, Virginia Plá 1, Michael Giannetto1, Hanna S. Vinitsky 1, Frederik Filip Stæger2, Tanner Metcalfe1, Rebecca Nguyen1, Abdellatif Benrais1 & Maiken Nedergaard1,2✉

The glymphatic system is a network of perivascular spaces that promotes movement of cerebrospinal fluid (CSF) into the brain and clearance of metabolic waste. This fluid transport system is supported by the water channel aquaporin-4 (AQP4) localized to vascular endfeet of astrocytes. The glymphatic system is more effective during sleep, but whether sleep timing promotes glymphatic function remains unknown. We here show glymphatic influx and clearance exhibit endogenous, circadian rhythms peaking during the mid-rest phase of mice. Drainage of CSF from the cisterna magna to the lymph nodes exhibits daily variation opposite to glymphatic influx, suggesting distribution of CSF throughout the animal depends on time-of-day. The perivascular polarization of AQP4 is highest during the rest phase and loss of AQP4 eliminates the day-night difference in both glymphatic influx and drainage to the lymph nodes. We conclude that CSF distribution is under circadian control and that AQP4 supports this rhythm.

[1] Center for Translational Neuromedicine, University of Rochester Medical Center, Rochester, NY 14642, USA. [2] Center for Basic and Translational Neuroscience, Faculty of Health and Medical Sciences, University of Copenhagen, 2200 Copenhagen, Denmark. ✉email: Lauren_Hablitz@urmc.rochester.edu; Nedergaard@urmc.rochester.edu

Sleep is an evolutionarily conserved biological function, clearing the brain of harmful metabolites such as amyloid beta that build up during wakefulness[1] and consolidating memory[2]. Acute sleep deprivation impairs cognitive function[3,4], and sleep disruption is often an early correlate of neurodegenerative disease[5]. Sleep quality is highly dependent on both sleep deficit and sleep timing controlled by circadian rhythms, 24 hour cycles in gene transcription, cell signaling, physiology, and behavior[6]. Understanding how circadian rhythms contribute to sleep quality is necessary to promote long-term brain health.

During sleep, there is increased cerebrospinal fluid (CSF) movement into the glymphatic system, promoting waste clearance[1]. The glymphatic system is a network of perivascular tunnels wrapped by astrocyte endfeet[7]. The localization of the water channel aquaporin-4 (AQP4) to astrocyte vascular endfeet promotes CSF movement through the brain[8]. This fluid movement is also dependent upon arterial pulsation[9,10], and correlated to decreased heart rate[11]. Although the cardiovascular system and sleep, two main influencers of the glymphatic system, are tightly regulated by circadian rhythms[12], it remains unknown whether glymphatic fluid movement is under circadian control.

CSF moves not only into the brain and spinal cord via the glymphatic system, but also drains out of the central nervous system to the lymphatic system of the body[13,14]. In fact, lymphatic drainage may prevail when the animals are awake and glymphatic influx is low[15]. The immune system is also regulated by circadian rhythms[16,17], yet whether lymphatic drainage of CSF to the lymph nodes is under clock control has to be determined.

Here, we test the hypothesis that glymphatic influx of CSF tracer from the cisterna magna (CM) into the brain, and clearance of solute from the brain varies across the day independent of arousal state. We also test whether AQP4 may have a role in rhythmic glymphatic function. We show that glymphatic influx and clearance of a small tracer from the parenchyma is increased during the day compared with night in mice independent of the type of anesthesia or light/dark cycle, corresponding to day/night changes in AQP4 localization. These rhythms in influx, clearance, and AQP4 persist after 10 days in constant light, indicating that the glymphatic system is under circadian control. AQP4 knockout (AQP4 KO) animals exhibit no day/night difference in glymphatic influx. In fully anesthetized animals, in vivo imaging of mandibular lymph nodes showed increased tracer outflow during the night compared with day, when entry of CSF to the brain is low. This day/night difference in lymph node filling persists in constant conditions, and is absent in animals without AQP4.

## Results

**The glymphatic system exhibits diurnal variation**. First, we tested the hypothesis that there was a day/night difference in influx of CSF tracer along the perivascular spaces around the middle cerebral artery (MCA) of the brain. Mice were anesthetized with ketamine/xylazine (KX) at mid-day or mid-night (ZT6 or ZT18, respectively; ZT0 is lights on). In order to measure glymphatic function, fluorescent tracer was injected into the CSF pool in the CM and allowed to circulate for 30 min prior to brain collection. During circulation, fluorescent tracer movement was imaged through the skull of anesthetized mice (Fig. 1a–c, Supplementary Videos 1 and 2). We found ~53% more CSF tracer influx to the brain during the day compared with night in vivo along the MCA (mean pixel intensity (MPI)) at 30 min during the day: $53.68 \pm 8.5$ MPI, $n = 6$ mice, night: $24.56 \pm 7.1$ MPI, $n = 6$ mice; $t$ test: $t(10) = 2.617$, $p = 0.0257$. Because there is also increased CSF tracer in areas outside of cortex, we re-analyzed the images using an ROI more tightly confined to the MCA and

confirmed similar results (Mann–Whitney test: $U = 4$, $p = 0.0260$; Supplementary Fig. 1a, b).

After establishing a day/night difference of influx in vivo, we generated a full time course of glymphatic influx under KX anesthesia (Fig. 1a, d). CM injections of CSF tracer were performed every 4 hours across a standard 24 h light cycle, for a total of six time points. Using ex vivo tissue processing, we prepared 100 μm thick coronal sections through the brains and quantified mean pixel fluorescence intensity across six slices collected at 600 μm intervals starting at 1.2 mm anterior to bregma, to calculate the mean total tracer influx per entire brain ($n = 10$–12 mice per time point; Supplementary Fig. 1b, Supplementary Fig. 2). Cosinor analysis of mean pixel intensity of brains across a 24 h cycle revealed a significant rhythm in glymphatic influx with a peak at ZT6.2, around mid-day (cosinor analysis: $F(2, 67) = 5.949$, $p = 0.004$; $R^2 = 0.151$). The difference in influx between the day and night was ~22% (average day influx: $7.91 \pm 0.4$ MPI, average night influx: $6.20 \pm 0.3$ MPI). Because these experiments were all performed under KX anesthesia, and some anesthetics have potential to shift circadian timing[18], we generated full time courses under two additional anesthetics, pentobarbital and 2,2,2-tribromoethanol (also known as avertin) (Fig. 1d; $n = 7$–11 mice per time point for each anesthetic). Anesthesia impacted overall tracer influx, as reported previously[11], where KX had significantly more influx (average influx under KX: $7.08 \pm 0.3$ MPI) when compared with pentobarbital ($4.42 \pm 0.1$ MPI) or tribromoethanol ($3.47 \pm 0.1$ MPI; Kruskal–Wallis $H(3) = 105.2$, $p < 0.0001$). More importantly, we show significant rhythms under both pentobarbital (cosinor analysis: $F(2,52) = 5.527$, $p = 0.0069$; $R^2 = 0.175$) and 2,2,2-tribromoethanol (cosinor analysis: $F(2,49) = 6.479$, $p = 0.0035$; $R^2 = 0.209$) with acrophases around mid-day (ZT7.8 and ZT6.6, respectively) and similar amplitudes (Fig. 1e, f).

In addition to using a cosinor analysis to determine rhythmicity and circadian parameters of glymphatic influx, we also analyzed the data using CircWave[19–21]. This re-analysis confirmed our previous findings, where the simplest fit of the data were a single-harmonic sinusoid curve that accounted for ~20% of the overall variance. We also compared the Center of Gravity (CoG), a measure of phase and mesor of the data sets, and found similar results to our 95% confidence intervals above, where anesthetic changed the mesor but each phase estimate was approximately mid-day (Supplementary Fig. 1c–e). These experiments demonstrate that regardless of anesthesia, glymphatic influx of CSF tracer into the brain exhibits diurnal variation with peak influx around mid-day.

Although glymphatic influx is significantly higher in the sleeping[1] or anesthetized[11] brain, we tested whether awake mice showed day/night variation of CSF influx. We found no significant difference of CSF tracer between the day (ZT4–8, $n = 7$ mice) and night (ZT16–20, $n = 9$ mice) in total influx, anterior to posterior distribution, or in distinct subregions (Supplementary Fig. 1f–h). This was unsurprising as the awake brain has limited glymphatic function[1].

Glymphatic influx does not reflect function of the whole glymphatic system, therefore we measured clearance of Evans blue (EB), a small molecule (960 Da) that freely diffuses through the brain and binds tightly to albumin in the blood, from the brain to the femoral vein in anesthetized animals during the day ($n = 9$ mice) and night ($n = 6$ mice). We chose this method because although we cannot measure where in the vasculature EB enters and binds to albumin, any fluorescence detected in the femoral vein had to be cleared from the brain parenchyma. Thus, this approach enabled us to test the hypothesis that net clearance of EB from the brain is under diurnal control. Mice were implanted with a microdialysis cannula into the striatum 24–36 h

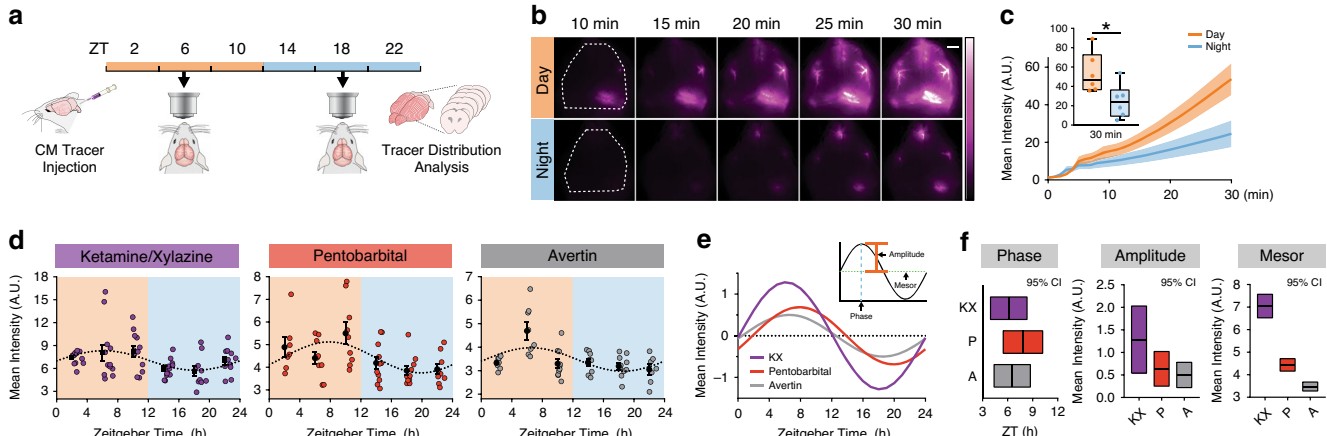

**Fig. 1 Glymphatic influx is higher during the day. a** Schematic of experiment. **b** Representative images for in vivo recordings of CSF tracer influx after injection in the CM either during the day (orange) or night (blue) under ketamine/xylazine (KX) anesthesia. Dotted white line indicates region of quantification in **c**. White scale bar: 2 mm. **c** Mean pixel intensity in arbitrary units (A.U.) over 30 min post CM injection. Thick lines indicate group means with SEM outlined in shaded regions. The inset is the magnitude of fluorescence at 30 min with boxplots, minima is minimum value, maxima is maximum value, center is median and quartiles shown by box and whiskers, individual animals represented as colored dots, two-sided $t$ test, $p = 0.0257$. Time 0 is tracer infusion start. $n = 6$ mice per group. Asterisk indicates $p < 0.05$. **d** Time course of mean pixel intensity from ex vivo coronal sections under KX (violet, $n$ = ZT22: 10 mice; ZT18: 11 mice; ZT2, ZT6, ZT10, and ZT14: 12 mice), pentobarbital (red, $n$ = ZT2: 7 mice; ZT6, ZT10, and ZT22: 9 mice; ZT18: 10 mice; ZT14: 11 mice), and Avertin (gray, $n$ = ZT22: 7 mice; ZT2 and ZT6: 8 mice; ZT14: 9 mice; ZT10 and ZT18: 10 mice). Each colored point is an animal, black points are mean ± SEM for each time bin. Dotted line indicates estimated cosinor curve. The light cycle is indicated by orange (day) and blue (night) coloring in the background. Zeitgeber Time (ZT) in hours, where ZT0 is lights on and ZT12 is lights off. **e** Estimated cosine curves aligned at the mesor for KX, pentobarbital (P), and avertin (A), with an inset depicting measurements from a cosine curve. **f** 95% confidence intervals for estimates of phase, amplitude, and mesor for each anesthetic. Midline is the model generated estimate, bars indicate 95% range around the estimate. Source data are provided as a Source Data file.

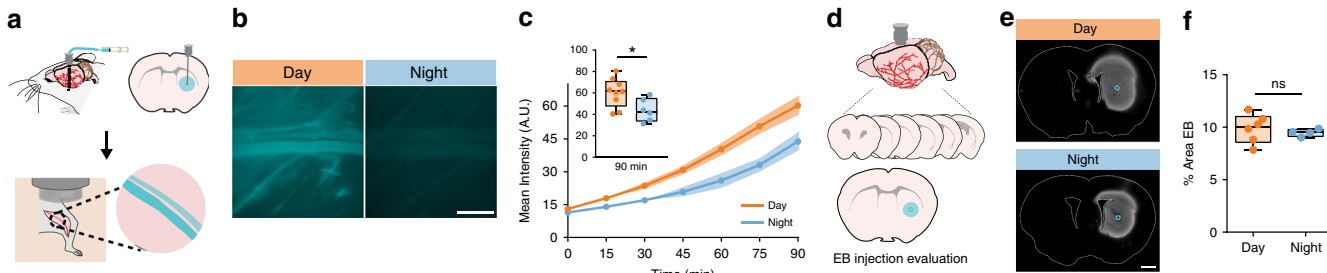

**Fig. 2 Interstitial fluid clearance is higher during the day. a** Schematic of experiment. **b** Representative Cy5 images of the femoral vein from animals 90 min after evans blue (EB) injection during the day (orange) or night (blue). White scale bar: 1 mm. **c** Mean pixel intensity in arbitrary units (A.U.) over 90 min post EB injection. Thick lines indicate group means with SEM shaded, individual points indicate time image was taken. The inset is a boxplot of the magnitude of fluorescence at 90 min, two-sided $t$ test, $p = 0.0256$. Time 0 is tracer infusion start. Day: $n = 9$ mice, night: $n = 6$ mice. Asterisk indicates $p < 0.05$. **d** Schematic of processing brains to check injection localization and specificity. **e** Representative striatal EB injection sites. Blue circle indicates probe end. White scale bar: 1 mm. **f** Boxplot of percent area of EB quantified in a subset of brains during the day ($n = 6$ mice) and night ($n = 4$ mice). All boxplots: minima is minimum value, maxima is maximum value, center is median and quartiles shown by box and whiskers, with individual animals shown as colored dots, ns: not significant. Source data are provided as a Source Data file.

prior to experiments. At the time of experiment, the microdialysis probe was inserted in the cannula, the femoral vein exposed, and placed under the macroscope (Fig. 2a). Imaging began at the beginning of infusion of EB (4% in ACSF, 1 μL, 0.2 μL min$^{-1}$). EB was significantly ($t$ test: $t(13) = 2.605$, $p = 0.0218$) higher in the femoral vein as early as 15 min post infusion (day: $17.96 ± 1.1$ MPI; night: $14.00 ± 0.8$ MPI). By 90 min there was 37% more EB during the day than the night ($t$ test: $t(13) = 2.519$, $p = 0.0256$; Fig. 2b, c). There was no significant difference in percent area covered by EB in the brain after the experiment (Mann–Whitney test: $U = 9$, $p = 0.6095$; Fig. 2d–f), indicating that infusion volumes were similar in each group. We conclude that, similar to glymphatic influx, clearance of solutes from the brain is increased during the day and decreased during the night, supporting the notion that global glymphatic function is upregulated during the day.

Influx of CSF into the brain parenchyma is highly dependent upon localization of the water channel AQP4 to the endfeet of astrocytes that surround brain vasculature[7,8,22–24]. Subsequently, we asked if AQP4 localization to the vasculature exhibits diurnal variation. Using immunohistochemistry in brain slices taken at ZT6 ($n = 11$ mice, 99 vessels) or ZT18 ($n = 12$ mice, 108 vessels), we found increased polarization of AQP4 around vascular structures across the cortex during the day (Fig. 3a–c; $t$ test: $t(21) = 4.173$, $p = 0.0004$), with clear trends persisting in both the hypothalamus ($n = 4$ mice per group; $t$ test: $t(6) = 2.342$, $p = 0.0577$) and CA1 of the hippocampus ($n = 4$ mice per group; Welch's $t$ test: $t(3.109) = 2.139$, $p = 0.1188$) (Supplementary Fig. 3). We next asked whether this change of localization corresponded to a change in total amounts of AQP4. Using western blot to measure protein in whole-brain extract, we found

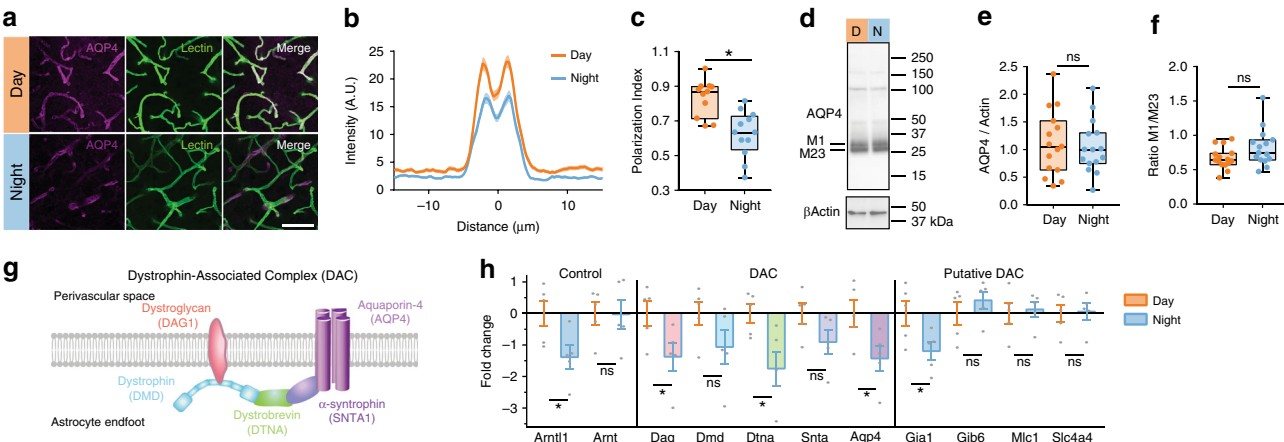

**Fig. 3 Diurnal variation in AQP4 protein and localization. a** Representative ×63 magnification confocal images during the day (ZT6) and night (ZT18). Purple is AQP4, green is vascular staining, white is overlap between both AQP4 and vascular staining. White scale bar: 50 μm. **b** Average intensity of AQP4 staining centered on vasculature in the cortex during the day (orange, for **b**, **c**: $n = 11$ mice, 99 vessels) and night (blue, for **b**, **c**: $n = 12$ mice, 108 vessels), with mean ± SEM indicated by the thick line (mean) with shading (SEM). **c** Average polarization index boxplot. Polarization index equals peak vascular end foot fluorescence minus 10 μm baseline. All values were normalized to the highest signal for ease of visualization. Two-sided $t$ test, $p = 0.0004$. **d** Representative western blot for AQP4 and βActin loading control during the day (orange) and night (blue). Molecular weights are indicated to the right. M1 and M23 splice variants of AQP4 monomers are indicated on the left. **e** Bloxplot of mean AQP4 densitometry normalized to the loading control. For **e**, **f**: $n = 15$ mice per group. **f** Bloxplot of the ratio of M1/M23 densitometry, both normalized to loading control. **g** Schematic of key components of the dystrophin-associated complex. **h** Bar plot of day (ZT6) and night (ZT18) mRNA expression of target genes assessed by RT-qPCR means with error bars indicating SEM. Gray dots are individual mice. $n = 5$ mice per group. Two-sided Welch's $t$ tests: Arntl1, $p = 0.037$; Dag, $p = 0.0483$; Dtna, $p = 0.0283$; Aqp4, $p = 0.038$; Gja1, $p = 0.045$. DTNA, DMD, DAG1, AQP4, GJA1, GJB6, MLC1, and SLC4A4 are all upregulated in astrocytes at minimum two times more than any other cell type[30]. All boxplots: minima is minimum value, maxima is maximum value, center is median and quartiles shown by box and whiskers, with individual animals shown as colored dots. Asterisk indicates $p < 0.05$; ns: not significant. Source data are provided as a Source Data file.

no difference in total AQP4 between ZT6 and ZT18 (Fig. 3d, e, day: $n = 15$ mice, night: $n = 15$ mice), or in higher molecular weight bands of AQP4 complexes (Supplementary Fig. 4). In mice, M1 or M23 splice variants can determine localization of AQP4, with higher levels of M23 at astrocytic endfeet[25,26]. Although we found no difference in intensity of the M1 or M23 bands between day and night (Supplementary Fig. 4), we found a trend in the ratio of M1 to M23 being lower during the day (day: $0.67 \pm 0.04$ M1/M23; night: $0.82 \pm 0.07$ M1/M23; Welch's $t$ test, $t$ (22.41) = 1.912, $p = 0.0688$; Fig. 3f), supporting the hypothesis that AQP4 localization to perivascular endfeet during the day promotes increased glymphatic influx.

AQP4 localization to the vascular endfeet of astrocytes is dependent upon the dystrophin-associated complex (DAC)[27–29], components of which exhibit highly enriched gene expression in astrocytes (Dtna, Dmd, Dag1, Aqp4) according to a cell-specific transcriptome data base[30]. We hypothesized that day/night changes in gene expression for components of this complex may support rhythmic localization of AQP4 to perivascular endfeet of astrocytes. Brains were harvested at ZT6 ($n = 5$ mice) and ZT18 ($n = 5$ mice), then reverse transcription PCR (RT-qPCR) was used to measure gene expression. First, we measured the clock gene Arntl1 and the non-clock gene homolog Arnt[31]. As expected, there was a significant day/night difference in Arntl1 (Welch's $t$ test, $t$ (7.912) = 2.504, $p = 0.037$; Fig. 3h), and no difference in Arnt. Next, we measured gene expression of key components of the DAC: Dag1, Dmd, Dtna, Snta1, and Aqp4 (Fig. 3g, h). We found significant increased expression of Dag, Dtna, and Aqp4 during the day compared to the night (Welch's $t$ tests, Dag: $t$ (7.874) = 2.335, $p = 0.0483$; Dtna: $t$ (6.272) = 2.837, $p = 0.0283$; Aqp4: $t$ (7.973) = 0.038). Both Dmd and Snta1 had similar trends of expression, but the day/night differences were not significant (Fig. 3g). In addition to AQP4, several genes encoding gap junctions (Gja1, Gjb6), cation channels (Mlc1), and bicarbonate transporters (Slc4a4) are upregulated in astrocytes[30],

and putatively associated with the DAC[27]. We found only Gja1 exhibited day/night differences in gene expression, with a reduction during the night (Welch's $t$ test, $t$ (7.201) = 2.43, $p = 0.0445$). This may indicate that temporal localization of connexin-43 to astrocytic endfeet is regulated by the DAC. Increased gene expression of key components of the DAC, including AQP4, match the increased localization of AQP4 and increased glymphatic influx of CSF into the brain during the day.

**Influx clearance and AQP4 rhythms persist in constant light.** Next, we tested the hypothesis that diurnal changes in glymphatic CSF tracer influx are not driven by cycles of light and dark (LD). We chose constant light (LL) over constant dark (DD) to avoid effects of light pulses that are inevitable during live-animal surgery. The surgery takes ~10 min, more than enough to phase shift mice in constant darkness[32,33]. Mice were housed in constant light (LL) for 10 days with continuous activity monitoring to ensure chi-squared periodogram estimates of free-running period were accurate[34]. Because long-term (>30 days) LL induces behavioral arrhythmicity and altered SCN organization[35,36], we wanted to ensure that general cage activity was unchanged in LL ($n = 6$ cages of 2 mice) compared with DD ($n = 10$ cages of two mice). Using chi-squared periodogram analysis we found animals in DD and LL had similar variability in periodicity, and every cage had significant Qp values from a chi-squared periodogram analysis (Supplementary Fig. 5a–c), though the amplitude of the Qp was reduced in the constant light group (DD: $1016.0 \pm 61$ Qp, LL: $757.7 \pm 85$ Qp; Welch's $t$ test: $t$ (10.13) = 2.477, $p = 0.0325$). Similar to previous reports, mice in LL had an average period of $25.35 \pm 0.1$ h. Using average activity profiles for 10 days in LL and DD, we found no evidence of splitting in LL (Supplementary Fig. 5d). Although there was a trend toward decreased activity, specifically in the active phase in LL ($t$ test: $t$ (14) = 2.089, $p = 0.0555$; Supplementary Fig. 5e, f), there was no significant difference in the average amount of activity between light cycles

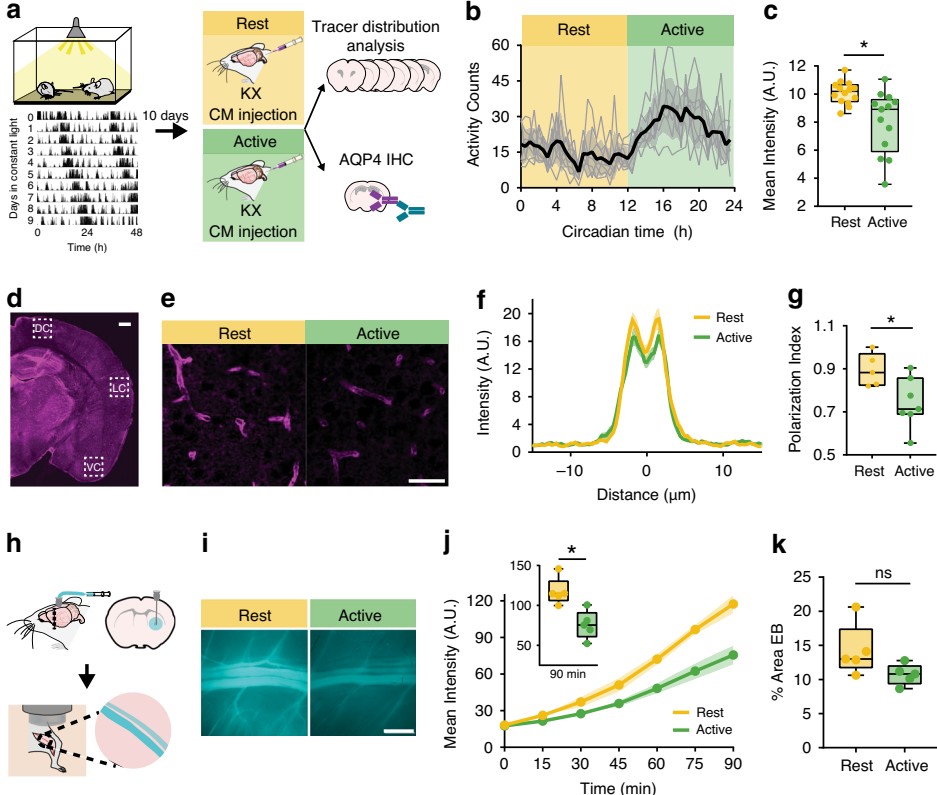

**Fig. 4 Differences in the glymphatic system persist in constant light. a** Experimental outline with representative double-plotted actogram of two animals in constant light (LL). Black tick marks: beam-breaks. **b** Average activity profile of animals housed in LL for 10 days. Black line and gray error bars: group mean ± SEM, thin gray lines: individual cage activity profiles. Yellow: rest phase, green: active phase. Circadian time (CT) 12 is activity onset. $n = 7$ cages, 14 mice. **c** Boxplot of mean intensity (arbitrary units, A.U.) from ex vivo tissue fluorescence. Mann–Whitney test: $p = 0.0033$, $n = 13$ mice rest, 14 mice active. **d** Representative slice of AQP4 staining. White boxes indicate areas of dorsal, lateral, and ventral cortex (DC, LC, VC, respectively) for AQP4 localization quantification. White scale bar: 500 μm. **e** Representative ×40 images stained for AQP4 (magenta). White scale bar: 50 μm. **f** Average intensity of AQP4 centered on vasculature in cortex during the rest (yellow, for **e**, **g**: $n = 5$ mice, 45 vessels) and active (green, for **f**, **g**: $n = 7$ mice, 63 vessels) phase, mean ± SEM is indicated by shading (SEM) around thick line (mean). **g** Boxplot of polarization index for staining in ventral, dorsal and lateral cortex. Polarization index equals peak fluorescence minus 10 μm baseline. All values normalized to the highest signal for better visualization. Two-sided $t$ test: $p = 0.0288$. **h** Schematic of clearance experiment. **i** Representative Cy5 images of the femoral vein 90 min after evans blue (EB) injection during rest (yellow) or active (green) phases. White scale bar: 1 mm. **j** Mean intensity over time. Thick lines: group means, outlines: SEM, individual points: image acquisition. Inset: boxplot of fluorescence at 90 min. Two-sided $t$ test: $p = 0.0051$. Time 0 is tracer infusion start. For **j**, **k**: $n = 5$ mice per group. **k** Boxplot of percent area EB during rest and active phases. All boxplots minima: minimum value, maxima: maximum value, center: median, and quartiles: box and whiskers, individual animals: colored dots. Asterisk indicates $p < 0.05$; ns: not significant. Source data are provided as a Source Data file.

($t$ test: $t(14) = 1.727$, $p = 0.1062$). Thus, 10 days in LL does not drastically alter circadian behavior, indicating the circadian system is intact.

On day 11, animals were anesthetized with KX, received a CM injection of CSF tracer, and ex vivo glymphatic influx analyses were performed during the mid-rest (circadian time 4–8, CT 4–8, where CT 12 is activity onset) or mid-active phase (CT 16–20) of the animal's behavior (Fig. 4a, b). Influx was increased during the rest phase compared with the active phase of animal behavior (Fig. 4c; rest: $10.10 \pm 0.2$ MPI, active: $8.00 \pm 0.6$ MPI, $n = 13$–14 mice per group; Mann–Whitney test: $U = 32$, $p = 0.0033$). In addition, AQP4 polarization to vascular endfeet was increased during rest phase of the animal (rest: $n = 5$ mice, 45 vessels; active: $n = 7$ mice, 63 vessels; $t$ test: $t(10) = 2.552$, $p = 0.0288$; Fig. 4d–g). Finally, we performed EB clearance assays, as described above, during the rest phase (CT 2–7, $n = 5$ mice) or active phase (CT 14–19, $n = 5$ mice). Similar to LD, clearance was significantly increased by 55% during the rest phase compared with the active phase at 90 min ($t$ test: $t(8) = 3.823$, $p = 0.0051$), with no significant differences in percent coverage of EB in the brain (Fig. 4h–k). These data support the hypothesis that the

rhythm in glymphatic function is under circadian control, as opposed to being driven by changes in the light cycle.

**Lymphatic drainage rhythms are opposite to glymphatic influx.** Glymphatic function is increased during sleep[1] and drainage of CSF to the lymph nodes is increased when awake[15]. Because we observed a strong underlying circadian rhythm in glymphatic influx, our next question was whether CSF outflow via the cervical lymphatic system showed circadian variation. Mice received CM injections of tracer under KX anesthesia, as discussed above, during the day ($n = 12$ mice) or night ($n = 15$ mice). The mandibular lymph nodes were then exposed and imaged for 50 min post CM. We found that filling of the submandibular lymph nodes was 46% lower during the day compared with the night (Fig. 5a–e; $t$ test: $t(25) = 2.25$, $p = 0.0335$). This effect persisted after 10 days in constant light (Fig. 5f–j; rest: $n = 8$ mice, active: $n = 6$ mice; $t$ test: $t(12) = 2.603$, $p = 0.0231$). Appearance of tracer in the lymph node occurs as early as the first frame of recording (on average 8 min post tracer infusion start; Fig. 5e, g), similar to first detectable tracer movement along the perivascular space of the MCA (5 min post injection; Fig. 1c), suggesting that the CSF

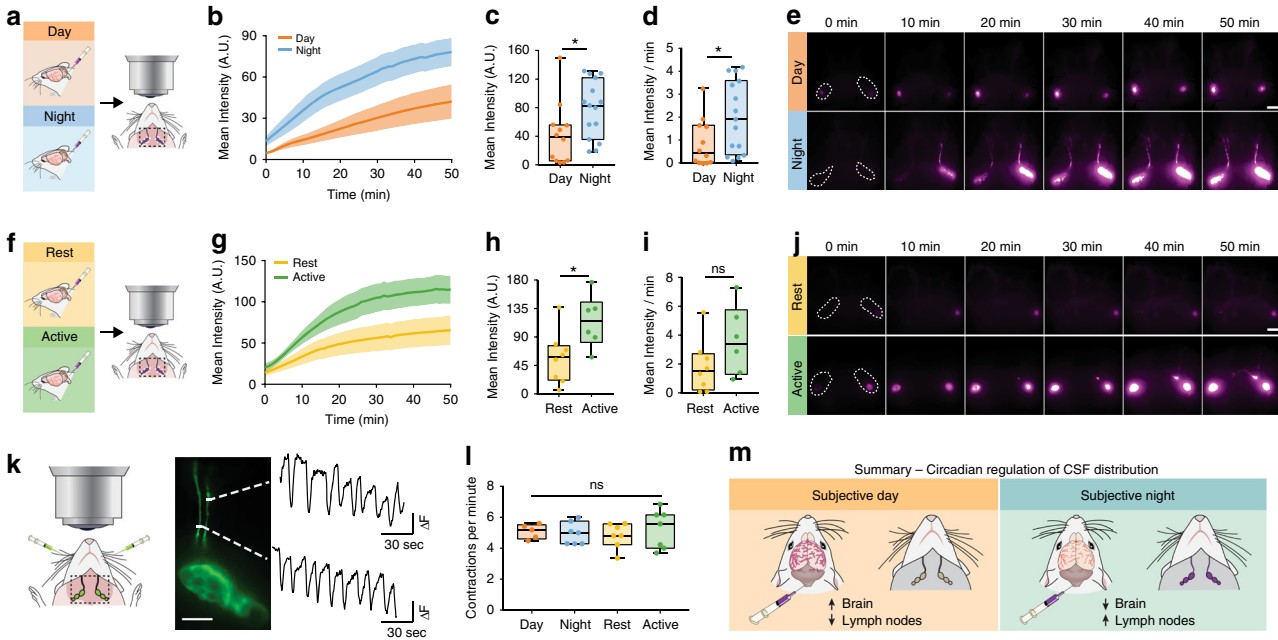

**Fig. 5 Lymph node drainage exhibits circadian variation. a** Experimental outline. Lymph nodes in purple, black dashed square: area imaged. White dashed lines: regions of interest (ROI) around lymph nodes. **b** Mean intensity (arbitrary units, A.U.) of lymph node fluorescence during day (orange, for **b**–**e**: $n = 12$ mice) and night (blue, for **b**–**e**: $n = 15$ mice). Time 0 on all graphs indicate start of imaging, on average 8 min after tracer infusion start. Thick line: mean, shading: SEM. **c** Boxplot of lymph node intensity at minute 50. Two-sided $t$ test, $p = 0.0335$. **d** Boxplot of the rate of lymph node filling calculated from the first 20 min recording. Mann–Whitney test, $p = 0.0321$. **e** Representative images from in vivo lymph node imaging across 50 min for day (orange) and night (blue). White scale bar: 2 mm. **f** Experimental outline. **g** Average fluorescence over time (thick line: mean, SEM: shading). **h** Boxplot of fluorescence in lymph nodes at 50 min, during active (green, for **f**–**j**: $n = 6$ mice) and rest (yellow, for **f**–**j** $n = 8$ mice) phases. Two-sided $t$ test, $p = 0.0231$. **i** Boxplot of rate of lymph node filling. **j** Representative time lapse images for animals housed in constant light for 10 days. White scale bar: 2 mm. **k** Diagram of experiment for lymph node contraction rate after FITC-dextran cheek injection, black dashed square: imaging area (left) with representative lymph vessel and trace of contractions from two regions of interest over 2 min (right). White scale bar: 1 mm. White dotted lines: area of measurement for traces. **l** Boxplot of contractions per minute of lymph vessels during the day ($n = 5$ mice) and night ($n = 7$ mice) in LD, and active ($n = 7$ animals) and rest ($n = 7$ animals) after 10 days in constant light. **m** Summary diagram of circadian regulation of CSF distribution. All boxplots minima: minimum value, maxima: maximum value, center: median, box and whiskers: quartiles, individual animals: colored dots. Asterisk indicates $p < 0.05$; ns: not significant. Source data are provided as a Source Data file.

evident in the mandibular lymph nodes has not entered the glymphatic system. If CSF had entered the glymphatic system, we would have expected a delay between influx into the brain and lymphatic drainage we measure in the lymph nodes. We also calculated the rate of lymph node filling during the first 20 min of recording, before the majority of recordings reached a plateau of fluorescent intensity (Fig. 5d, i). We found lymph nodes filled significantly faster during the night than the day (night: $2.04 \pm 0.4$ MPI min$^{-1}$, day $0.87 \pm 0.3$ MPI min$^{-1}$; Mann–Whitney test: $U = 46$, $p = 0.0321$) with a trend for this pattern to persist in constant conditions (active: $3.60 \pm 1.0$ MPI min$^{-1}$, rest: $1.81 \pm 0.6$ MPI min$^{-1}$; Mann–Whitney test: $U = 12$, $p = 0.1419$), indicating there are diurnal timing mechanisms for how fast the CSF can distribute into different compartments. We conclude that CSF drainage to the cervical lymph nodes is under circadian control with a peak during the active phase that temporally is in antiphase to CSF entry into the glymphatic system of the brain (Fig. 5m).

Lymphatic vessels contract to draw fluid into the lymph nodes[37]. We next tested whether circadian variation in mandibular lymph node filling was dependent on circadian differences in lymph vessel contraction rate (Fig. 5k, l). KX anesthetized mice were injected subcutaneously in the cheek with fluorescein isothiocyanate (FITC)-conjugated 3 kD dextran (0.25% in normal saline, 20 μL per cheek), then placed on their backs with the mandibular lymph nodes exposed via skin resection. Imaging acquisition began following lymph vessel filling (<2 min post cheek injection). In both a standard LD cycle (day: $n = 7$ mice;

night $n = 5$ mice) and in 10 days LL (rest: $n = 7$ mice; active: $n = 7$ mice) we found no significant differences in contractions per minute between the rest and active phases (two-way analysis of variance (ANOVA), main effect of phase: $F(1, 22) = 0.603$, $p = 0.4457$; interaction light*phase: $F(1, 22) = 0.173$, $p = 0.6815$). We conclude that intrinsic contraction rate of the mandibular lymph node vessels is not responsible for circadian drainage of CSF to the lymph nodes.

**Loss of AQP4 eliminates circadian CSF distribution.** We performed CM injections of CSF tracer (BSA647) into AQP4 KO mice during the day (ZT4–8; $n = 10$ mice) or night (ZT14–20; $n = 10$ mice) to test whether this day/night difference in AQP4 is necessary for day/night differences in glymphatic influx. AQP4 KO animals did not have a day/night difference in glymphatic influx, as measured by average slice intensity ($t$ test: $t(18) = 1.511$, $p = 0.1481$; Fig. 6a–c). Because lymphatic drainage shows circadian variation that was temporally antiphase to glymphatic influx, we did the same lymph node drainage assay described above in the AQP4 KO mice during the day ($n = 6$ mice) and night ($n = 7$ mice). There was no difference in lymph node drainage after 50 min between day and night ($t$ test: $t(11) = 1.103$, $p = 0.2937$; $n = 6$–7 mice), nor in the slope of drainage calculated from the first 20 min of recording ($t$ test: $t(11) = 0.9202$, $p = 0.3772$; Fig. 6d–h).

The reduction in day-time influx in AQP4 KO animals compared with littermate (LM) controls has already been

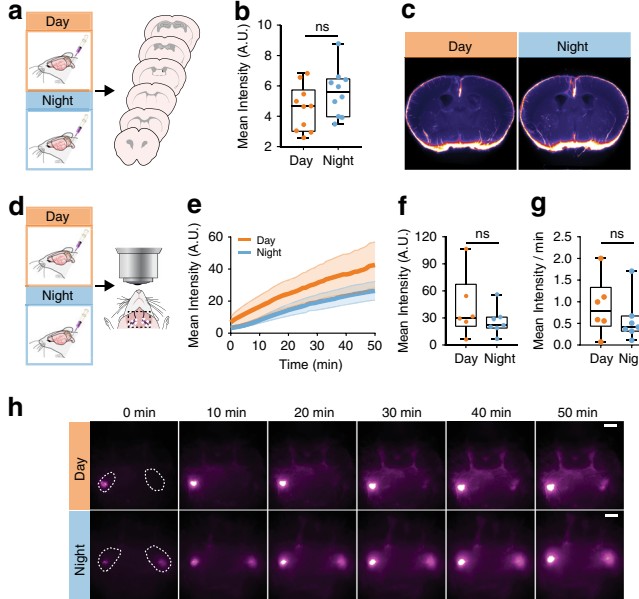

**Fig. 6 AQP4 KO animals lack a rhythm in glymphatic influx.**
**a** Experimental outline for **b**. **b** Boxplot of mean pixel intensity of slices from AQP4 KO animals during the day (orange) and night (blue). $n = 10$ mice per group. **c** Population-based averaging of slice intensity in AQP4 KO animals in the day and night. **d** Experimental outline. **e** Average fluorescence over time, mean is the thick line, SEM indicated by shading. **f** Boxplot of fluorescence in the lymph nodes at 50 min. **g** Boxplot of the rate of lymph vessel filling for the first 20 min. **h** Time lapse imaging across 50 min in AQP4 KO mice during the day (orange, for **d**–**h**: $n = 6$ mice) and night (blue, for **d**–**h**: $n = 7$ mice). White scale bar: 2 mm. Time 0 on all graphs indicate start of imaging, on average 8 min after tracer infusion start. All boxplots: minima is minimum value, maxima is maximum value, center is median and quartiles shown by box and whiskers, with individual animals shown as colored dots. ns: not significant. Source data are provided as a Source Data file.

published[7,8]. Next, we tested whether this persists during the night, and whether there are LM/KO day/night differences in drainage to the lymph nodes. Using the ex vivo tissue processing as described above, we found no significant difference in nighttime influx ($t$ test: $t(14) = 0.2208$, $p = 0.8284$) between LM ($n = 3$ mice) and KO animals ($n = 13$ mice; three mice in addition to those shown in Fig. 6b) (Supplementary Fig. 6a). Previous work has shown that LM controls for AQP4 KO animals have similar day-time influx compared to C57BL6 (C57) animals[8]. Because our animals are on a C57 background, we grouped the littermate controls with the nighttime C57 under KX from Fig. 1. All three LM fit within the median quartiles of the C57 data, indicating they are comparable. Though trending toward decreased influx in KOs, there was no significant difference between WT ($n = 37$ mice) and KO influx in the brain ($t$ test: $t(48) = 1.962$, $p = 0.0556$).

To measure lymph node drainage, we compared fluorescence in the mandibular lymph nodes after 30 min of CSF circulation during the day and night in LM ($n = 3$ mice per time point) and KO (day: $n = 9$ mice, night: $n = 10$ mice; three additional mice per group pooled with 30 min from pump start in Fig. 6) (Supplementary Fig. 6b). With the small sample size, we found no significant differences between genotypes (two-way ANOVA: F$(1,21) = 0.184$, $p = 0.672$), time (F$(1,21) = 0.191$, $p = 0.667$), or the interaction between the two factors (F$(1,21) = 0.902$, $p = 0.353$). Because our AQP4 KO line is backcrossed to C57, and the majority of LM fell within the two median quartiles of drainage

distribution, we next pooled LM with C57 values at 30 min post pump start (Fig. 5) (WT day: $n = 15$ mice, WT night: $n = 18$ mice). Overall, drainage to the mandibular lymph nodes was different between groups (Kruskal–Wallis test: H(3) = 8.464, $p = 0.037$; Fig. 6b). Prior to Bonferroni correction, WT day vs. night, and WT night vs. KO night comparisons were significantly different ($p = 0.020$ and $p = 0.011$, respectively) supporting increased drainage only during the night in WT mice, though this effect was gone after correction.

Overall, our observations support the hypothesis that daily changes in glymphatic function and drainage of CSF to the lymph nodes is supported by daily rhythms in polarization of AQP4 to the vascular endfeet of astrocytes.

## Discussion

Here, we show the difference in glymphatic function is not solely based on arousal state, but exhibits a daily rhythm that peaks mid-day when mice are mostly likely to sleep. Glymphatic influx under three different anesthesia paradigms (ketamine-xylazine, pentobarbital, and avertin), and an increase in clearance of EB from the brain parenchyma to the periphery indicate that glymphatic system function peaks during the day. Conversely, drainage of CSF tracer to the mandibular lymph nodes is highest in anesthetized mice during the night, when animal activity will be at its peak. Day/night differences in glymphatic influx, solute clearance, and CSF drainage to the lymph nodes persists under constant light, supporting the hypothesis that these are endogenous circadian oscillations. Circadian glymphatic function is supported by circadian regulation of AQP4 polarization in astrocytes, and genetic deletion of AQP4 effectively eliminates the circadian regulation of CSF distribution detected as an absence of day/night differences both in cortical tracer influx and in drainage of tracer to the lymph node.

Recent work has provided evidence that astrocytes in the clock center of the brain, the suprachiasmatic nucleus (SCN), may be responsible for setting periodicity of circadian behavior[38]. In addition, restoring molecular clock rhythms in SCN astrocytes is sufficient to drive rhythmic sleep/wake behavior in animals that are genetically arrhythmic[39]. Our data provide evidence supporting the idea that astrocytic regulation of glymphatic rhythms occurs at least in part via AQP4 vascular polarization in the cortex, which is supported by rhythmic gene expression of key components in the DAC. Deletion of the molecular clock in astrocytes induces mis-regulation of AQP4 gene expression, supporting our hypothesis that the DAC is under molecular clock control[40]. Circadian regulation of proteins in the DAC beyond AQP4 may provide insight into conserved astrocyte clock function controlling the release and buffering of extracellular signals such as glutamate, as in the SCN[41], or regulating sleep- and wake-promoting extracellular ion concentrations and fluid transport throughout the cortex[1,42]. Our data show that expression of connexin-43 (Cx43, *Gja1*) is diurnally regulated in a pattern similar to genes in the DAC, consistent with Cx43 being highly enriched in the vascular endfeet of astrocytes[43], and provide additional mechanisms for clock-controlled astrocytic signaling. These observations support the notion that astrocytes stabilize and perhaps drive circadian behavior, in part by regulating bulk fluid movement through and CSF/ISF exchange across the brain to entrain circadian rhythms.

CSF is produced in the choroid plexus (CP) of the brain, an epithelial layer of cells located within the ventricles. The CP exhibits robust cycling of the molecular clock in vitro, and can tune periodicity of molecular rhythms in SCN co-cultures and in behavior[44]. In addition, CSF production in humans may be rhythmic with peak CSF production during the night[45,46], though

results may be inconclusive[47]. This poses at least two potential paths the CP might regulate glymphatic rhythms: first, by regulating concentration of signaling molecules in the CSF and second by mechanically supporting glymphatic influx by providing a larger volume of CSF during the rest phase. When improved methodology for CSF production in rodents becomes available[48,49], it should be investigated whether CSF production regulates glymphatic CSF influx along the perivascular spaces of the brain.

Proulx and colleagues recently proposed a model whereby rapid CSF turnover through lymphatics precludes significant bulk flow into the brain[15]. This was based on two main observations: no tracer enters perivascular spaces of penetrating arterioles prior to death based on macroscopic imaging, and lymphatic outflow is increased in wakefulness, whereas glymphatic influx is decreased along the MCA. Our laboratory has already demonstrated CSF tracer alongside penetrating arterioles in live animals using two-photon microscopy[1,7,9], which has considerably higher spatial resolution than macroscopic imaging[10,50]. In addition, we have used rapid decapitation followed by drop fixation to process all of our brain samples since 2017 to reduce the death[15] and perfusion artifact[10] of CSF tracer localization in the brain. Here, we collected both in vivo and ex vivo data in fully anesthetized mice and observe changes in glymphatic and lymphatic CSF distribution dependent upon circadian timing, with underlying molecular rhythms in the DAC and AQP4 polarization. Based on these data, we propose that the lack of glymphatic influx is not simply a downstream effect of the lymphatic system drainage of CSF before it reaches the brain. Instead, the awake brain actively suppresses periarterial influx[10], has less perivascular AQP4, and prevents CSF/ISF exchange by reducing the interstitial space volume[1]. This would cause CSF to follow alternative, pre-existing routes out of the subarachnoid space around the brain and spinal cord, such as into the mandibular lymph nodes, meningeal lymphatics, and/or the deep cervical lymph nodes[13,51]. In short, CSF distribution will change based on fluid changes and space availability within the brain.

The mechanisms behind brain-regulated suppression of glymphatic activity during wakefulness remain unknown. EEG activity tightly correlates to glymphatic influx and the patterns of EEG activity that resemble wakefulness strongly suppress CSF influx[11]. One hypothesis is that the synchronous, slow wave firing characteristics of non-REM sleep drive consolidated movement of extracellular ions across cortex, supporting pulsating inflow of CSF into the neuropil. This process would not be compatible with the asynchronous patterns of neuronal activity that characterize the awake brain.

There are multiple circadian rhythms that may interact to promote rhythmic glymphatic function. Recent work hypothesized increased glymphatic clearance during the sleep phase, driving rhythmic fluid flow in the brain[52]. Cortical neuronal activity as measured by EEG has strong circadian components in humans[53,54]. The cardiovascular system as a whole is under tight circadian control[55], and arterial pulsatility is a key driving factor in transport of CSF along the penetrating arteries[9,10]. In flies, there is evidence of a circadian-clock in the equivalent of the blood–brain barrier[56,57]. Finally, immune system functionality is regulated by circadian timing[16,17,58] and we describe a clear interaction of CSF entry to the brain and lymphatic clearance of CSF. Although glymphatic function has yet to be studied in models of circadian disruption, such as in shiftwork, it has been established that shift workers are at increased risk for neurodegenerative disorders, cardiovascular disease, and exhibit increased markers of systemic inflammation[16,59–61]. Understanding how these rhythms, all with different timing and biological functions, interact to affect glymphatic function and lymphatic drainage may help prevent morbidity associated with circadian misalignment.

## Methods

**Animals.** Male and female C57BL/6 mice (aged 3–5 months, weight between 25 g and 30 g) were acquired from Charles River Laboratories (Wilmington, MA) in equal numbers for each experimental group to control for any potential sex differences. Male and female AQP4 KO mice[62] were bred in the University of Rochester vivarium, and backcrossed to C57BL/6 mice for 20+ generations before use. A minimum of five mice were used in each group, with exact animal numbers stated in the results and figure legends. Mice were group-housed either in a 12:12 light/dark cycle or under constant light with ad libitum access to food and water. All experiments were approved by the University of Rochester Medical Center Committee on Animal Resources. All of the University of Rochester's animal holding rooms are maintained within temperature (18–26 degrees Celsius) and humidity ranges (30–70%) described in the ILAR Guide for the Care and Use of Laboratory Animals (1996). All efforts were made to keep animal usage to a minimum.

For constant light experiments, animals were housed two per cage to reduce risk of hypothermia[63]. Activity was monitored continuously in 5 min bins via the Comprehensive Lab Animal Monitoring System (Columbus Instruments). Circadian behavioral analysis was completed with ActogramJ. Free-running period of each cage was determined using at least 10 days of activity and a $\chi^2$ periodogram, which was used to confirm behavioral rhythmicity and estimate experimental times in conjunction with activity onset of the cage.

**Drugs.** Anesthesia was administered as follows: 100 mg kg$^{-1}$ racemic ketamine and 20 mg kg$^{-1}$ xylazine kg i.p. (KX); pentobarbital 60 mg kg$^{-1}$ i.p.; 2,2,2-tribromoethanol (also known as Avertin) 120 mg kg$^{-1}$ i.p.. Depth of anesthesia was determined by the pedal reflex test; if the mouse responded to toe pinch, an additional one-tenth the initial dosage was given and the tracer experiment delayed until full unconsciousness was obtained. Directly prior to CM infusion, the animal received an additional one-tenth the initial dosage, and the pedal reflex was tested every 5–10 min during the tracer circulation time to ensure proper anesthesia throughout the study. Animals under avertin anesthesia were most likely to need supplemental dosing.

**Intracisternal CSF tracer infusion.** Fluorescent CSF tracer (bovine serum albumin, Alexa Fluor™ 647 conjugate; 66 kDa; Invitrogen, Life Technologies, Eugene, OR) was formulated in artificial CSF at a concentration of 0.5% weight by volume. Anesthetized mice were fixed in a stereotaxic frame, the CM surgically exposed, and a 30 gauge needle connected to PE10 tubing filled with the tracer was inserted into the CM. Ten microliters of CSF tracer was infused at a rate of 2 μl min$^{-1}$ for 5 min with a syringe pump (Harvard Apparatus)[10]. For the awake animal cohort: mice were anesthetized with 1–2% isoflurane, and 0.25% bupivicane was applied topically to the surgery site at the beginning of and after recovery from the procedure. Mice were allowed to resurface from anesthesia before pump start. Total volume of CSF tracer was increased to 12 μL to compensate for decreased glymphatic function, pump speed remained the same (2 μl min$^{-1}$).

To visualize tracer movement from the cisternal compartments into the brain parenchyma, the animals were killed by decapitation and the brain removed 30 min after the start of intracisternal infusion (note that the needle was left in place after the infusion to prevent backflow of CSF). The brain was fixed overnight by immersion in 4% paraformaldehyde in phosphate-buffered saline (PBS). Coronal vibratome slices (100 μm) were cut and mounted. Tracer influx into the brain was imaged ex vivo by macroscopic whole-brain and whole-slice conventional fluorescence microscopy (Olympus; Stereo Investigator Software).

Tracer influx was quantified by a blinded investigator using ImageJ software[11]. The cerebral cortex in each slice was manually outlined, and the mean fluorescence intensity within the cortical ROIs was measured. An average of fluorescence intensity was calculated between six slices for a single animal, resulting in a single biological replicate (Supplementary Fig. 2). Equivalent slices were used for all biological replicates.

**Transcranial imaging.** Animals were anesthetized with KX at ZT6 or ZT18 and were prepared for transcranial imaging[64,65]. In brief, animals were placed in a stereotaxic frame and the skin was removed from the top of the head to expose the skull. The animal was then placed under the macroscope (MVX10 Research Macro Zoom Microscope, Olympus). A CM injection was then performed as described above, with imaging beginning at pump start. Image acquisition was once per minute on the Cy5 channel, 30 min total (31 images per animal).

**Parenchymal clearance assay.** All experiments occurred between CT 2–7 or CT 14–19 depending on if it was the rest or active phase of the animal, respectively. Cannulas (guide cannula: 26 G, C315G SPC, 4.5 mm bellow pedestal, dummy: 33 G, C315DC/SP, 0.1 mm projection; PlasticsOne, Roanoke, VA) were implanted into mice 24–36 h prior to experimentation into striatum (AP: +0.6 from Bregma, LM: −2 and DV: −3.25 mm). At the appropriate experimental time points, mice in either LD or 10 days LL were anesthetized with KX and fitted with a microdialysis probe (inner cannula: 33 G, C315I/SP, 0.1 mm projection), had skin over the femoral vein resected to better visualize EB in the blood stream, and EB was injected into the brain (4% in ACSF, 1 μL, 0.2 μL min$^{-1}$). Images of the femoral

vein were taken under a macroscope (MVX10 Research Macro Zoom Microscope, Olympus) using the Cy5 filter set, for EB visualization, and GFP, for venous structure visualization, once every 15 min for 90 min (14 images per animal) beginning at the start of EB injection. Brains were collected upon experiment end, drop-fixed overnight in paraformaldehyde (PFA) and sliced the next day in the same manner as the CSF influx experiments. Slices were used to ensure parenchymal injections did not hit the ventricle and volume of injection, as measured by % area, was similar between animals. All images were analyzed in ImageJ. Mean pixel intensity in the femoral vein was quantified on the Cy5 channel, with the GFP channel used to ensure placement of the intra-venous region of interest.

**AQP4 immunohistochemistry and quantification**. Equivalent 100 μm thick brain sections between −1.2 mm to −1.8 mm from Bregma were selected for staining for each mouse. Slices were permeabilized with 0.1% Triton-X-100 in PBS, blocked with 7% normal donkey serum (Jackson Immunoresearch) in PBS with 0.03% Triton-X-100 and incubated with primary antibody overnight, followed by three washes in PBS and incubation with the fluorophore-linked secondary antibodies (Invitrogen) for 2 hours. Stained slices were mounted with Fluoromount G (Thermofisher Scientific). Primary antibody used was rabbit anti-AQP4 (AB3594, Millipore), secondary antibody used was Alexa 594 donkey anti-rabbit (A21207, Invitrogen, 1:500 dilution), cell nuclei were identified using DAPI (D1306, Invitrogen). A subset of animals were cardiac-perfused with wheat germ agglutinin conjugated with Alexa Fluor 488 (Thermofisher Scientific) at a working concentration of 15 μg mL$^{-1}$ in PBS, followed by 4% PFA to enable better visualization of the vasculature. The brains were harvested and processed for immunohistochemistry as described above.

All analyses were done with investigators blinded to time-of-day, as images were taken by a separate investigator who randomized file names. Equivalent images of dorsal cortex, lateral cortex, ventral cortex, were imaged for each slice (three images per slice). For a subset of animals, additional images were taken of CA1 in hippocampus and hypothalamus. The mean immunofluorescence intensity was quantified using ImageJ. For quantification of AQP4 polarization, 50 μm segments centered on blood vessels (as identified by vascular-shaped AQP4 localization) were analyzed using the line-plot tool in ImageJ. Vessels chosen were ~6 μm wide (range of 5–8 μm) and 32 μm long (range of 20–55 μm). Three vessels that matched this criteria were chosen per image.

For polarity calculation, the baseline was defined as the average intensity over 10 μm, −20 μm to −10 μm from peak fluorescence. Baseline fluorescence was subtracted from the peak intensity of the segment. This subtraction method is ideal because it looks at changes beyond background fluorescence localized to vascular endfeet. For each dataset (LD and LL), the adjusted peak fluorescence values were normalized to the highest value out of all groups compared in the statistical analysis, one value per graph for ease of comparison and visualization, giving a range from 0 to 1.

**Western blotting**. Mice were killed at ZT6 and ZT18, mice at ZT18 were killed in a dark room using red light lamps to avoid light pulses. Brains were extracted and cut in half along the sagittal plane with the cerebellum removed, one half was used for protein extraction. Brain tissue was homogenized in lysis buffer (25 mM Tris•HCl pH 7.6, 150 mM NaCl, 1% NP-40, 1% sodium deoxycholate, 0.1% sodium dodecyl sulphate (SDS)) with a protease inhibitor cocktail (1:100 ratio; Sigma Aldrich) using a Fisherbrand 150 homogenizer. Samples were kept at 4 °C or on ice, to avoid freeze thaw cycles. Samples were sonicated 3 × 10 s (Branson Sonifier 150D) and centrifuged at 13,000 × g for 5 min at 4 °C. Supernatant was transferred to a new tube and diluted 1:10 in lysis buffer. Protein concentration was determined by a Pierce BCA assay (Thermofisher). Samples were prepared for loading in sample buffer (100 mM β-mercaptoethanol and Laemmli Buffer, Bio-rad) at 20 μg protein per well. Dual Color Ladder (Bio-rad) was loaded to identify molecular weights. The gel was 4–15% Bis-acrylamide (Bio-rad). Gels were transferred to PVDF membranes (Bio-rad), then membranes were blocked in: Tris-buffered saline (Bio-rad), 0.1% Tween-20 (Sigma), and 5% fat free milk (Bio-rad), for 1 h on an orbital rocker. Membranes were incubated with primary Rabbit anti-AQP4 antibody (1:1,000 dilution; Chemicon, AB3594, Lot #: 3213917) in TBST 5% milk overnight at 4 °C. Then, incubated 1 h at room temperature with Donkey anti-Rabbit-HRP secondary antibody (1:10,000 dilution; Jackson Immunoresearch, #711-035-152, Lot#59390). Membranes were developed using Femto kit (1:20 dilution; Thermofisher) and imaged on a Chemidoc MP imaging system (Bio-rad). Membranes were stripped using stripping buffer (2% SDS, 62.5 mM Tris, pH 6.7, 100 mM β-mercaptoethanol) at 50 °C for 10 min. Membranes were blocked again, then incubated with mouse anti-βactin primary antibody (1:5000 dilution; Cell Signaling Technology, #12262 s, Lot #: 2) in TBST 5% milk at 4 °C overnight. Donkey anti-Mouse-HRP secondary antibody (1:10,000 dilution; Jackson Immunoresearch #715-035-150, Lot #: 143140) was applied for 1 h and the membranes developed and imaged.

**Quantitative reverse transcription PCR**. Brains were collected at ZT6 or ZT18, cut in half along the midline with the cerebellum removed, and flash-frozen on dry ice then stored in a −80 °C freezer until RNA isolation (n = 5 per group). RNA was isolated using Trizol®, cleaned via RNeasy Mini kit (Qiagen, Germany) following

manufacturer's instructions and quantified on a nanodrop spectrophotometer. First-strand cDNA was synthetized using TaqMan Reverse Transcription Reagents (Applied Biosystems, USA). Real-time PCR samples were prepared in triplicate with 5 ng of RNA in FastStart Universal SybrGreen Mastermix (Roche Diagnostics, Germany) and amplified on a CFX Connect Real-Time System Thermocycler (Bio-Rad, USA). Primer sequences are listed in Supplementary Table 1. Melting-curve analysis was performed following each PCR to confirm reaction specificity. Results were normalized within samples to 18s gene expression. Fold changes were calculated using the ΔΔCt method[66].

**Lymph node imaging**. Mice were anesthetized with KX between CT 4–8 or CT 16–20 and received a CM injection as described above. Upon completion of the tracer infusion (5 min) the mice were placed supine under the macroscope (MVX10 Research Macro Zoom Microscope, Olympus) with all skin over the neck resected to reveal the superficial lymph nodes. Images were taken on the Cy5 channel for 50 minutes at a rate of one image per minute, 51 images total (ORCA Flash 4.0 CMOS Camera, Hammamatsu). The speed of the lymph node filling (fluorescent intensity per min) was evaluated by the slope of a linear regression on the first 20 minutes of each recording (20 images).

**Lymph vessel contractility assay**. Mice were anesthetized with KX between CT 4–8 or CT 16–20. Mice had superficial lymph nodes exposed and were placed under the macroscope, as described above. FITC-conjugated 3 kD dextran (Invitrogen, 0.25% in normal saline, 20 μL per cheek) was injected into each cheek of the mouse. Imaging acquisition began following lymph vessel filling (<2 min post cheek injection) at a rate of 2.5 hz for 5 min (751 images per animal). Images were analyzed in ImageJ. A horizontal line segment centered on the lymph vessel was drawn and fluorescence over time was calculated using this region of interest. Contractions per minute (the number of oscillations of fluorescence, similar to ref. [37]) was calculated from each lymph vessel, and averaged for each animal.

**Population-based average videos and images**. For visual evaluation of the differences in tracer intensity and dynamics of the day and night animals, population-based averages were created. The dorsal view time series (Fig. 1b) were collapsed into videos, motion corrected and spatially aligned using ITK-SNAP[67]. At each time frame, the mean of each group was computed and displayed (Supplementary Video 1). For the coronal brain slices (Fig. 6c) an initial template was computed as the mean of the unaligned images. All slices were then registered to the template and a new mean computed. This iterative process was repeated eight times with increasingly refined registration methods (four iterations of rigid registrations, two of affine registrations, and two of non-linear registrations). Image registration was performed using Advanced Normalization Tools (ANTs) 2.1.0 and scripted with Python 3.6.

**Statistical analysis**. All statistical analysis was performed in Graphpad Prism version 7.0, with the exception of the cosinor analyses. Cosinor analyses[68,69] were performed using PASW Statistics 18. For comparisons of means in samples with normal distributions and homogeneous variances (as indicated by a Levene's test), an unpaired t test was used for two groups. For more than two groups an ANOVA was used for comparisons between means, followed by Bonferroni p value correction for multiple comparisons. In cases of a non-normal distribution (as indicated by a Shapiro–Wilk test) or unequal variances (Levene's test), a nonparametric Mann–Whitney test or a Welch's t test was used for comparisons between two means, and a nonparametric Kruskal–Wallis test was used for comparisons between more than two means, followed by Bonferroni p value correction for multiple comparisons. Significance was ascribed at p < 0.05.

**Reporting summary**. Further information on research design is available in the Nature Research Reporting Summary linked to this article.

## Data availability

All data needed to evaluate the conclusions in the paper are present in the paper and/or the Supplementary Materials. Additional data available from authors upon request. Source data are provided with this paper.

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

## Acknowledgements

We thank Dan Xue for graphic illustrations and animations. Funding: United States Department of Defense | United States Army | U.S. Army Research, Development and Engineering Command | Army Research Office (ARO) - MURI W911NF1910280 [Nedergaard]. U.S. Department of Health & Human Services | NIH | National Institute of Neurological Disorders and Stroke (NINDS) - R01NS100366 [Nedergaard]. U.S. Department of Health & Human Services | NIH | National Institute of Neurological Disorders and Stroke (NINDS) - RF1AG057575 [Nedergaard]. Fondation Leducq Transatlantic Networks of Excellence Program, Novo Nordisk and Lundbeck Foundations, and the EU Horizon 2020 research and innovation program (grant no. 666881; SVDs@target) [Nedergaard].

## Author contributions

L.M.H. and M.N. were responsible for experimental design. L.M.H., H.S.V., M.G., T.M., R.N., V.P., and A.B. were responsible for data collection. L.M.H., H.S.V., M.G., F.F.S., T.M., R.N., V.P., and A.B. were responsible for data analysis. L.M.H., M.G., and F.F.S. were responsible for figure preparation. L.M.H. and M.N. were responsible for manuscript writing and preparation. All authors have read and have approved the final version of this manuscript.

## Competing interests

The authors declare no competing interests.
