## [Peer Review File · Nature Communications]

Reviewers' comments:

Reviewer #1 (Remarks to the Author):

In the manuscript "Circadian control of brain glymphatic/lymphatic fluid flow", Hablitz et al. demonstrate that the speed of glymphatic uptake and lymph node filling follows a circadian-controlled rhythm, with glymphatic influx greatest mid-day and lymph node drainage most prominent mid-night cycle. These data are demonstrated thoroughly and concisely through quantification of a fluorescent tracer introduced into the cisterna magna in both live imaging and fixed sections of murine cortex. Glymphatic uptake was not dependent on light cycle and fluctuated in a pattern that paralleled mRNA expression of circadian clock and dystrophin-associated complex genes. AQP4 immunohistochemistry also demonstrated greater intensity colocalization with brain vasculature during the day (or rest period), corresponding with the greatest periods of glymphatic influx. This provides an important advancement in our understanding as to movement of CSF within and from the brain. There are some relatively minor issues that need to be addressed.

- 1) The methods of cortical and lymph node imaging over time need to be described in more detail. (Fig 1c, 4b/g, 5f) What is the time interval between data collection? How many data points were acquired and quantified per mouse? How many mice were imaged? How was the curve fit to the existing data points?
- 2) Sample size needs to be noted in the methods or figure legends for each experiment. This is especially true for figures 1c, 4b/g, and 5f which do not have visible data points.
- 3) It would be much easier to understand the methodology of quantifying tracer in brain sections if a representative image was provided with an outlined area of quantification associated with 1d and 3c. It would also be interesting to see how tracer distribution in coronal sectioning compares to the timelapse images/videos. Could similar images used in panel 5d, which averages the intensity of tracer in the brain for many samples, be presented for the data shown in 1d and 3c? If not, representative day/night images would be sufficient. Without these images, there is also nothing to compare panel 5d to, depriving it of relevance.
- 4) Would it be possible to compare anesthetic-free glymphatic influx with the various post-anesthesia tracer quantifications in figure 1d? This could provide relevant commentary on whether glymphatic influx is also influenced between asleep and awake animals.
- 5) 2b and 2c do not demonstrate changes in AQP4 polarization. If it did, we would expect to see more AQP4 external to the vasculature, or AQP4 in astrocyte bodies would need to be quantified. 2b/c only demonstrate that AQP4 signal intensity was universally lower in night images compared to day. Combined with the western blots in 2d-f and the supplement, a case can be made for AQP4 polarization under circadian control, but this is not what 2b/c demonstrate alone. Along these lines, the "polarization index" described in 2c is a bit misleading. Describe what it actually is in the figure legend – the ratio of max:baseline fluorescence.
- 6) It is relevant to cite whether the clock and DAC genes in figure 2h are astrocyte-specific from published databases. This would help explain if the day/night differences originate from astrocytes and astrocyte AQP4 localization or whether they can be attributed to non-astrocyte cell types.
- 7) Why was constant light used for Figure 3, when constant darkness is the typical standard to

isolate the circadian clock? Please provide a rationale in the text.

8) Figures 3 and 4 were easy to understand and well put-together, clearly demonstrating circadian control of glymphatic influx and lymph node drainage in the context of Figures 1 and 2. However, Figure 4 was very small in the PDF, and may require reorganization to accommodate a printed version.

9) The descriptions of the data presented in Figure 5 are dramatically abbreviated in the text. These data are important to tie the clearly-defined circadian glymphatic/lymphatic accumulation patterns to corresponding regulation of AQP4 localization and DAC gene expression. A more thorough explanation would be appreciated.

10) The gene measured in Figure 5a is not described anywhere in the methods, legend, or text. Assuming the gene is AQP4, why do AQP4 $-/-$ mice have 1 copy number of this gene and AQP4 $+/+$ mice have 0?

11) In Figure 5, littermate controls are necessary to account for strain differences (AQP4 $+/+$, AQP4 $+/-$, or cre-deficient animals, whichever is most relevant), as this data cannot be compared to Fig 1-4 without evidence of normal circadian control in littermates.

12) Nowhere in the results or the discussion is it postulated what anatomical pathway the tracer travels from the cisterna magna to the lymph nodes. It is mentioned that this pathway may be independent of the glymphatics due to the lack of delay in tracer accumulation after imaging begins. While experiments to determine the answer may be beyond the scope of this paper, discussion of the potential pathways would contribute to our understanding of CSF outflow dynamics.

Reviewer #2 (Remarks to the Author):

In this interesting manuscript, the authors address the question of whether there is an underlying circadian rhythm to CSF flow, assessing both CSF influx into perivascular spaces of the brain and CSF drainage to the lymphatic system in anesthetized mice at different times of day. This study raises important points, including the relevance of reporting the time of day experiments are carried out.

However, a number of issues should be addressed by the authors, mostly involving data interpretation and the need for more detailed description of some methods used.

Major:

1. Glymphatic influx vs glymphatic system

The authors should precisely and consistently identify what they are measuring. They speak of the "glymphatic system" and "glymphatic function" and of "glymphatic influx", however, experimentally only "glymphatic influx" is assessed, with no evidence presented of an effect of circadian rhythm on other components of the glymphatic hypothesis including flow through the parenchyma and outflow from the brain tissue.

Furthermore, we ask that the authors acknowledge the controversy surrounding the glymphatic hypothesis and not present this as irrefutable fact (line 43-44, line 62-64).

2. Influx into the brain vs flow along perivascular spaces at the brain surface

In different parts of the manuscript, "glymphatic influx" is assessed in different ways: either in

vivo by visualizing tracer in the perivascular spaces around the middle cerebral artery at the surface of the brain, or ex vivo in post-mortem brain sections. The authors should make clear for which parts of the study each method was employed and why these different approaches were used. Methods in vivo and ex vivo are not interchangeable, and the authors should avoid general phrases such as "influx into the brain" (line 94, 116, 191) when the data are showing influx of tracer to the perivascular spaces of the middle cerebral artery at the surface of the brain and not to the brain parenchyma (line 94) or when both types of influx are referred to (line 116, 191). There are reports (Ma et al., 2019) that some CSF influx into the brain appears to occur only after death rather than in vivo and this limitation of using post-mortem brain sections to assess CSF influx into the brain should be mentioned. The authors' results also appear to reflect differences between the magnitude of influx to the PVS on the brain surface and influx into the brain itself, as the difference between day and night was far greater at the brain surface in vivo (53%, line 93) than in the post-mortem brain sections (22%, line 105). Wouldn't one expect that if a one-directional flow from the perivascular spaces on the surface proceeded to brain tissue as proposed by the glymphatic concept that the differences between groups would be more comparable? Could this discrepancy be related to the post-mortem artifact?

3. Imaging through the skull and analysis

While no description of the analysis of tracer spread at the brain surface in vivo is given (see comments under Methods below), the ROIs indicated on Supplemental video 1 appear to include signal which is emanating from CSF spaces outside the surfaces of the cortex such as the quadrigeminal and olfactofrontal cisterns. If the authors wish to quantify CSF influx to the perivascular spaces at the brain surface, rather than spread of tracer from the cisterna magna to these other CSF compartments, the analysis ROI should be limited to the middle cerebral artery regions.

In Supplemental video 2, the 4th mouse in the "Day" group appears to show much more tracer spread than any of the others. Is this a statistically significant outlier? If so, what do the analysis and averaged videos look like if this mouse is excluded?

4. Circadian changes yes - but do these follow a sinusoid curve?

We read with interest that CSF tracer influx into brain surface perivascular spaces in vivo and on ex vivo brain sections shows a diurnal variation, and appreciate the results presented showing differences between time points. However, when examining the graphs in Fig. 1d, both individual data points and means at each time point, the idea that the circadian variation follows a cosinor curve (although the authors say this is significant) is not entirely convincing. Could the authors further elucidate why this type of curve fitting was chosen and acknowledge that the curves indicated by dotted lines may not be a perfect fit (especially for pentobarbital and avertin)?

5. Circadian rhythms in CSF production or lymphatic function in mouse?

The authors cite studies which assessed diurnal variations in CSF production in humans (line 233). It is important to note that this early phase contrast MRI work has limitations and this particular finding of increased CSF production at night in humans was unable to be reproduced in a more modern study (Takahashi et al, 2011). Does CSF production in mouse change with circadian rhythm or activity state? This is key to the interpretation of the experimental results in this manuscript, as changes in CSF production will affect its flow. Relevant studies should be cited and this should either be addressed experimentally by the authors or clearly mentioned as a limitation. Furthermore, it would be of interest if AQP4 KO affects CSF production. Again, this should be addressed or mentioned as a limitation.

Related to this: Line 238: "brain states regulate glymphatic influx independently of the endogenous rhythm in CSF production. For example, waking an animal up during daytime will suppress glymphatic influx, while anesthesia immediately increases influx in awake mice". The authors should rephrase this to clarify the logic behind this sentence, which does not address what

happens to CSF production in those conditions, independently of its endogenous rhythm. Again, whether there is a circadian variation in CSF production in mice should be clarified.

Lymphatic function in general is highly linked to activity state. Are there circadian variations in lymphatic function? These factors are also very relevant and important for the interpretation of the results presented in the manuscript.

6. Relationship between CSF influx and CSF efflux

Line 245. The data in this manuscript do support the conclusion by Proulx and colleagues that CSF influx to the perivascular spaces and CSF clearance to lymphatics are inversely correlated. We disagree that the data in this manuscript support the conclusion that glymphatic influx is reduced and lymphatic drainage peaks "during wakefulness". To our knowledge, no awake mice were used in this study. Please rephrase to clarify that the results of the manuscript indicate reduced CSF influx and increased lymphatic drainage during the mouse's active phase compared to its rest phase in anesthetized mice.

Line 245. We disagree that the conclusion that data in this manuscript "do not fit into the model". Circadian regulation and AQP4 localisation were not a component of the model proposed by Proulx et al. These additional components are not in rejection of the model but rather add further facets to what is clearly a complex process. Overall, with the exception of the AQP4 KO data (although see comment 7 below) the data is consistent with the model proposed by Proulx and colleagues in which CSF clearance to lymphatics and potential influx to perivascular spaces are inversely correlated.

Line 250: "Instead, lymphatic drainage of CSF is a shunt that opens when the awake brain is actively suppressing glymphatic influx...". Please rephrase to clarify that this is a hypothesis/proposed mechanism. The authors of this manuscript, and others, have shown that there is indeed CSF drainage to lymphatics during sleep/anesthesia/resting phase (e.g. Fig. 4). Thus, it appears that the proposed "shunt" is not "closed" in these conditions. How do the authors reconcile this with their proposed mechanism?

7. Accounting for overall reduced signals in the AQP4 KO mice

In Figure 5, the combined intensity of fluorescence of the lymph nodes and the ex vivo brain sections of the Day/Night AQP4 KO groups appears to be lower overall than that in the Day/Night and Active/Rest WT groups. Can the authors account for where the tracer is in the AQP4 KO mice, if it is not being carried by a substantial CSF influx into the brain or cleared to the lymphatics? One potential explanation could be that more tracer is retained in the CSF space (indicating perhaps a slower turnover in AQP4 mice). Did the authors assess this in their study?

Minor:

1. Figures

Color coding throughout: The authors have used different colors to label the groups of mice used throughout, which is a great idea and very much aids in legibility of the results. However, the fact that the "Day" and "Rest" groups are both color-coded in orange leads to some confusion. From our understanding, this is not the same experimental group, and we advise that different colors be used.

Graphs throughout: The authors have displayed data in graphs in different formats throughout, e.g. individual data points with mean and SEM, min/max dotplots with or without shading of bars, mean with SEM but no individual data points. We ask that the authors either use the same type of column graph throughout, or otherwise give convincing explanations why different types of data display are needed. Furthermore, unless there is a plausible reason otherwise, graph axes should start at 0 or include an axis break.

Figure 1:

Legend e. The curves shown appear to be sine rather than cosine curves.

Figure 2:

By eye, the AQP4 images from the "Day" group look less polarized, as there is evidently more AQP4 staining which is not localized to the vessel. A clarification of the analysis method would help to understand how the polarization index data were obtained (see comments under Methods below).

Perhaps, the authors have found a difference in AQP4 localisation based on their polarization index, but not in the total levels of AQP4 protein because they have studied localization in the cortex only, while Western blotting was carried out with whole brain lysates excluding only the cerebellum? This limitation should be mentioned.

We believe the graph in 2b should be labeled "Distance" or "Distance From Vessel" instead of "Vessel Length".

Figure 3:

3b and c. Did the authors also carry out in vivo imaging through the skull in the Rest/Active groups? If not, why was only ex vivo analysis of tracer influx carried out?

3d. Why is DAPI staining shown here? We do not feel this is necessary and that it obscures the visibility of the AQP4 staining. The presentation with lectin staining, as in Fig. 2a, was much easier to interpret.

Figure 4:

4b, 4g. The graphs of lymph node uptake of tracer (4b, 4g; also 5f) do not seem to start at Time=0 as there is a gap between the x-axis and the start of the plotted line. Is this a feature of the graph style or are there no data points near T=0? If the latter is the case, how did the authors calculate the slope from T=0 to T=20? The inset graph in 4b of the slope from 0 to 20 minutes is not essential, and in its current size not legible.

Please clarify if time "0 minutes" refers to the start of imaging or a specific time after tracer infusion. Displaying the data relative to the time after tracer infusion would more readily allow the reader to appreciate the real time course of tracer clearance to lymph nodes.

4c, 4h. Could the authors clarify at which time point the mean intensity values of the lymph nodes were calculated? The values appear to differ from those shown at the end of the in vivo imaging (50 min). In Fig. 1c, the values for the imaging through the skull appear to match between the end of the dynamic imaging and the inset graph with the statistical analysis. This comment would also apply to Fig. 5f vs. 5g.

4k. The labels and color coding of the schematic should be changed to also include the Day/Night groups as the conclusion is the same.

Please clarify why only the submandibular lymph nodes were chosen for analysis (accessibility for in vivo imaging??), and why deep cervical lymph nodes, which have recently been reported to be the major lymphatic site of CNS drainage (Raper et al., 2016; Ahn et al., 2019), were not assessed.

Figure 5:

5a: We believe this panel is not necessary and could be removed completely or moved to a supplemental figure.

5c and d: Again, did the authors also carry out in vivo imaging through the skull in the AQP4 and AQP4 KO groups? If not, why was only ex vivo analysis of tracer influx carried out?

5f: Please add quantification of the slope from 0 to 20 min as done in all other groups (Fig 4).

2. Methods

Overall, we have found the information on methods and n numbers for individual experiments laborious to gather as it is scattered between Methods section, figure legends and text.

Please add a description of the method used for imaging through the skull and for the quantification of tracer spread at the brain surface in vivo.

Line 314: Please add more detail of the method used for AQP4 quantification. How many slices were stained per mouse? Were equivalent slices taken? How many images were taken? Were these taken in a standardized location within the cortex, which has been reported to have heterogenous vascular density? Which blood vessels were chosen for analysis? How many blood vessels were analysed per mouse/section? Why were images taken at different magnifications and different staining carried out for Fig 2a and Fig 3d? Was the analysis also different, as the baseline level of staining intensity appears to be lower (1 in 3e compared to 3 in 2b)? Please clarify how the polarization index was calculated, as based on the description given, for e.g Fig 2b we would expect a value of 20 (peak) divided by 4 (baseline) giving an index of 5. However, the values in 2c are much lower. Please also clarify how the start of the vessel was identified in order to determine the 10 microns "before the vessel" (line 325).

Line 326: in the figures, the wheat germ agglutinin staining is referred to as "Lectin". Please also add this term in the text here.

Line 370: please define "MVX". Please also clarify that the entire neck skin was cut away, and not just the epidermis at the skin surface.

3. The term "mesor" in line 114 appears to be incorrect, as the values given seem to relate to the time of phase/acrophase/peak/maximum rather than the mesor value.

4. Line 133. Please rephrase to replace "is necessary for" with "is linked to", "is related to", ... or similar.

5. Line 180-181. Please clarify that CSF does not drain "directly" from the cisterna magna to lymph nodes, but first circulates through subarachnoid spaces associated with the brain or spinal column.

6. Line 189. Please rephrase to make clear that while CSF drainage and CSF entry are temporally related, this relationship is inverse.

7. Line 184. Are the images of the signal with the lymph nodes saturated? If so, this does not mean that the CSF drainage to the nodes has reached a plateau. It only indicates that at the camera settings used, the detected fluorescence intensity has reached a plateau and is no longer increasing.

References

Ahn J., et al. Meningeal lymphatic vessels at the skull base drain cerebrospinal fluid. *Nature*. doi:10.1038/s41586-019-1419-5
Ma, Q., Ries, M., Decker, Y., Müller, A., Riner, C., Bücker, A., Fassbender, K., Detmar, M., Proulx, S.T., 2019. Rapid lymphatic efflux limits cerebrospinal fluid flow to the brain. *Acta Neuropathol.* 137, 151-165. doi:10.1007/s00401-018-1916-x

Raper, D., Louveau, A., Kipnis, J., 2016. How Do Meningeal Lymphatic Vessels Drain the CNS? *Trends Neurosci.* 39, 581-586. doi:10.1016/j.tins.2016.07.001

Takahashi, H., Tanaka, H., Fujita N., Murase K, Tomiyama, N. Variation in supratentorial cerebrospinal fluid production rate in one day: measurement by nontriggered phase-contrast magnetic resonance imaging. *Jpn J Radiol* (2011) 29:110-115 doi:10.1007/s11604-010-0525-y

Reviewer #3 (Remarks to the Author):

This manuscript describes a series of experiments testing the time of day control of brain glymphatic/lymphatic fluid flow.

There are a number of strengths including tracer analysis every 4 hours for 24hrs with mice under different types of anesthesia (Figure 1). These are labor intensive experiments requiring attention to detail as well as significant time and resources. The outcomes of these experiments provide convincing data for changes in fluid flow over time of day with highest flow during the light/rest phase.

Experiments in Figure 2 show that the change in flow is correlated with changes in Aqp4 polarization with no change in protein content. Day:night differences in mRNA expression of members of the dystrophin associated complex are also reported.

The concern comes with the design of the experiments in Figure 3 in which the mice are placed into constant light (LL) conditions. It is not clear why they chose LL vs. total darkness (DD). It is most common in the circadian field to use DD to test if an outcome is likely circadian. Also, there are several studies that indicate that constant light is associated with behavioral and physiological arrhythmicity and studies by Ohta et al., (*Nature Neuroscience* 2005) suggested this occurs via desynchronizing clock cells in the SCN. The data in Figure 3 show rhythmic behavior under LL but the period length is >24hr rhythm which is not their normal clock rhythm (< 24hrs for C57bl6 mice). In addition they measured cage activity with 2 mice/cage?? With these design issues the authors can conclude that the changes in fluid flow do not require a light:dark cycle but caution should be used to make interpretations to the circadian clock.

We would like to thank the reviewers for their constructive feedback on our manuscript. Several key points have been addressed, including:

- Expanded rationale for using 10 days constant light (LL) to measure circadian CSF distribution.
- New behavioral analysis comparing circadian behavior of mice in 10 days constant dark or LL.
- New analysis of CSF influx time courses using CircWave to test rhythmicity.
- New analysis of *in vivo* transcranial imaging specific to the middle cerebral artery.
- A new dataset of awake mouse CSF influx during the day and night.
- New datasets of glymphatic clearance in both a light/dark (LD) and LL light cycles.
- A new dataset of mandibular lymph vessel contractility in both LD and LL.
- New quantification of AQP4 polarization in subcortical brain regions.
- A new figure to explain our *ex vivo* slice analysis.
- Statistical comparison of CSF influx in WT and AQP4 KO animals.
- Significant improvements to the methods and discussion sections.

Below, we have addressed each reviewer's concerns and questions to the best of our abilities. We look forward to your feedback and would appreciate consideration for publication in your journal.

Reviewers' comments:

Reviewer #1:

1) The methods of cortical and lymph node imaging over time need to be described in more detail. (Fig 1c, 4b/g, 5f) What is the time interval between data collection? How many data points were acquired and quantified per mouse? How many mice were imaged? How was the curve fit to the existing data points?

We have expanded upon our methodology for both transcranial imaging and lymph node imaging, as well as added references for previous studies that included the transcranial imaging methodology. We have also included time-of-day, frame rate, and total frames per mouse for each technique. Additionally, we have included sample size not only in the results section of the paper, but in figure legends.

Transcranial imaging

Animals were anesthetized with KX at ZT6 or ZT18 and were prepared for transcranial imaging as described previously^{1,2}. In brief: animals were placed in a stereotaxic frame and the skin was removed from the top of the head to expose the skull. The animal was then placed under the macrocope (MVX10 Research Macro Zoom Microscope, Olympus). A CM injection was then performed as described above, with imaging beginning at pump start. Image acquisition was once per minute on the Cy5 channel, 30 minutes total (31 images per animal, including time 0). [Page 16; Lines 409-415]

Lymph node imaging

Mice were anesthetized with the ketamine/xylazine cocktail between CT 4-8 or CT 16-20 and received a CM injection as described above. Upon completion of the tracer infusion (5 min) the mice were placed supine under the macrocope (MVX10 Research Macro Zoom Microscope, Olympus) with all skin over the neck cut away to reveal the superficial lymph nodes. Images were taken on the Cy5 channel for fifty minutes at a rate of one image per minute, 51 images total (ORCA Flash 4.0 CMOS Camera, Hammamatsu). The speed of the lymph node filling (fluorescent intensity/min) was evaluated by the slope of a linear regression on the first twenty minutes of each recording (20 images). [Page 20; Lines 499-507]

Lymph vessel contractility assay

Mice were anesthetized with ketamine/xylazine between CT4-8 or CT16-20. Mice had superficial lymph nodes exposed and were placed under the macrocope, as described above. FITC-conjugated 3 kD dextran (Invitrogen, 0.25% in normal saline, 20 μ L per cheek) was injected into each cheek of the mouse. Imaging acquisition began following lymph vessel filling (less than 2 minutes post cheek injection) at a rate of 2.5 hz for 5 minutes (750 images/animal). Images were analyzed in ImageJ. A horizontal line segment centered on the lymph vessel was drawn and fluorescence over time was calculated using this region of interest. Beats per minute (the number of oscillations of fluorescence) was calculated from each lymph vessel, and averaged for each animal. [Page 20; Lines: 508-517]

2) Sample size needs to be noted in the methods or figure legends for each experiment. This is especially true for figures 1c, 4b/g, and 5f which do not have visible data points.

We have included sample size not only in the results section, but also in all the figure legends.

3) It would be much easier to understand the methodology of quantifying tracer in brain sections if a representative image was provided with an outlined area of quantification associated with 1d and 3c. It would also be interesting to see how tracer distribution in coronal sectioning compares to the timelapse images/videos. Could similar images used in panel 5d, which averages the intensity of tracer in the brain for many samples, be presented for the data shown in 1d and 3c? If not, representative day/night images would be sufficient. Without these images, there is also nothing to compare panel 5d to, depriving it of relevance.

We have included representative images (Fig. S1b, below) and a new supplemental figure on which slices were taken with representative regions of interest per slice (Fig. S2, below).

Multiple studies have validated both transcranial macroscopic imaging and ex vivo slice analysis as two comparable measures of CSF movement across murine cortex during the day (Fig. 1, S1), when altering hypertonicity (See Fig. 1-3, S2, S3, and S10 in Plog et. al.², <https://insight.jci.org/articles/view/120922>), and between WT and AQP4KO animals (See Fig. 5 in Mestre et. al.³, <https://elifesciences.org/articles/40070>). There has yet to be any evidence of ex vivo tissue analysis not corroborating findings from transcranial imaging.

Fig. S1. Glymphatic influx has a diurnal rhythm in anesthetized mice. (a) Representative images for *in vivo* recordings of CSF tracer influx after injection during the day (orange) and night (blue) under ketamine/xylazine (KX) anesthesia (left). Dotted white line indicates region of quantification. (right) Mean pixel intensity in arbitrary units (A.U.) over 30 min post CM injection. Thick lines indicate group means with SEM outlined. The inset is the fluorescence at 30 min with min/max boxplots and individual animals represented as dots. Time 0 is tracer infusion start. $n = 6$ mice per group. $*p < 0.05$. (b) Representative slices for CSF tracer influx during the day or night under KX anesthesia. (c) Time course of mean pixel intensity from ex vivo coronal sections under KX (violet, $n = 10-12$ mice per time point), Pentobarbital (red $n = 7-11$ mice per time point), and Avertin (gray $n = 7-11$ mice per time point). Each colored point is an animal. Dotted line indicates CircWave (CW) fitted curve. Black point with bars are estimated Center of Gravity (CoG) from the CW fit with standard deviation. Light cycle is indicated by orange (day) and blue (night) coloring in the background. Zeitgeber Time (ZT) in hours, where ZT0 is lights on and ZT12 is lights off. (d) CW curves and CoG (left) for KX, pentobarbital (P) and avertin (A). Mesor-aligned CW and Cosinor curves for the three anesthetics (right), dotted lines indicate cosinor fit, solid lines indicate CW fit. (e) 95% confidence intervals from the cosinor analysis, CoG estimates from CW, and peaks from CW curve for phase and mesor for each anesthetic. (f) Schematic of experiment and min/max boxplot of average mean pixel intensity for awake mice receiving a CM injection during the day (white, $n = 7$ mice) and night (gray, $n = 9$ mice), individual mice represented by single dots, ns: not significant. (g) Min/max boxplot of mean intensity from anterior to posterior slices. Individual animals represented by dots. Slice diagrams shown above. (h) Min/max boxplot of mean intensity for hippocampus (HC), thalamus (TH), hypothalamus (HT), dorsal cortex (DC), lateral cortex (LC), and ventral cortex (VC). Individual animals shown by dots. Schematic of subregions shown in inset.

Fig. S2. Overview of slice analysis. (Top) Cartoon images of the six slices normally collected after CM injection of tracer. (Middle) Representative images from a brain receiving a CM injection of BSA647. Tracer shown in purple and white, yellow dotted line indicates ROI drawn around slice. (Bottom) Estimated distance from Bregma of each slice.

4) Would it be possible to compare anesthetic-free glymphatic influx with the various post-anesthesia tracer quantifications in figure 1d? This could provide relevant commentary on whether glymphatic influx is also influenced between asleep and awake animals.

Our group has shown, several times, that wakefulness essentially shuts-down glymphatic influx of CSF along the perivascular space^{2,4,5}. Due to the wakefulness effect, we did not expect to find day/night differences in influx. However, a new manuscript using contrast-enhanced MRI in restrained, awake rats reports increased influx of intraventricularly injected contrast agent in the midbrain, hippocampus, hypothalamus, pons, and thalamus during the night⁶. We attempted to replicate the findings of Cai et. al. in mice with cisterna magna delivery of tracer.

Because influx is extremely low in awake animals we increased the amount of tracer infused from 10 μL to 12 μL . We placed the cannula into the cisterna magna under isoflurane anesthesia, followed by local administration of bupivacaine as analgesia. The mouse was then restrained in a tube and allowed to wake up. The injection rate of BSA647 was 2 $\mu\text{L}/\text{min}$ for 6 min. Brains were harvested at 30 min after start of tracer infusion. Brains were drop-fixed overnight in PFA then processed similar to our time course experiments. [Page 16; Lines: 392-396]

We found no significant difference in average CSF influx between day (ZT4-8) and night (ZT16-20). Because Cai et. al. found the most significant differences after subregion analysis, we next calculated average tracer signal in dorsal cortex, lateral cortex, ventral cortex, hippocampus, thalamus, and hypothalamus. We found no significant difference between day and night in any subregion.

The rat study had several caveats including: Disrupting sleep in the light phase animals for at least 5 days of acclimation to the MRI, a procedure that will lead to a sleep loss in one but not the other group. Also, there is evidence that T1-weighted images have lower background under sleep deprivation conditions⁷. Since the authors did baseline subtraction to see the difference the time-intensity-curve, they may have significantly under-estimated the light phase signal intensity. Given this information, it is unsurprising we were not able to replicate their findings. Interestingly, the authors hypothesize that the day/night differences were due to increased clearance of tracer during the light phase. Although they did not show data for this hypothesis, our data supports this conclusion.

Overall, based on our data and previous work, we conclude that awake glymphatic influx is overall low and highly variable with no clear day/night difference, and that glymphatic influx is most important in anesthetized or sleep states. These data have been added as supplemental results. [Fig. S1fgh (above); Page 6; Lines: 131-135]

5) 2b and 2c do not demonstrate changes in AQP4 polarization. If it did, we would expect to see more AQP4 external to the vasculature, or AQP4 in astrocyte bodies would need to be quantified. 2b/c only demonstrate that AQP4 signal intensity was universally lower in night images compared to day. Combined with the western blots in 2d-f and the supplement, a case can be made for AQP4 polarization under circadian control, but this is not what 2b/c demonstrate alone. Along these lines, the “polarization index” described in 2c is a bit misleading. Describe what it actually is in the figure legend – the ratio of max:baseline fluorescence.

There was an error in our methods section leftover from a previous draft of the paper, thank you to Reviewer 1 for pointing out discrepancies in our descriptions of interpretation and methodology. When it was first written

we did the ratio analysis, but then realized that there may be increased background in the day animals as Reviewer 1 has pointed out. This led to re-calculation of vascular-localized AQP4 signal by subtracting the baseline intensity (defined as the average intensity over 10 μm , $-20\ \mu\text{m}$ to $-10\ \mu\text{m}$ from peak fluorescence) from the peak intensity within the vascular end-feet. This method would take into account intensity changes between slices or time points. We then took all values in the dataset and divided by the highest value to give us a range of 0 to 1 in order to make the data easier to visualize. It is also the reason we did western blotting for AQP4, to determine if the background increase was increased AQP4 protein. We have edited the methods section to fix this error [Pages 17, 18; Lines: 433-460], and have added further clarification on the methodology in the figure legends for Fig. 3 and 4 (formerly Fig. 2 and 3). Furthermore, we tested whether polarization to the vasculature was increased in hypothalamus and CA1 of the hippocampus in a small subset of animals. We found strong trends for increased polarization during the day. We have added this to our results section [Fig. S3 (below); Page 7; Lines: 157-159].

Fig. S3. Quantification of AQP4 polarization in Hypothalamus and Hippocampus. (a) Schematics of the hypothalamus (HT) and CA1 of the hippocampus where in the brain images in (b) were taken. Dotted boxes indicate regions of interest for imaging. (b) representative images for HT and CA1 during the day (orange) and night (blue) for AQP4 (magenta) polarization analysis. Dotted lines indicate pyramidal cell layer (PCL) in hippocampus. (c) Average intensity of AQP4 staining centered on vasculature in HT (top) and CA1 (bottom) during the day (orange) and night (blue), $n = 4$ mice, 36 vessels per group, with SEM indicated by shading around the thick line. (d) Average polarization index min/max boxplots with individual mice shown as colored dots. Polarization index = peak vascular end foot fluorescence - 10 μm baseline. All values were normalized to the highest signal for ease of visualization.

6) It is relevant to cite whether the clock and DAC genes in figure 2h are astrocyte-specific from published databases. This would help explain if the day/night differences originate from astrocytes and astrocyte AQP4 localization or whether they can be attributed to non-astrocyte cell types.

While clock genes are prevalent in all cell types, DAC genes are distinctly enriched in astrocytes⁸. Specifically, DTNA, DMD, DAG1, AQP4, GJA1, GJB6, MLC1, and SLC4A4 are all upregulated in astrocytes at minimum 2 times more than any other cell type. SNTA1 has higher expression in microglia (FPKM value of approximately 90 compared to 20-30 in astrocytes), which could indicate extra-DAC functions. Thus, we conclude this is an astrocyte phenomenon. Additionally, work with conditional BMAL knockout in astrocytes shows that AQP4 gene expression is altered⁹, supporting this conclusion. We have added these points to the results [Page 7; Lines: 170-172] and discussion [Page 12, Lines: 296-298], and included the genes specifically enriched into the legend of Fig. 3 (formerly Fig. 2).

7) Why was constant light used for Figure 3, when constant darkness is the typical standard to isolate the circadian clock? Please provide a rationale in the text.

We have added rationale for constant light over constant dark to our results section [Page 8; Lines: 192-194].

We would like to emphasize that we find significant active/inactive differences in glymphatic influx to the brain and drainage of CSF to the lymph nodes, even with any potential caveats of the 10d constant light (LL) paradigm (Fig. 4, 5), indicating an endogenous, circadian rhythm of this physiological process. We chose LL over constant dark (DD) to avoid effects of light pulses that are inevitable during live-animal surgery. The surgery takes approximately 10 minutes, more than enough to phase shift mice in constant darkness^{10,11}. The mechanisms of light altering CSF influx to the brain is an ongoing line of research for our lab.

The observed average period of 25h in LL (Fig. S5ab) is positive because it shows that the animals are truly free-running in their environment. It is incorrect to assume that mice have a free-running period of less than 24h in DD. This is true for animals with a running wheel, but without that nonphotic cue periodicity is very close

24h (Fig. S5ab)¹²⁻¹⁷. This is a problem for determining if the rhythm of influx is “circadian” compared to “diurnal” because you cannot differentiate between an endogenous rhythm or a lack of controlling for other entrainment cues^{18,19}.

When designing the experiments, we first assessed the circadian behavior in DD and LL. Using chi-squared periodogram analysis we found animals in DD and LL had similar variability in periodicity, and every cage had significant Qp values from a chi-squared periodogram analysis, though the amplitude of the Qp was reduced in the constant light group (Fig. S5c). Next, we made average activity profiles for 10 days in LL and DD. We found no evidence of splitting in LL (Fig. S5d). Although there was a trend toward decreased activity, specifically in the active phase in LL (Fig. S5f), there was no statistically significant difference in the average amount of activity between light cycles (Fig. S5def). It is possible that a later time point in constant light would have reduced rhythmicity and decreased activity in the mice since both long term (>1mo) LL and DD effect animal behavior, metabolism, physiology, and brain function^{20,21}. However, we can conclude that after 10-12 days circadian behavior is intact.

We have added the comparison of DD and LL to the results section [Page 8; lines: 196-208; Fig. S5 (below)].

Fig. S5. Comparison of mouse cage activity in constant dark (DD) and constant light (LL). (a) Average chi-squared periodogram for animals in DD (black, n = 10 cages of 2 mice, 1 cage of 1 mouse) and LL (yellow, n = 6 cages of 2 mice). Thick lines indicate mean, shading is standard error. Dotted line represents significant Qp amplitude at different periods. (b) Mean and standard deviation scatter plot of free running period in DD or LL. *p < 0.05. Individual cages represented by single dots. (c) Mean and standard deviation scatter plot of amplitude measures from the χ^2 test. *p < 0.05. Individual cages represented by single dots. (d) Average daily activity profile of animals housed in either DD (n = 10 cages of 2 mice) or LL (n = 6 cages of 2 mice) for 10 days. The thick lines and shaded error bars indicate group average \pm SEM. CT is circadian time where CT 12 is activity onset. (e) Mean and standard deviation scatter plot for total activity counts over a 24h day in DD or LL. Individual dots indicate single cage, ns: not significant. (f) Mean and standard deviation scatter plot for activity counts during the active (top) and rest (bottom) phase in DD or LL. Individual dots indicate single cage, ns: not significant.

8) Figures 3 and 4 were easy to understand and well put-together, clearly demonstrating circadian control of glymphatic influx and lymph node drainage in the context of Figures 1 and 2. However, Figure 4 was very small in the PDF, and may require reorganization to accommodate a printed version.

We have reorganized Fig. 5 (formerly Fig. 4, below) to minimize problems with sizing.

Fig. 5. Lymph node drainage exhibits circadian variation. (a) Experimental outline. Lymph nodes are shown in purple, with the area imaged indicated by a black dashed square. White dashed lines indicate the regions of interest (ROI) drawn around the lymph nodes. (b) Mean pixel intensity of lymph node fluorescence during the day (orange, $n = 12$ mice) and night (blue, $n = 15$ mice). Time 0 on all graphs indicate start of imaging, on average 8 minutes after tracer infusion start. Thick line is the mean, shading indicates SEM. (c) Min/max boxplot of lymph node intensity at minute 50, individual mice represented by colored dots. $*p < 0.05$. (d) Min/max boxplot of the rate of lymph node filling calculated from the first 20 min recording. Individual mice represented by colored dots. $*p < 0.05$. (e) Representative time lapse images from *in vivo* lymph node imaging across 50 min for day (orange) and night (blue). (f-j) Experimental outline, average fluorescence over time, fluorescence in the lymph nodes at 50 minutes during active (green, $n = 6$ mice) and rest (yellow, $n = 8$ mice) phases, slope of lymph node filling, and representative time lapse images for animals housed in constant light for 10 days. ns: not significant. (k) Diagram of experimental flow for lymph node contractility after FITC-dextran cheek injection, including imaging area denoted by black dashed square (left) with representative lymph vessel and trace of contractility from two regions of interest over 2 min (right). (l) Min/max boxplot of the beats per minute of lymph vessels during the day ($n = 5$ mice) and night ($n = 7$ mice) in LD, and active ($n = 7$ animals) and rest ($n = 7$ animals) after 10 days in LL. Individual mice represented by colored dots, ns: not significant. (m) Summary diagram of circadian regulation of CSF distribution.

9) The descriptions of the data presented in Figure 5 are dramatically abbreviated in the text. These data are important to tie the clearly-defined circadian glymphatic/lymphatic accumulation patterns to corresponding regulation of AQP4 localization and DAC gene expression. A more thorough explanation would be appreciated.

Based on reviewer concerns, we have edited the introduction to these experiments, and emphasized that the assays described in Fig. 6 (formerly Fig. 5; CSF brain influx and lymph node drainage) were the same techniques used in Fig. 1, 4, and 5. These sets of experiments are simply showing that loss of AQP4 eliminated day/night differences in CSF movement [Pages 10, 11; Lines: 259-276].

10) The gene measured in Figure 5a is not described anywhere in the methods, legend, or text. Assuming the gene is AQP4, why do AQP4^{-/-} mice have 1 copy number of this gene and AQP4^{+/+} mice have 0?

The values given in 5a were relative copy numbers, not exact copy numbers. A relative value of 1 means 100% of the gene replicates were the knockout sequence. Reviewer 2 has suggested that this figure is redundant since we know these are knockout animals and as such, we have removed it (see below).

Fig. 6. AQP4KO animals lack a rhythm in glymphatic influx. (a) Experimental outline for (b). (b) Mean pixel intensity of slices from AQP4KO animals during the day (orange, $n = 10$ mice) and night (blue, $n = 10$ mice). Min/max boxplots with individual mice shown as dots. ns: not significant. (c) Population-based averaging of slice intensity in AQP4 KO animals in the day and night. (d-h) Experimental outline, average fluorescence over time, fluorescence in the lymph nodes at 50 minutes, rate of lymph vessel filling for the first 20 min, and time lapse imaging across 50 min in AQP4KO mice during the day (orange, $n = 6$ mice) and night (blue, $n = 7$ mice). ns: not significant. Time 0 on all graphs indicate start of imaging, on average 8 minutes after tracer infusion start.

11) In Figure 5, littermate controls are necessary to account for strain differences (AQP4+/+, AQP4+/-, or cre-deficient animals, whichever is most relevant), as this data cannot be compared to Fig 1-4 without evidence of normal circadian control in littermates.

Previous work has demonstrated that WT littermates have glymphatic influx comparable to normal C57BL/6 animals during the day, indicating that the reduction we see is specific to global loss of AQP4 and not a fluke of the strain^{3,22}. Additionally, the KO animals used for this experiment were back-crossed at least 20+ generations to C57BL/6 animals from Charles River prior to use. Therefore, comparing KO animals to normal C57BL/6 animals is genetically appropriate. When comparing influx in C57BL/6 animals (Fig. 1) and KO animals (Fig. 5) between day and night, the only significant reductions in influx were during the day in the KO animals supporting our hypothesis that daytime localization of AQP4 is critical for rhythms in glymphatic function. This analysis has been added to the results section: “We then compared the day/night influx in WT (Fig. 1) and AQP4KO (Fig. 5) animals. As expected, glymphatic influx was only reduced during the day in AQP4 animals (Kruskal-Wallis test: $H(4) = 22.58$, $p < 0.0001$; WT vs KO Day: $p = 0.0002$, WT vs KO Night: $p > 0.999$), when AQP4 localization is highest.” [Page 11; Lines: 264-269].

12) Nowhere in the results or the discussion is it postulated what anatomical pathway the tracer travels from the cisterna magna to the lymph nodes. It is mentioned that this pathway may be independent of the glymphatics due to the lack of delay in tracer accumulation after imaging begins. While experiments to determine the answer may be beyond the scope of this paper, discussion of the potential pathways would contribute to our understanding of CSF outflow dynamics.

We have edited our discussion to include potential basins of CSF after exiting the subarachnoid space of the brain:

“Based on these data, we propose the lack of glymphatic influx is not simply a downstream effect of the lymphatic system drainage of CSF before it reaches the brain. Instead, the awake brain actively suppresses periarterial influx²³, has less perivascular AQP4, and prevents CSF/ISF exchange by reducing the interstitial space volume⁵. This would cause CSF to follow alternative, pre-existing routes out of the subarachnoid space around the brain and spinal cord, such as into the mandibular lymph nodes,

meningeal lymphatics, and/or the deep cervical lymph nodes^{24,25}. In short, CSF distribution can change based on fluid changes and space availability within the brain.” [Pages 13, 14; Lines: 333-340]

Defining the exact CSF efflux route from brain and their dependence on brain state is a matter of active research and beyond the reach of this study, which just emphasizes that CSF can redistribute throughout the body based on the circadian cycle.

Reviewer #2:

Major:

1. Glymphatic influx vs glymphatic system: The authors should precisely and consistently identify what they are measuring. They speak of the "glymphatic system" and "glymphatic function" and of "glymphatic influx", however, experimentally only "glymphatic influx" is assessed, with no evidence presented of an effect of circadian rhythm on other components of the glymphatic hypothesis including flow through the parenchyma and outflow from the brain tissue. Furthermore, we ask that the authors acknowledge the controversy surrounding the glymphatic hypothesis and not present this as irrefutable fact (line 43-44, line 62-64).

The glymphatic system is the network of perivascular spaces that enables CSF to penetrate into the brain, ultimately assisting with clearance of metabolites and solutes from the parenchyma. We apologize for any confusion, and agree that in order to refer to the system as a whole, or the function of the system, we require another metric beyond glymphatic influx (CSF entering the glymphatic system along the perivascular space). As such, we have included exciting new data an endogenous, circadian variation to solute clearance from the brain (Fig. 2). Additionally, we have reviewed the manuscript to ensure we use glymphatic influx, clearance, function and the system as a whole carefully.

We measured clearance of unbound Evans blue (EB) from brain tissue into the peripheral circulation during the active and rest phases under both a standard light/dark cycle and 10 days of constant light. EB is a small fluorescent molecule (960 Da) that has a high binding affinity to albumin. The brain is deficient in albumin, meaning EB is able to penetrate brain tissue until it reaches the blood that has a high percentage of albumin. Once EB binds albumin, it can be imaged in the femoral vein of a live animal over time.

Because this technique has not been regularly used in the context of brain function and fluid movement, we characterized EB injections and solute clearance through a variety of control experiments. We have shown that signal from EB is stable in the blood for at least two hours, with minimal if any renal clearance during that time frame. We have also ensured that EB did not enter the ventricular system upon parenchymal injection, and that the same volume of fluid was injected into the brain by postmortem analysis of all brains used for experimentation. Finally, we have shown that if injection of EB occurs at least 24h after cannula placement, disruption of glymphatic influx and glial scarring is minimal. We have attached a preliminary manuscript for this methodology for the reviewer's information.

Cannulas were implanted into mice 24-36h prior to experimentation into striatum (AP: +0.6 from Bregma, LM: -2 and DV: -3.25mm). The following day, mice in either LD or 10 days LL were anesthetized with KX, fitted with a microdialysis probe, had skin over the femoral vein resected to better visualize EB in the blood stream, and EB was injected into the brain (4% in ACSF, 1 μ L, 0.2 μ L/min). Images of the femoral vein were taken under a microscope (MVX10 Research Macro Zoom Microscope, Olympus) on using Cy5, for EB visualization, and GFP, for venous structure visualization, once every 15 minutes for 90 minutes (14 images/animal). Brains were collected upon experiment end. The brain was drop-fixed overnight in PFA and sliced the next day in the same manner as the CSF influx experiments. Slices were used to ensure parenchymal injections did not hit the ventricle and volume of injection, as measured by % area, was similar between animals. All experiments occurred between CT 2-7 or CT 14-19 depending on if it was the rest or active phase of the animal, respectively. All images were analyzed in ImageJ. Mean pixel intensity in the femoral vein was quantified on the Cy5 channel, with the GFP channel used to ensure placement of the intra-venous region of interest [Page 17; Lines: 416-432].

In LD, EB was higher in the femoral vein as early as 15 min post infusion. By 90 min there was 37% more EB during the day than the night. There was no significant difference in percent area covered by EB in the brain after the experiment, indicating that infusion volumes were similar in each group [Fig. 2 (below); Page 6; Lines: 136-150]. These findings persisted in constant light [Fig. 4h-k (below); Page 9; Lines: 216-220], indicating that clearance from the brain is an endogenous, circadian rhythm that peaks during rest when glymphatic influx of CSF into the brain is highest. Thus, we have demonstrated that glymphatic influx, molecular regulation of AQP4 polarization, and glymphatic function in the form of solute clearance from the parenchyma is under circadian control, indicating the glymphatic system has a circadian rhythm. We have integrated this experiment into the results.

We believe that whatever "controversy" surrounds the glymphatic system is not relevant to the conclusions of this paper. We begin by clearly introducing our definition of the glymphatic system. Then, we show that

glymphatic influx and clearance of EB from the brain tissue is regulated in a circadian manner, demonstrate a daily oscillation in the DAC and AQP4 polarization. And, finally, we present data where CSF is cleared to the mandibular lymph nodes in a rhythm opposite of glymphatic influx indicating CSF can be redistributed to different compartments based on clock timing. Our study is a primary report of new data, not a review. We are using standard terminology and what other investigators choose to call such basic biological phenomena does not change the data or conclusions.

Fig. 2. Interstitial fluid clearance is higher during the day. (a) Schematic of experiment. (b) Representative Cy5 images of the femoral vein from animals 90 min after evans blue (EB) injection during the day (orange) or night (blue). (c) Mean pixel intensity in arbitrary units (A.U.) over 90 min post EB injection. Thick lines indicate group means with SEM outlined, individual point indicate time image was taken. The inset is the magnitude of fluorescence at 90 min with min/max boxplots and individual animals represented as scatter plots. Time 0 is tracer infusion start. Day: $n = 9$ mice, Night: $n = 6$ mice. $*p < 0.05$. (d) Schematic of processing brains to check injection localization and specificity. (e) Representative striatal EB injection sites. Blue circle indicates probe end. (f) Min/max boxplot with individual animals represented as scatter plots of percent area of EB quantified in a subset of brains during the day ($n = 6$ mice) and night ($n = 4$ mice), ns: not significant.

Fig. 4. Differences in the glymphatic system persist in constant light. (a) Experimental outline with representative double-plotted actogram of two animals running in constant light (LL). Black tick marks indicate beam-breaks. (b) Average activity profile of animals housed in LL for 10 days. The black line and gray error bars indicate group average \pm SEM, the thin gray lines are individual cage activity profiles. Yellow is rest phase, green is active phase. CT is circadian time, CT 12 is activity onset. $n = 7$ cages, 14 mice. (c) Average mean pixel intensity (arbitrary units, A.U.) from ex vivo tissue fluorescence. Min/max boxplot, individual mice indicated by dots. $n = 13-14$ mice per group. $*p < 0.05$. (d) Representative slice of AQP4 staining. White boxes indicate areas of dorsal cortex (DC), lateral cortex (LC), and ventral cortex (VC) where AQP4 localization was quantified. (e) Representative 40x confocal images stained for AQP4 (magenta). (f) Average intensity of AQP4 centered on vasculature in the cortex during the rest (yellow, $n = 5$ mice, 45 vessels) and active (green, $n = 7$ mice, 63 vessels) phase, SEM indicated by shading around the thick line. (g) Average polarization index for staining in ventral, dorsal and lateral cortex. Min/max boxplot with individual mice shown as colored dots. $*p < 0.05$. Polarization index = peak fluorescence - 10 μ m baseline. All values were normalized to the highest signal for better visualization. (h) Schematic of clearance experiment. (i) Representative Cy5 images of the femoral vein 90 min after evans blue (EB) injection during the rest (yellow) or active (green) phase. (j) Mean pixel

intensity over 90 min post EB injection. Thick lines indicate group means, outlines indicate SEM, individual points indicate time image was taken. The inset is the magnitude of fluorescence at 90 min with min/max boxplots, individual animals represented as single dots. Time 0 is tracer infusion start. $n = 5$ mice per group. $*p < 0.05$. (k) Percent area of EB. Min/max boxplot with individual animals represented as dots during the rest and active phases ($n = 5$ mice per group), ns: not significant.

2. Influx into the brain vs flow along perivascular spaces at the brain surface: In different parts of the manuscript, "glymphatic influx" is assessed in different ways: either in vivo by visualizing tracer in the perivascular spaces around the middle cerebral artery at the surface of the brain, or ex vivo in post-mortem brain sections. The authors should make clear for which parts of the study each method was employed and why these different approaches were used. Methods in vivo and ex vivo are not interchangeable, and the authors should avoid general phrases such as "influx into the brain" (line 94, 116, 191) when the data are showing influx of tracer to the perivascular spaces of the middle cerebral artery at the surface of the brain and not to the brain parenchyma (line 94) or when both types of influx are referred to (line 116, 191). There are reports (Ma et al., 2019) that some CSF influx into the brain appears to occur only after death rather than in vivo and this limitation of using post-mortem brain sections to assess CSF influx into the brain should be mentioned. The authors' results also appear to reflect differences between the magnitude of influx to the PVS on the brain surface and influx into the brain itself, as the difference between day and night was far greater at the brain surface in vivo (53%, line 93) than in the post-mortem brain sections (22%, line 105). Wouldn't one expect that if a one-directional flow from the perivascular spaces on the surface proceeded to brain tissue as proposed by the glymphatic concept that the differences between groups would be more comparable? Could this discrepancy be related to the post-mortem artifact?

Multiple studies have validated both transcranial macroscopic imaging and ex vivo slice analysis as two comparable measures of CSF movement across murine cortex during the day (Fig. 1, S1abc [below]), when altering hypertonicity (See Fig. 1-3, S2, S3, and S10 in Plog et. al.², <https://insight.jci.org/articles/view/120922>), and between WT and AQP4KO animals (See Fig. 5 in Mestre et. al.³, <https://elifesciences.org/articles/40070>). Additionally, the methodology for transcranial imaging along with a discussion of how different tracer analyses can provide different measures of CSF distribution is published and widely available¹. Finally, we have demonstrated that ex vivo processing of tissue provides information not just of tracer penetration into cortex, but also to sub-cortical areas under anesthesia (See Fig. S3 in Hablitz et. al.²⁶, <https://advances.sciencemag.org/content/5/2/eaav5447>). We would also like to point out that there has yet to be any evidence of ex vivo tissue analysis not corroborating findings from transcranial imaging.

We concur with the data reported in Ma et al. 2019²⁷. To account for this observation, we have used rapid decapitation followed by drop fixation to process all of our brain tissue samples since 2017. This method is useful because: 1) decapitation occurs when the animal is still breathing, 2) it eliminates potential confounds of high-dose anesthesia, 3) it removes the pool of CSF at the cisterna magna (where the post-mortem artifact originates), and 4) the brain is harvested within two minutes and drop fixed in paraformaldehyde, eliminating the death effect and the perfusion artifact²⁸.

In the submitted manuscript, we have comparative analyses of transcranial macroscopic imaging and ex vivo slice analysis: transcranial macroscopic imaging established a strong day/night difference in CSF influx to the middle cerebral artery and this observation was confirmed using ex vivo tissue analysis across three separate 24h time courses and anesthetics (Fig. 1). We find that transcranial imaging would be redundant and an unethical use of more mice to simply replicate ex vivo tissue analysis already completed for active/rest phase differences in constant light (Fig. 4) and a lack of influx in AQP4 KO animals (Fig. 6).

We agree that circadian variation of influx is more pronounced along the larger perivascular spaces of the MCA detected by macroscopic imaging than in the average slice intensity across all brain tissue. The purpose of this study was not to calculate what percentage of tracer detected by macroscopic imaging (which has major caveats to data interpretation as the reviewer describes in point 3) does or does not get into the brain, but to simply show that entrance of CSF tracer to perivascular networks is regulated by an endogenous biological clock mechanism.

3. Imaging through the skull and analysis: While no description of the analysis of tracer spread at the brain surface in vivo is given (see comments under Methods below), the ROIs indicated on Supplemental video 1 appear to include signal which is emanating from CSF spaces outside

the surfaces of the cortex such as the quadrigeminal and olfactofrontal cisterns. If the authors wish to quantify CSF influx to the perivascular spaces at the brain surface, rather than spread of tracer from the cisterna magna to these other CSF compartments, the analysis ROI should be limited to the middle cerebral artery regions. In Supplemental video 2, the 4th mouse in the "Day" group appears to show much more tracer spread than any of the others. Is this a statistically significant outlier? If so, what do the analysis and averaged videos look like if this mouse is excluded?

We have confirmed that mouse 4 is not more than 2 standard deviations from the mean, so we cannot consider it a statistical outlier. However, we have also re-analyzed the data without it and get the same day/night difference. Furthermore, we have included new analysis of the MCA portion as the reviewer suggested [Fig. S1a; Page: 4; Lines: 98-100]. As expected, influx was significantly higher during the day than the night.

Fig. S1. Glymphatic influx has a diurnal rhythm in anesthetized mice. (a) Representative images for in vivo recordings of CSF tracer influx after injection during the day (orange) and night (blue) under ketamine/xylazine (KX) anesthesia (left). Dotted white line indicates region of quantification. (right) Mean pixel intensity in arbitrary units (A.U.) over 30 min post CM injection. Thick lines indicate group means with SEM outlined. The inset is the fluorescence at 30 min with min/max boxplots and individual animals represented as dots. Time 0 is tracer infusion start. $n = 6$ mice per group. $*p < 0.05$. (b) Representative slices for CSF tracer influx during the day or night under KX anesthesia. (c) Time course of mean pixel intensity from ex vivo coronal sections under KX (violet, $n = 10-12$ mice per time point), Pentobarbital (red $n = 7-11$ mice per time point), and Avertin (gray $n = 7-11$ mice per time point). Each colored point is an animal. Dotted line indicates CircWave (CW) fitted curve. Black point with bars are estimated Center of Gravity (CoG) from the CW fit with standard deviation. Light cycle is indicated by orange (day) and blue (night) coloring in the background. Zeitgeber Time (ZT) in hours, where ZT0 is lights on and ZT12 is lights off. (d) CW curves and CoG (left) for KX, pentobarbital (P) and avertin (A). Mesor-aligned CW and Cosinor curves for the three anesthetics (right), dotted lines indicate cosinor fit, solid lines indicate CW fit. (e) 95% confidence intervals from the cosinor analysis, CoG estimates from CW, and peaks from CW curve for phase and mesor for each anesthetic. (f) Schematic of experiment and min/max boxplot of average mean pixel intensity for awake mice receiving a CM injection during the day (white, $n = 7$ mice) and night (gray, $n = 9$ mice), individual mice represented by single dots, ns: not significant. (g) Min/max boxplot of mean intensity from anterior to posterior slices. Individual animals represented by dots. Slice diagrams shown above. (h) Min/max boxplot of mean intensity for hippocampus (HC), thalamus (TH), hypothalamus (HT), dorsal cortex (DC), lateral cortex (LC), and ventral cortex (VC). Individual animals shown by dots. Schematic of subregions shown in inset.

4. Circadian changes yes - but do these follow a sinusoid curve? We read with interest that CSF tracer influx into brain surface perivascular spaces in vivo and on ex vivo brain sections shows a diurnal variation, and appreciate the results presented showing differences between time points. However, when examining the graphs in Fig. 1d, both individual data points and means at each time point, the idea that the circadian variation follows a cosinor curve (although the authors say this is significant) is not entirely convincing. Could the authors further elucidate why this type of curve fitting was chosen and acknowledge that the curves indicated by dotted lines may not be a perfect fit (especially for pentobarbital and avertin)?

Cosinor analysis is a standard technique in the field of chronobiology for determining if discrete data points (i.e. each data point is an independent animal) in a time series exhibit a ~24h rhythm. It fits a simple sinusoid wave through the dataset, and compares this with a horizontal line through the data mean via an F test. The purpose of this test was to show that there was a day/night rhythm, and to estimate parameters (amplitude, mesor, and phase) of each dataset that could be compared between anesthetics. There are other analyses for circadian data that do not assume a sinusoidal wave, but many cannot generate parameter estimates and have increased risk of false-positives of rhythmicity if the data is not structured properly. We already state that the magnitude of change in average slice intensity during the day and night is 22% [Page 5; Lines 109-111]. We have included F test statistics and R^2 values, which demonstrate that a sine wave accounts for around 15-20% of the variability in the data [Page 5; Lines: 109, 118-120]. However, the reviewer does have a point that this may not be the best line fit for each dataset.

To address this concern, we have reanalyzed this data with CircWave (to download CircWave: <https://www.euclock.org/results/item/circ-wave.html>; for applications see: ²⁹⁻³¹). This software is similar to a cosinor analysis where the data is fit to a sinusoid wave then compared to a linear mean. But, it can also fit the data multiple harmonics of the sinusoidal wave, and thus can determine if there is a better waveform that can explain more of the variance. This re-analysis confirmed our previous findings, where the simplest fit of the data was a single-harmonic sinusoid curve that accounted for approximately 20% of the overall variance. We also compared the Center of Gravity (CoG), a measure of phase and mesor of the datasets, and found similar results to our 95% confidence intervals above, where anesthetic changed the mesor but each phase estimate was approximately midday. Pentobarbital had a more delayed CoG compared to the other anesthetics, yet the deviations overlapped and the predicted peak of the curve matched that of the cosinor analysis [Fig. S1cde (above); Page 5; Lines: 122-128].

We are confident that this new analysis, combined with our cosinor analysis, demonstrates that there is a rhythm to glymphatic influx that peaks around mid-day. Time courses are very difficult to generate requiring huge time commitments, live animal surgeries, and very large cohorts. Analyzing them is even more difficult because each data point is one individual mouse, and biology is highly variable. Therefore, it is extremely impressive that each of the three time courses had similar phase, amplitude, and described the same amount of variability in the data, and that we confirmed these findings using two separate analyses.

5. Circadian rhythms in CSF production or lymphatic function in mouse? The authors cite studies which assessed diurnal variations in CSF production in humans (line 233). It is important to note that this early phase contrast MRI work has limitations and this particular finding of increased CSF production at night in humans was unable to be reproduced in a more modern study (Takahashi et al, 2011). Does CSF production in mouse change with circadian rhythm or activity state? This is key to the interpretation of the experimental results in this manuscript, as changes in CSF production will affect its flow. Relevant studies should be cited and this should either be addressed experimentally by the authors or clearly mentioned as a limitation. Furthermore, it would be of interest if AQP4 KO affects CSF production. Again, this should be addressed or mentioned as a limitation.

We agree with the reviewer that the studies estimating CSF production in humans across the day have been inconsistent and mildly inconclusive. Thus, we have been very careful in the discussion about saying there is definitively whether there is an endogenous rhythm. Mouse CSF production has in the past been quantified using the dilution methods which experimentally have been shown to have major caveats^{32,33}. Due to temporal, technical, and precision concerns this technique would not be sufficient to answer the question of rhythmic or arousal-state dependent changes in CSF production. Our lab is actively working to generate better techniques to measure CSF production, but until such a time as it is possible we are unable to answer this important biological question. We have included an additional sentence in the discussion to emphasize that we do not know how CSF production may alter glymphatic influx: "When improved methodology for CSF production in rodent becomes available^{33,32}, it should be investigated whether CSF production regulates glymphatic CSF influx along the perivascular spaces of the brain." [Page 13, Lines: 319-321]

Related to this: Line 238: "brain states regulate glymphatic influx independently of the endogenous rhythm in CSF production. For example, waking an animal up during daytime will suppress glymphatic influx, while anesthesia immediately increases influx in awake mice". The authors should rephrase this to clarify the logic behind this sentence, which does not address

what happens to CSF production in those conditions, independently of its endogenous rhythm. Again, whether there is a circadian variation in CSF production in mice should be clarified.

CSF production is an intriguing, and ultimately important part of this physiology. However, we cannot measure it in mice at this time.

This paper defines a rhythm to the distribution of CSF in the brain and the mandibular lymph nodes. As far as the authors are aware this is the first time that the idea of differential CSF distribution between two physiological systems (the glymphatic system in the brain and the lymphatic system in the periphery) based explicitly on time-of-day has been published. Between the lack of technical possibility and because we are measuring CSF distribution after production, CSF production is beyond the scope of this paper.

We have included the idea of CSF production changing with arousal states into the discussion: "For example, waking an animal up during daytime will suppress glymphatic influx, while anesthesia immediately increases influx in awake mice, though it should be noted that how arousal state and anesthesia alter rates of CSF production remain unknown." [Page 13; Lines 316-319]

Lymphatic function in general is highly linked to activity state. Are there circadian variations in lymphatic function? These factors are also very relevant and important for the interpretation of the results presented in the manuscript.

We are curious how understanding all rhythms underlying CSF production, all routes of CSF clearance from the skull compartment, and functionality of the lymphatic system will change the fundamental and novel conclusions of this paper: that distribution of CSF in the glymphatic system, clearance of solute from the brain, and drainage of CSF to the peripheral mandibular lymph nodes are dependent upon circadian timing.

In response to the reviewer's question about lymphatic function: some lymphatic vessels use contractions of the vessel walls to draw fluid into the lymph nodes³⁴. We have included new data testing whether this might be an underlying mechanism for the increased lymphatic drainage we see during the active phase in the mandibular lymph nodes. KX anesthetized mice were injected subcutaneously in the cheek with FITC-conjugated 3 kD dextran (0.25% in normal saline, 20 μ L per cheek), then placed their backs with the mandibular lymph nodes exposed via skin resection. Imaging acquisition began following lymph vessel filling (less than 2 minutes post cheek injection) at a rate of 2.5 hz for 5 minutes (750 images/animal). Images were analyzed in ImageJ. A horizontal line segment centered on the lymph vessel was drawn and fluorescence over time was calculated using this region of interest. Beats per minute (the number of oscillations of fluorescence) was calculated from each lymph vessel, and averaged for each animal. [For methods see: Page 20; Lines: 508-517]

*In both a standard LD cycle (day: n = 7 mice; night n = 5 mice) and in 10 days LL (rest: n = 7 mice; active: n = 7 mice) we found no significant differences in beats per minute between the rest and active phases (2-way ANOVA, main effect of phase: $F(1, 22) = 0.603$, $p = 0.4457$; interaction light*phase: $F(1, 22) = 0.173$, $p = 0.6815$). Although skin resection and anesthesia may alter BPM, both groups received the same anesthesia and surgery and still show active/rest differences in lymph node filling after CM injection of CSF tracer. Thus, we conclude that intrinsic contractility of the mandibular lymph node vessels are not responsible for circadian drainage of CSF to the lymph nodes. [Fig. 5kl (below); Page 10; Lines: 247-258]*

This paper was focused on circadian regulation of glymphatic function and day/night differences in CSF distribution. Circadian regulation of the immune system is a new and growing area of research.³⁵ There have been reported changes in lymphocyte recruitment to blood depending on time-of-day³⁶, and pubmed searches for "lymphatic circadian" and "lymph node circadian" bring up 81 and 60 articles, respectively. Future studies investigating underlying mechanisms of rhythmic functions of the lymphatic system would be better suited for experts in the field. We have added reference 35 and 36 to our discussion [Page 14; Line: 352] where we already discussed how understanding rhythms in the cardiovascular system and immune system are necessary to understanding CSF distribution.

Fig. 5. Lymph node drainage exhibits circadian variation. (a) Experimental outline. Lymph nodes are shown in purple, with the area imaged indicated by a black dashed square. White dashed lines indicate the regions of interest (ROI) drawn around the lymph nodes. (b) Mean pixel intensity of lymph node fluorescence during the day (orange, $n = 12$ mice) and night (blue, $n = 15$ mice). Time 0 on all graphs indicate start of imaging, on average 8 minutes after tracer infusion start. Thick line is the mean, shading indicates SEM. (c) Min/max boxplot of lymph node intensity at minute 50, individual mice represented by colored dots. $*p < 0.05$. (d) Min/max boxplot of the rate of lymph node filling calculated from the first 20 min recording. Individual mice represented by colored dots. $*p < 0.05$. (e) Representative time lapse images from *in vivo* lymph node imaging across 50 min for day (orange) and night (blue). (f-j) Experimental outline, average fluorescence over time, fluorescence in the lymph nodes at 50 minutes during active (green, $n = 6$ mice) and rest (yellow, $n = 8$ mice) phases, slope of lymph node filling, and representative time lapse images for animals housed in constant light for 10 days. ns: not significant. (k) Diagram of experimental flow for lymph node contractility after FITC-dextran cheek injection, including imaging area denoted by black dashed square (left) with representative lymph vessel and trace of contractility from two regions of interest over 2 min (right). (l) Min/max boxplot of the beats per minute of lymph vessels during the day ($n = 5$ mice) and night ($n = 7$ mice) in LD, and active ($n = 7$ animals) and rest ($n = 7$ animals) after 10 days in LL. Individual mice represented by colored dots, ns: not significant. (m) Summary diagram of circadian regulation of CSF distribution.

6. Relationship between CSF influx and CSF efflux Line 245. The data in this manuscript do support the conclusion by Proulx and colleagues that CSF influx to the perivascular spaces and CSF clearance to lymphatics are inversely correlated. We disagree that the data in this manuscript support the conclusion that glymphatic influx is reduced and lymphatic drainage peaks "during wakefulness". To our knowledge, no awake mice were used in this study. Please rephrase to clarify that the results of the manuscript indicate reduced CSF influx and increased lymphatic drainage during the mouse's active phase compared to its rest phase in anesthetized mice.

Line 245. We disagree that the conclusion that data in this manuscript "do not fit into the model". Circadian regulation and AQP4 localisation were not a component of the model proposed by Proulx et al. These additional components are not in rejection of the model but rather add further facets to what is clearly a complex process. Overall, with the exception of the AQP4 KO data (although see comment 7 below) the data is consistent with the model proposed by Proulx and colleagues in which CSF clearance to lymphatics and potential influx to perivascular spaces are inversely correlated.

Line 250: "Instead, lymphatic drainage of CSF is a shunt that opens when the awake brain is actively suppressing glymphatic influx...". Please rephrase to clarify that this is a hypothesis/proposed mechanism. The authors of this manuscript, and others, have shown that there is indeed CSF drainage to lymphatics during sleep/anesthesia/resting phase (e.g. Fig. 4).

Thus, it appears that the proposed "shunt" is not "closed" in these conditions. How do the authors reconcile this with their proposed mechanism?

Upon rereading this paragraph, we can understand how it may be misleading. We have rewritten this paragraph:

"Proulx and colleagues recently proposed a model whereby rapid CSF turnover through lymphatics precludes significant bulk flow into the brain²⁷. This was based on two main conclusions from the manuscript: no tracer enters perivascular spaces of penetrating arterioles prior to death based on their macroscopic imaging, and lymphatic outflow is increased in wakefulness while glymphatic influx is decreased along the MCA. Our lab has already demonstrated CSF tracer alongside penetrating arterioles in live animals using two-photon microscopy^{5,22,23,37}, which has considerably higher spatial resolution than macroscopic imaging^{23,37}. Additionally, we have used rapid decapitation followed by drop fixation to process all of our brain samples since 2017 to eliminate the death effect²⁷ and perfusion artifact²³ of CSF tracer localization in the brain. Here, we use fully anesthetized mice and observe changes in glymphatic and lymphatic CSF distribution dependent upon circadian timing, with underlying molecular rhythms in the DAC and AQP4 polarization. Based on these data, we propose that the lack of glymphatic influx is not simply a downstream effect of the lymphatic system drainage of CSF before it reaches the brain. Instead, the awake brain actively suppresses periarterial influx²³, has less perivascular AQP4, and prevents CSF/ISF exchange by reducing the interstitial space volume⁵. This would cause CSF to follow alternative, pre-existing routes out of the subarachnoid space around the brain and spinal cord, such as into the mandibular lymph nodes, meningeal lymphatics, and/or the deep cervical lymph nodes^{24,25}. In short, CSF distribution will change based on fluid changes and space availability within the brain." [Pages 13, 14; Lines: 322-340]

We hope this has clarified our revised model of CSF dynamics in the brain and lymphatic system.

7. Accounting for overall reduced signals in the AQP4 KO mice. In Figure 5, the combined intensity of fluorescence of the lymph nodes and the ex vivo brain sections of the Day/Night AQP4 KO groups appears to be lower overall than that in the Day/Night and Active/Rest WT groups. Can the authors account for where the tracer is in the AQP4 KO mice, if it is not being carried by a substantial CSF influx into the brain or cleared to the lymphatics? One potential explanation could be that more tracer is retained in the CSF space (indicating perhaps a slower turnover in AQP4 mice). Did the authors assess this in their study?

The reviewer is correct that AQP4 KO animals have significantly lower glymphatic influx during the day. We compared the day/night influx in WT (Fig. 1) and AQP4KO (Fig. 5) animals. As expected, glymphatic influx was only reduced during the day in AQP4 animals (Kruskal-Wallis test: $H(4) = 22.58$, $p < 0.0001$; WT vs KO Day: $p = 0.0002$, WT vs KO Night: $p > 0.999$), when AQP4 localization is highest [Page 11; Lines: 264-269]. However, the lymph node imaging in KO animals was done several months after the WT cohort. Because these are fluorescent microscopy studies, these two cohorts cannot be compared and we can only see the presence or absence of day/night variation in drainage. Further detailed analysis of how KO's regulate peripheral CSF distribution would be necessary, which is beyond the scope of this paper.

Minor:

Color coding throughout: The authors have used different colors to label the groups of mice used throughout, which is a great idea and very much aids in legibility of the results. However, the fact that the "Day" and "Rest" groups are both color-coded in orange leads to some confusion. From our understanding, this is not the same experimental group, and we advise that different colors be used.

We have changed the colors on rest animals in constant light to yellow instead of orange. Initially, colors were the same to indicate the same circadian time in each experiment though these animals were under two different light cycles.

Graphs throughout: The authors have displayed data in graphs in different formats throughout, e.g. individual data points with mean and SEM, min/max dotplots with or without shading of bars, mean with SEM but no individual data points. We ask that the authors either use the same type of column graph throughout, or otherwise give convincing explanations why different types of data display are needed. Furthermore, unless there is a plausible reason otherwise, graph axes should start at 0 or include an axis break.

We found different types of data display are necessary for different experimental paradigms and analyses. We are surprised at the request to represent all data as column graphs, especially when data may non-parametric (with different distributions or variances), data may be either continuous (as in in vivo experiments) or discrete (time course data, molecular data, CM injection slice analysis, etc.), and at a time when the science community at large is pushing towards more transparency and variety in data representation.

All experiments (excluding Fig. 1d-f and 3h see rationale below) have group data shown as scatter plots with min/max boxplots. All min/max boxplots are color coded based on which experiment they are for: 12:12LD day/night is orange/blue (Fig. 1, 2, 3 and 5), constant light rest/active is yellow/green (Fig. 4, 5), AQP4KO day/night has black and white boxplots with orange/blue scatter to indicate time of day (Fig. 6).

For continuous data (e.g. transcranial imaging: Fig. 1bc, mandibular lymph node imaging: Fig. 5, 6e), we used mean \pm SEM over time. Shading for these graphs follows the same color coding described above, with the exception of lymph node imaging for AQP4 KO animals (Fig. 6), where day/night is still orange/blue, but the SEM shading is less opaque to avoid confusion with data shown in WT animals under a normal light cycle. This was done to understand overall trends in a continuous data plot. We showed group data as min/max boxplots overlaid with scatter plots for the final time point of each recording (30 min for transcranial, 50 min for lymph node), color coded as described above, to visualize within-group variability. This combination allowed for visualization of a variety of descriptive statistics including: mean, SEM, percentiles, median, and individual animal variability. This is necessary because, as the reviewer points out in 5c (formerly 4c), the median and mean are not always the same value.

For the time course data (Fig. 1d-f, S1c-e): the large animal number, three anesthetic groups, and the circular statistics necessary to analyze circadian data caused us to graph the data differently. We were concerned that column graphs or boxplots would make data visualization difficult due to the multiple time points. We chose to highlight within time point variability with scatter plots, across time point group trends with mean \pm SEM, and to show the fitted curves from the cosinor analysis (Fig. 1d). We chose to change the graph y-axes on these plots because although anesthesia alters total influx, as we've reported previously (see Hablitz et. al. 2019²⁶) and report in the same figure (Fig. 1f), it does not alter overall amplitude of the rhythm (Fig. 1df). This means that starting at 0 is not necessary and decreases visibility of the data. For comparison of cosinor curves (Fig. 1ef), we were comparing three curves across multiple parameters using 95% confidence intervals, standard for this analysis type. Column graphs would not be suitable for this analysis.

RT-qPCR data (Fig. 3h) has been plotted as is standard for this type of data. Additionally, this dataset was normally distributed and had equal variance between groups, thus displaying the mean \pm SEM was adequate to represent the data.

We believe the above rationale justifies our visualization of each dataset.

Figure 1: Legend e. The curves shown appear to be sine rather than cosine curves.

The curves shown were those predicted using the cosinor analysis, fitting to a sinusoidal wave that is both, mathematically, a sine and cosine. These curves peak around ZT6 and trough around ZT18. Sine and cosine curves differ only in phase relationship, and have the same waveforms. Describing them as either sine or cosine will not change the meaning of the graph or the analyses, therefore this comment is unnecessary.

Figure 2: By eye, the AQP4 images from the "Day" group look less polarized, as there is evidently more AQP4 staining which is not localized to the vessel. A clarification of the analysis method would help to understand how the polarization index data were obtained (see comments under Methods below).

Due to the "by eye" differences in AQP4 staining between day and night, vascular-localized AQP4 signal was calculated by subtracting the baseline intensity (average intensity from 10 μ m before the end-feet) from the peak intensity within the vascular end-feet. This method would take into account intensity changes between slices or time points. We then took all values in the dataset and divided by the highest value to give us a range of 0 to 1 in order to make the data easier to visualize. It is also the reason we did western blotting for AQP4, to determine if the background increase was increased AQP4 protein. We have added further clarification on the methodology to the methods section [Page 18, Lines: 446-460] and in the figure legends of Fig. 3,4.

Perhaps, the authors have found a difference in AQP4 localisation based on their polarization index, but not in the total levels of AQP4 protein because they have studied localization in the

cortex only, while Western blotting was carried out with whole brain lysates excluding only the cerebellum? This limitation should be mentioned.

The reviewer poses a valid concern. Although cortex is the most abundant brain region in whole brain lysate, we have re-imaged brain slices from the day (n = 4 mice) and night (n = 4 mice) in two additional brain areas, the hypothalamus and CA1 of the hippocampus. Though not statistically significant, we found clear trends for increased AQP4 polarization in both hypothalamus (Day: 0.82 ± 0.08 a.u., Night: 0.57 ± 0.07 a.u.; t-test: $t(6) = 2.342$, $p = 0.0577$), and CA1 (Day: 0.97 ± 0.01 a.u., Night: 0.88 ± 0.04 a.u.; Welch's t-test: $t(3.109) = 2.139$, $p = 0.1188$). These data support the conclusion that overall protein levels of AQP4 do not exhibit a strong day/night differences, but underlying rhythms in the DAC support circadian variation in the localization of AQP4 to vascular end feet of astrocytes. They have been added to the results section [Page 7; Lines: 157-159], as well as Fig. S3.

We believe the graph in 2b should be labeled "Distance" or "Distance From Vessel" instead of "Vessel Length".

Figure 3b and 4f (formerly 2b and 3e) have been corrected as suggested.

Figure 3: 3b and c. Did the authors also carry out in vivo imaging through the skull in the Rest/Active groups? If not, why was only ex vivo analysis of tracer influx carried out?

We have, as discussed above, critically compared macroscopic live imaging of CSF tracer transport vs quantification CSF tracer distribution in slices ex vivo in 3 independent experimental paradigms. We hope the reviewer agrees that it is ethically irresponsible to kill a large number of mice when data already exists. We chose to collect data using the ex vivo average slice intensity analysis because it has higher spatial resolution, and is directly correlated to how much tracer can get into the brain tissue², including sub-cortical regions²⁶.

3d. Why is DAPI staining shown here? We do not feel this is necessary and that it obscures the visibility of the AQP4 staining. The presentation with lectin staining, as in Fig. 2a, was much easier to interpret.

We have removed the DAPI staining, and increased the magnification of AQP4 to better visualize localization. The mice we used for this experiment were not wheat germ agglutinin perfused. The vessels are still visible, thus it is not necessary to expand this cohort of animals for a more attractive representative figure.

Figure 4: 4b, 4g. The graphs of lymph node uptake of tracer (4b, 4g; also 5f) do not seem to start at Time=0 as there is a gap between the x-axis and the start of the plotted line. Is this a feature of the graph style or are there no data points near T=0? If the latter is the case, how did the authors calculate the slope from T=0 to T=20? The inset graph in 4b of the slope from 0 to 20 minutes is not essential, and in its current size not legible.

It was a default setting on the graph, we have replotted all drainage experiments to start at zero. All slopes were from $t = 0$ to $t = 20$. The inset has been removed.

Please clarify if time "0 minutes" refers to the start of imaging or a specific time after tracer infusion. Displaying the data relative to the time after tracer infusion would more readily allow the reader to appreciate the real time course of tracer clearance to lymph nodes.

Time 0 for all in vivo imaging represents the start of imaging, both for through the skull and the lymph nodes. For through the skull: this imaging occurs at the start of tracer infusion. For lymph node imaging: this occurs on average 8 minutes after tracer infusion start. These are solely based on logistics of the live animal surgery, animal protocols, and placement under the microscope. This timing is clarified in the figure legends for Fig. 5 and 6, and was already discussed in the results section.

4c, fh. Could the authors clarify at which time point the mean intensity values of the lymph nodes were calculated? The values appear to differ from those shown at the end of the in vivo imaging (50 min). In Fig. 1c, the values for the imaging through the skull appear to match between the end of the dynamic imaging and the inset graph with the statistical analysis. This comment would also apply to Fig. 5f vs. 5g.

The time points for all in vivo imaging insets (Fig. 1, Fig. 5, and Fig. 6) were at the end of each imaging session, as indicated by the figure legends (skull: 30 min, lymph nodes: 50 min). We believe the confusion is because boxplots show the median MPI within the group, while the in vivo imaging shows the mean. We

intentionally chose these plots (mean \pm SEM over time, scatter and min/max boxplots for the final time point) to allow the reader to visualize all descriptive statistics of the datasets.

4k. The labels and color coding of the schematic should be changed to also include the Day/Night groups as the conclusion is the same.

We have changed the colors of Fig. 5m (above) to halfway between LD and LL colors, and have clearly indicated subjective day and subjective night (i.e. rest phase and active phase) so readers will not mistake our model for arousal states.

Please clarify why only the submandibular lymph nodes were chosen for analysis (accessibility for in vivo imaging??), and why deep cervical lymph nodes, which have recently been reported to be the major lymphatic site of CNS drainage (Raper et al., 2016; Ahn et al., 2019), were not assessed.

As the reviewer suggests, the mandibular lymph nodes were chosen for accessibility for in vivo imaging. Deep cervical lymph nodes were not collected because the purpose of these studies were to simply show circadian CSF distribution in more than one CSF compartment, not to fully characterize all CSF outflow routes.

Figure 5: 5a: We believe this panel is not necessary and could be removed completely or moved to a supplemental figure.

This panel has been removed.

5c and d: Again, did the authors also carry out in vivo imaging through the skull in the AQP4 and AQP4 KO groups? If not, why was only ex vivo analysis of tracer influx carried out?

Daytime in vivo transcranial imaging has already been published in this strain of AQP4 KO animals, and is therefore redundant when day/night ex vivo comparisons are shown in Fig. 6 (See Mestre et al.³: Fig. 5AB for transcranial imaging and Fig. 5C-I for ex vivo analysis, <https://elifesciences.org/articles/40070>). Again, please refer to the discussion above. We hope that the reviewer agrees that it is ethically irresponsible to kill a large number of mice when data already exists. Ex vivo analysis of tracer distribution in slices provides information on influx into brain with high spatial resolution and is not associated with the caveats of macroscopic imaging.

5f: Please add quantification of the slope from 0 to 20 min as done in all other groups (Fig 4).

We have included the slope for AQP4KO day vs. night. As expected, there was no significant difference between day vs. night (t-test: $t(11) = 0.9202$, $p = 0.3772$; Fig. 6g). [Page 11; Lines: 273-274]

Overall, we have found the information on methods and n numbers for individual experiments laborious to gather as it is scattered between Methods section, figure legends and text.

We state animal number for every experiment in the results section. We have also added animal number to every figure legend. We hope this further clarifies any issues finding sample size information.

Please add a description of the method used for imaging through the skull and for the quantification of tracer spread at the brain surface in vivo.

We have added a paragraph to the methods section about transcranial imaging: "Animals were anesthetized with KX at ZT6 or ZT18 and were prepared for transcranial imaging as described previously^{1,2}. In brief: animals were placed in a stereotaxic frame and the skin was removed from the top of the head to expose the skull. The animal was then placed under the microscope (MVX10 Research Macro Zoom Microscope, Olympus). A CM injection was then performed as described above, with imaging beginning at pump start. Image acquisition was once per minute on the Cy5 channel, 30 minutes total (31 images per animal)." [Page 16; Lines 409-415]

We apologize for not including the methodology in the initial submission.

Line 314: Please add more detail of the method used for AQP4 quantification. How many slices were stained per mouse? Were equivalent slices taken? How many images were taken? Were these taken in a standardized location within the cortex, which has been reported to have heterogenous vascular density? Which blood vessels were chosen for analysis? How many blood vessels were analysed per mouse/section? Why were images taken at different magnifications and different staining carried out for Fig 2a and Fig 3d? Was the analysis also different, as the baseline level of staining intensity appears to be lower (1 in 3e compared to 3 in 2b)? Please clarify how the polarization index was calculated, as based on the description given,

for e.g Fig 2b we would expect a value of 20 (peak) divided by 4 (baseline) giving an index of 5. However, the values in 2c are much lower. Please also clarify how the start of the vessel was identified in order to determine the 10 microns "before the vessel" (line 325).

The differences between AQP4 staining in LD compared to LL is because these slices were two separate cohorts of animals, approximately one year apart in sample collection and data analysis. Therefore, beyond day/night and active/rest comparisons within groups, they cannot be compared to one another.

After rereading our description of AQP4 polarization analysis, we can understand how it may be unclear. As such, we have rewritten the section with more information:

“Equivalent 100 μm thick brain sections between -1.2 mm to -1.8 mm from Bregma were selected for staining for each mouse. Slices were permeabilized with 0.1% Triton-X-100 in PBS, blocked with 7% normal donkey serum (Jackson ImmunoResearch) in PBS with 0.03% Triton-X-100 and incubated with primary antibody overnight, followed by three washes in PBS and incubation with the fluorophore-linked secondary antibodies (Invitrogen) for two hours. Stained slices were mounted with Fluoromount G (ThermoFisher Scientific). Primary antibody used was rabbit anti-AQP4 (AB3594, Millipore), secondary antibody used was Alexa 594 donkey anti-rabbit (A21207, Invitrogen), cell nuclei were identified using DAPI (D1306, Invitrogen). A subset of animals were cardiac-perfused with wheat germ agglutinin conjugated with Alexa Fluor 488 (ThermoFisher Scientific) at a working concentration of 15 $\mu\text{g}/\text{mL}$ in PBS, followed by 4% PFA to enable better visualization of the vasculature. The brains were harvested and processed for immunohistochemistry as described above.

All analyses were done with investigators blinded to time-of-day, as images were taken by a separate investigator who randomized file names. Equivalent images of dorsal cortex, lateral cortex, ventral cortex, were imaged for each slice (3 images per slice). For a subset of animals, additional images were taken of CA1 in hippocampus and hypothalamus. The mean immunofluorescence intensity was quantified using ImageJ. For quantification of AQP4 polarization, 50 μm segments centered on blood vessels (as identified by vascular-shaped AQP4 localization) were analyzed using the line-plot tool in ImageJ. Vessels chosen were approximately 6 μm wide (range of 5-8 μm) and 32 μm long (range of 20-55 μm). Three vessels that matched this criteria were chosen per image.

For polarity calculation: The baseline was defined as the average intensity over 10 μm , -20 μm to -10 μm from peak fluorescence. Baseline fluorescence was subtracted from the peak intensity of the segment. This subtraction method is ideal because it looks at changes beyond background fluorescence localized to vascular end-feet. For each dataset (LD and LL), the adjusted peak fluorescence values were normalized to the highest value in each group for ease of comparison and visualization, giving a range from 0 to 1.”
[Pages 17, 18; Lines: 433-460]

We hope this clarifies our analysis.

Line 326: in the figures, the wheat germ agglutinin staining is referred to as "Lectin". Please also add this term in the text here.

We have removed the term “lectin” from the figure legends, and instead use “vascular staining” which we describe in the methods [Page 17, 18; Lines: 441-445].

Line 370: please define "MVX". Please also clarify that the entire neck skin was cut away, and not just the epidermis at the skin surface.

We have clarified our lymph node methodology to: Upon completion of the tracer infusion (5 min) the mice were placed supine under the macroscope (MVX10 Research Macro Zoom Microscope, Olympus) with all skin over the neck resected to reveal the superficial lymph nodes.” [Page 20; Lines: 501-503]

3. The term "mesor" in line 114 appears to be incorrect, as the values given seem to relate to the time of phase/acrophase/peak/maximum rather than the mesor value.

We apologize for this mistake. We have replaced "mesor" with "acrophase" [Page 5; Line 120].

4. Line 133. Please rephrase to replace "is necessary for" with "is linked to", "is related to", ... or similar.

We have changed this sentence to: "supporting the hypothesis that AQP4 localization to perivascular endfeet during the day promotes increased glymphatic influx." [Page 7; Line 168]

5. Line 180-181. Please clarify that CSF does not drain "directly" from the cisterna magna to lymph nodes, but first circulates through subarachnoid spaces associated with the brain or spinal column.

We have changed this to: "Appearance of tracer in the lymph node occurs as early as the first frame of recording (on average 8 min post tracer infusion start; Fig. 4bg), similar to first detectable tracer movement along the perivascular space of the middle cerebral artery (5 min post injection; Fig. 1c), suggesting that the CSF evident in the mandibular lymph nodes has not entered the glymphatic system. If the latter was the case, we would have expected a delay between influx into the brain and lymphatic efflux." [Pages 9, 10; Lines: 232-238].

6. Line 189. Please rephrase to make clear that while CSF drainage and CSF entry are temporally related, this relationship is inverse.

We have changed this sentence to: "We conclude that CSF drainage to the cervical lymph nodes is under circadian control with a peak during the active phase that temporally is in antiphase to CSF entry into the glymphatic system of the brain (Fig. 4k)." [Page 10; Lines: 244-246].

7. Line 184. Are the images of the signal with the lymph nodes saturated? If so, this does not mean that the CSF drainage to the nodes has reached a plateau. It only indicates that at the camera settings used, the detected fluorescence intensity has reached a plateau and is no longer increasing.

We tested for image saturation by reanalyzing original images, measuring what percentage of pixels within the ROI have reached the limit of brightness (255 for an 8-bit image). For both groups in LD and the rest group in LL, under 7% of all pixels within the lymph node ROIs were saturated. For the active group, less than 16% of all pixels within the ROIs reached saturation. This means that the plateaus we found in average fluorescence were most likely BSA accumulation, not an over-exposure artifact. We are not claiming this is a physiological phenomenon; just that the kinetics of our tracer change after 20 min of imaging. We believe some of the concern comes from the representatives in the figure, which were all thresholded at the same intensity across all groups to better visualize differences. All analyses were done on un-thresholded images.

Reviewer #3:

The concern comes with the design of the experiments in Figure 3 in which the mice are placed into constant light (LL) conditions. It is not clear why they chose LL vs. total darkness (DD). It is most common in the circadian field to use DD to test if an outcome is likely circadian. Also, there are several studies that indicate that constant light is associated with behavioral and physiological arrhythmicity and studies by Ohta et al., (Nature Neuroscience 2005) suggested this occurs via desynchronizing clock cells in the SCN. The data in Figure 3 show rhythmic behavior under LL but the period length is >24hr rhythm which is not their normal clock rhythm (< 24hrs for C57bl6 mice). In addition they measured cage activity with 2 mice/cage?? With these design issues the authors can conclude that the changes in fluid flow do not require a light:dark cycle but caution should be used to make interpretations to the circadian clock.

First, we would like to emphasize the significant active/inactive differences in glymphatic function and drainage of CSF to the lymph nodes, even with potential caveats of the 10d LL paradigm (Fig. 4, 5), indicating an endogenous, circadian rhythm of this physiological process. We chose LL over DD to avoid effects of light pulses that are inevitable during live-animal surgery. The surgery takes approximately 10 minutes, more than enough to phase shift mice in constant darkness^{10,11}. The mechanisms of light altering CSF influx to the brain is an ongoing line of research for our lab.

The observed average period of 25h in LL (Fig. S5ab) is positive because it shows that the animals are truly free-running in their environment. It is incorrect to assume that mice have a free-running period of less than 24h. This is true for mice with a running wheel, but without periodicity is very close 24h (Fig. S5ab)¹²⁻¹⁷. This is a problem for determining if the rhythm of influx is "circadian" compared to "diurnal" because you cannot differentiate between an endogenous rhythm or a lack of controlling for other entrainment cues^{18,19}. We have added rationale for constant light over constant dark to our results section [Page 8; Lines: 192-194].

Pittendrigh and Daan reported that: 1) free-running period of animals in LL was typically longer than 24h, 2) arrhythmicity or splitting of behavior only happened after a minimum 40 days, and 3) not all animals became arrhythmic.³⁸ These three points are consistent across LL paradigms, including Ohta et. al.³⁹. Additionally, Ohta et. al. demonstrated that the synchronicity of cells within the SCN was similar in rhythmic LL animals and animals from an LD cycle, indicating that if the animal is behaviorally rhythmic the circadian system is intact.

When designing the experiments, we first assessed the circadian behavior in DD and LL. Using chi-squared periodogram analysis we found animals in DD and LL had similar variability in periodicity, and every cage had significant Qp values from a chi-squared periodogram analysis (Fig. S5), though the amplitude of the Qp was reduced in the constant light group (Fig. S5c). Next, we made average activity profiles for 10 days in LL and DD. We found no evidence of splitting in LL (Fig. S5d). Although there was a trend toward decreased activity, specifically in the active phase in LL (Fig. S5f), there was no statistically significant difference in the average amount of activity between light cycles (Fig. S5def). It is possible that a later time point in constant light would have reduced rhythmicity and decreased activity in the mice since both long term (>1mo) LL and DD effect animal behavior, metabolism, physiology, and brain function^{20,21}. However, we can conclude that after 10-12 days circadian behavior is intact. We have added the comparison of DD and LL to the results section [Page 8; lines: 196-208; Fig. S5 (below)].

Finally, we housed mice two per cage to reduce potential hypothermia⁴⁰. The literature for group-housing of C57BL/6 mice and its effect on circadian rhythms is sparse⁴¹. Work in BALB/cByJ mice suggests that entrainment of mice by social cues require >2 animals⁴². In hamsters, social stimuli do not act as entrainment agents of circadian rhythms⁴³. Because we are only separating our studies between active and inactive, group activity data should be sufficient to determine CT, especially with significant Qp values in the X² periodogram. Based on the evidence above, we do not anticipate pairing the animals to act as a social cue, or to affect the differences in glymphatic influx observed in LL. Importantly, double housing avoids confounding effects of social isolation⁴⁴⁻⁴⁶.

Fig. S5. Comparison of mouse cage activity in constant dark (DD) and constant light (LL). (a) Average chi-squared periodogram for animals in DD (black, $n = 10$ cages of 2 mice, 1 cage of 1 mouse) and LL (yellow, $n = 6$ cages of 2 mice). Thick lines indicate mean, shading is standard error. Dotted line represents significant Qp amplitude at different periods. (b) Mean and standard deviation scatter plot of free running period in DD or LL. * $p < 0.05$. Individual cages represented by single dots. (c) Mean and standard deviation scatter plot of amplitude measures from the χ^2 test. * $p < 0.05$. Individual cages represented by single dots. (d) Average daily activity profile of animals housed in either DD ($n = 10$ cages of 2 mice) or LL ($n = 6$ cages of 2 mice) for 10 days. The thick lines and shaded error bars indicate group average \pm SEM. CT is circadian time where CT 12 is activity onset. (e) Mean and standard deviation scatter plot for total activity counts over a 24h day in DD or LL. Individual dots indicate single cage, ns: not significant. (f) Mean and standard deviation scatter plot for activity counts during the active (top) and rest (bottom) phase in DD or LL. Individual dots indicate single cage, ns: not significant.

References:

- 1 Sweeney, A. M. *et al.* In Vivo Imaging of Cerebrospinal Fluid Transport through the Intact Mouse Skull using Fluorescence Macroscopy. *J Vis Exp*, doi:10.3791/59774 (2019).
- 2 Plog, B. A. *et al.* Transcranial optical imaging reveals a pathway for optimizing the delivery of immunotherapeutics to the brain. *JCI Insight* **3**, doi:10.1172/jci.insight.120922 (2018).
- 3 Mestre, H. *et al.* Aquaporin-4-dependent glymphatic solute transport in the rodent brain. *Elife* **7**, doi:10.7554/eLife.40070 (2018).
- 4 von Holstein-Rathlou, S., Petersen, N. C. & Nedergaard, M. Voluntary running enhances glymphatic influx in awake behaving, young mice. *Neurosci Lett* **662**, 253-258, doi:10.1016/j.neulet.2017.10.035 (2018).
- 5 Xie, L. *et al.* Sleep drives metabolite clearance from the adult brain. *Science* **342**, 373-377, doi:10.1126/science.1241224 (2013).
- 6 Cai, X. *et al.* Imaging the effect of the circadian light-dark cycle on the glymphatic system in awake rats. *Proc Natl Acad Sci U S A* **117**, 668-676, doi:10.1073/pnas.1914017117 (2020).
- 7 Shan, Z. Y. *et al.* Medial prefrontal cortex deficits correlate with unrefreshing sleep in patients with chronic fatigue syndrome. *NMR Biomed* **30**, doi:10.1002/nbm.3757 (2017).
- 8 Zhang, Y. *et al.* An RNA-sequencing transcriptome and splicing database of glia, neurons, and vascular cells of the cerebral cortex. *J Neurosci* **34**, 11929-11947, doi:10.1523/jneurosci.1860-14.2014 (2014).
- 9 Lananna, B. V. *et al.* Cell-Autonomous Regulation of Astrocyte Activation by the Circadian Clock Protein BMAL1. *Cell Rep* **25**, 1-9.e5, doi:10.1016/j.celrep.2018.09.015 (2018).
- 10 Pittendrigh, C. S. & Daan, S. Vol. 106 253-266 (Journal of Comparative Physiology A, 1976).
- 11 Morin, L. P. & Studholme, K. M. Light pulse duration differentially regulates mouse locomotor suppression and phase shifts. *J Biol Rhythms* **29**, 346-354, doi:10.1177/0748730414547111 (2014).
- 12 Hablitz, L. M., Molzof, H. E., Paul, J. R., Johnson, R. L. & Gamble, K. L. Suprachiasmatic nucleus function and circadian entrainment are modulated by G protein-coupled inwardly rectifying (GIRK) channels. *J Physiol* **592**, 5079-5092, doi:10.1113/jphysiol.2014.282079 (2014).
- 13 Edgar, D. M., Martin, C. E. & Dement, W. C. Activity feedback to the mammalian circadian pacemaker: influence on observed measures of rhythm period length. *J Biol Rhythms* **6**, 185-199 (1991).
- 14 Edgar, D. M., Kilduff, T. S., Martin, C. E. & Dement, W. C. Influence of running wheel activity on free-running sleep/wake and drinking circadian rhythms in mice. *Physiol Behav* **50**, 373-378 (1991).
- 15 Kuroda, H., Fukushima, M., Nakai, M., Katayama, T. & Murakami, N. Daily wheel running activity modifies the period of free-running rhythm in rats via intergeniculate leaflet. *Physiol Behav* **61**, 633-637 (1997).
- 16 Deboer, T. & Tobler, I. Running wheel size influences circadian rhythm period and its phase shift in mice. *J Comp Physiol A* **186**, 969-973 (2000).
- 17 Harrington, M. *et al.* Behavioral and neurochemical sources of variability of circadian period and phase: studies of circadian rhythms of npy-/- mice. *Am J Physiol Regul Integr Comp Physiol* **292**, R1306-1314, doi:10.1152/ajpregu.00383.2006 (2007).
- 18 Vitaterna, M. H., Takahashi, J. S. & Turek, F. W. Overview of circadian rhythms. *Alcohol Res Health* **25**, 85-93 (2001).
- 19 Aschoff, J. Exogenous and endogenous components in circadian rhythms. *Cold Spring Harb Symp Quant Biol* **25**, 11-28, doi:10.1101/sqb.1960.025.01.004 (1960).
- 20 Hamaguchi, Y., Tahara, Y., Hitosugi, M. & Shibata, S. Impairment of Circadian Rhythms in Peripheral Clocks by Constant Light Is Partially Reversed by Scheduled Feeding or Exercise. *J Biol Rhythms* **30**, 533-542, doi:10.1177/0748730415609727 (2015).
- 21 Gonzalez, M. M. C. Dim Light at Night and Constant Darkness: Two Frequently Used Lighting Conditions That Jeopardize the Health and Well-being of Laboratory Rodents. *Front Neurol* **9**, 609, doi:10.3389/fneur.2018.00609 (2018).
- 22 Iliff, J. J. *et al.* A paravascular pathway facilitates CSF flow through the brain parenchyma and the clearance of interstitial solutes, including amyloid beta. *Sci Transl Med* **4**, 147ra111, doi:10.1126/scitranslmed.3003748 (2012).
- 23 Mestre, H. *et al.* Flow of cerebrospinal fluid is driven by arterial pulsations and is reduced in hypertension. *Nat Commun* **9**, 4878, doi: 10.1038/s41467-018-07318-3 (2018).
- 24 Da Mesquita, S., Fu, Z. & Kipnis, J. The Meningeal Lymphatic System: A New Player in Neurophysiology. *Neuron* **100**, 375-388, doi:10.1016/j.neuron.2018.09.022 (2018).

- 25 Sun, B. L. *et al.* Lymphatic drainage system of the brain: A novel target for intervention of neurological
diseases. *Prog Neurobiol* **163-164**, 118-143, doi:10.1016/j.pneurobio.2017.08.007 (2018).
- 26 Hablitz, L. M. *et al.* Increased glymphatic influx is correlated with high EEG delta power and low heart
rate in mice under anesthesia. *Sci Adv* **5**, eaav5447, doi:10.1126/sciadv.aav5447 (2019).
- 27 Ma, Q. *et al.* Rapid lymphatic efflux limits cerebrospinal fluid flow to the brain. *Acta Neuropathol* **137**,
151-165, doi:10.1007/s00401-018-1916-x (2019).
- 28 Mestre, H. *et al.* Flow of cerebrospinal fluid is driven by arterial pulsations and is reduced in
hypertension. *Nat Commun* **9**, 4878, doi:10.1038/s41467-018-07318-3 (2018).
- 29 Oster, H., Damerow, S., Hut, R. A. & Eichele, G. Transcriptional profiling in the adrenal gland reveals
circadian regulation of hormone biosynthesis genes and nucleosome assembly genes. *J Biol Rhythms*
21, 350-361, doi:10.1177/0748730406293053 (2006).
- 30 Keller, M. *et al.* A circadian clock in macrophages controls inflammatory immune responses. *Proc Natl
Acad Sci U S A* **106**, 21407-21412, doi:10.1073/pnas.0906361106 (2009).
- 31 van der Spek, R., Fliers, E., la Fleur, S. E. & Kalsbeek, A. Daily Gene Expression Rhythms in Rat White
Adipose Tissue Do Not Differ Between Subcutaneous and Intra-Abdominal Depots. *Front Endocrinol
(Lausanne)* **9**, 206, doi:10.3389/fendo.2018.00206 (2018).
- 32 Oreskovic, D. & Klarica, M. Measurement of cerebrospinal fluid formation and absorption by ventriculo-
cisternal perfusion: what is really measured? *Croat Med J* **55**, 317-327, doi:10.3325/cmj.2014.55.317
(2014).
- 33 Oreskovic, D., Klarica, M., Vukic, M. & Marakovic, J. Evaluation of ventriculo-cisternal perfusion model
as a method to study cerebrospinal fluid formation. *Croatian medical journal* **44**, 161-164 (2003).
- 34 Bachmann, S. B., Proulx, S. T., He, Y., Ries, M. & Detmar, M. Differential effects of anaesthesia on the
contractility of lymphatic vessels in vivo. *J Physiol*, doi:10.1113/jp277254 (2019).
- 35 Haspel, J. A. *et al.* Perfect timing: circadian rhythms, sleep, and immunity — an NIH workshop
summary. *JCI Insight* **5**, doi:10.1172/jci.insight.131487 (2020).
- 36 Druzd, D. *et al.* Lymphocyte Circadian Clocks Control Lymph Node Trafficking and Adaptive Immune
Responses. *Immunity* **46**, 120-132, doi:10.1016/j.immuni.2016.12.011 (2017).
- 37 Mestre, H. *et al.* Cerebrospinal fluid influx drives acute ischemic tissue swelling. *Science*,
doi:10.1126/science.aax7171 (2020).
- 38 Pittendrigh, C. S. & Daan, S. Vol. 106 333-355 (Journal of Comparative Physiology A, 1976).
- 39 Ohta, H., Yamazaki, S. & McMahon, D. G. Constant light desynchronizes mammalian clock neurons.
Nat Neurosci **8**, 267-269, doi:10.1038/nn1395 (2005).
- 40 Kallikokki, O., Teilmann, A. C., Jacobsen, K. R., Abelson, K. S. & Hau, J. The lonely mouse - single
housing affects serotonergic signaling integrity measured by 8-OH-DPAT-induced hypothermia in male
mice. *PLoS One* **9**, e111065, doi:10.1371/journal.pone.0111065 (2014).
- 41 Castillo-Ruiz, A., Paul, M. J. & Schwartz, W. J. In search of a temporal niche: social interactions. *Prog
Brain Res* **199**, 267-280, doi:10.1016/b978-0-444-59427-3.00016-2 (2012).
- 42 Paul, M. J., Indic, P. & Schwartz, W. J. Social synchronization of circadian rhythmicity in female mice
depends on the number of cohabiting animals. *Biol Lett* **11**, 20150204, doi:10.1098/rsbl.2015.0204
(2015).
- 43 Refinetti, R., Nelson, D. E. & Menaker, M. Social stimuli fail to act as entraining agents of circadian
rhythms in the golden hamster. *J Comp Physiol A* **170**, 181-187, doi:10.1007/bf00196900 (1992).
- 44 Li, B. J. *et al.* Social isolation induces schizophrenia-like behavior potentially associated with HINT1,
NMDA receptor 1, and dopamine receptor 2. *Neuroreport* **28**, 462-469,
doi:10.1097/wnr.0000000000000775 (2017).
- 45 Arakawa, H. Ethological approach to social isolation effects in behavioral studies of laboratory rodents.
Behav Brain Res **341**, 98-108, doi:10.1016/j.bbr.2017.12.022 (2018).
- 46 Matsumoto, K., Pinna, G., Puia, G., Guidotti, A. & Costa, E. Social isolation stress-induced aggression
in mice: a model to study the pharmacology of neurosteroidogenesis. *Stress* **8**, 85-93,
doi:10.1080/10253890500159022 (2005).

Reviewers' comments:

Reviewer #1 (Remarks to the Author):

The authors largely answered all the critiques with this resubmission.

We disagree on one point, and that is the importance of littermate controls. Even though they have bred their mice for 20 generations, littermate controls are very important. C57b6 raised in different locations can have drastic differences in their physiology. Mating 20 generations can also lead to specific mutations within the specific line. Littermate controls are the best way to control for these.

Reviewer #2 (Remarks to the Author):

We were very impressed with the efforts taken by the authors to address our comments, including extensive additional experiments. The authors have addressed a number of our initial concerns to our satisfaction, and we believe that this has substantially strengthened the manuscript. Some concerns, however, remain, including new issues regarding the interpretation and representation of some additional experiments. We have added line numbers to the manuscript to facilitate communications.

1) Additional major comment: Dependence of CSF influx on arousal state

In the revised manuscript, the authors included additional data assessing CSF influx in ex vivo slices from mice injected with CSF tracer while awake. We congratulate them for this feat!

However, the results showing no circadian variation in CSF influx in awake mice lead to some questions/require clarification of some aspects of the manuscript.

-in the abstract (line 47) - "independent of anesthesia", please rephrase to "independent of anesthesia type", since anesthesia itself (in comparison to an awake state) does clearly play a role

-in the introduction (line 77) - "varies across the day independent of arousal state", please rephrase to clarify that arousal state does play a role, since no variation is seen in awake mice.

Perhaps the authors mean independent of light or dark conditions?

-in the introduction (line 79) - "independent of sleep/wake state", please rephrase to clarify that solute clearance from the parenchyma was not assessed in awake mice, and that CSF influx was not increased during the day compared to the night in awake mice.

2) Additional major comment: New experiments using Evans blue infusion into the brain parenchyma and signal increase in peripheral blood to assess interstitial fluid clearance.

We thank the authors for the additional experiments on the impact of circadian rhythms on interstitial fluid clearance to peripheral blood, and for sending us the preliminary manuscript on their methodology. The use of a real-time in vivo assessment of brain parenchymal clearance to blood would have clear advantages over postmortem assessment of brain slices.

However, it is not clear that the data generated through this method are consistent with the authors' conclusions regarding circadian control of the glymphatic system. As acknowledged by the authors, due to its low molecular weight, the Evans blue shows diffusion within the brain tissue rather than an obvious efflux through a paravascular route. Do the authors have any evidence of how the dye is reaching the subarachnoid space or lymphatic vessels? Is there a direct route for this dye to the bloodstream? The initial time point assessed (15 min) appears to indicate that the dye is already within the systemic circulation. Wouldn't this be a surprisingly rapid efflux from the brain parenchyma? There is also no indication of the sensitivity of this assay. Given these concerns and the fact that the methodology has not yet been peer reviewed, it might be prudent to not include these data in the current manuscript.

3) Re: Previous major comment 1 - Glymphatic system

On the whole, the authors have taken a very constructive approach to responding to our comments, and seemed to take our comments seriously in an effort to strengthen the manuscript through the author-reviewer interaction. We were therefore somewhat surprised at their statement that "whatever 'controversy' surrounds the glymphatic system is not relevant to the conclusions of this paper". This manuscript describes novel and exciting new data implicating a circadian regulation of CSF circulation and outflow. If other mechanisms besides the proposed glymphatic hypothesis could be responsible for these data, then these mechanisms are highly relevant. We believe that in order to ensure an unbiased interpretation of their results, the authors must mention that other putative mechanisms of solute CSF-ISF exchange could also be contributing (diffusion, periarterial drainage, BBB, ...).

4) Re: Previous major comment 2 - Influx to the brain

We thank the authors for their reworking of the manuscript to use more exact terminology regarding the location of CSF influx assessed in different experiments. However, some instances remain where this is still unclear:

line 95 "CSF tracer influx to the brain": please clarify location assessed

line 110 "difference of influx": please clarify location assessed

line 237 "influx into the brain": please clarify location assessed

Additionally, we ask that the authors replace "eliminate" in line 330 with "reduce" or similar, as we are sure they agree that there is no certainty that nothing changes with death, even with their improved protocol. We would like to stress that other CSF pools except those at the cisterna magna could contribute to increased tracer spreading after death, including at the basal cisterns and in the spaces surrounding the MCA on the brain surface. These areas would not be removed by decapitation.

5) Re: previous major comment 3 - Analysis ROI for through-skull imaging

We thank the authors for adding analysis for an ROI focused on the MCA to confirm their findings. We suggest they briefly describe the extent of the original ROI utilized in Figure 1 (line 96) to make clear to the reader how the new ROI differs.

6) Re: previous major comment 5 - circadian changes in CSF production and lymphatic activity.

The authors believe it is unlikely that circadian changes in CSF production volume could have an effect on CSF influx, "since brain states regulate glymphatic influx independently of any potential endogenous rhythm in CSF production" across the day (line 315). By "brain states", do the authors mean arousal states? If so, this sentence appears to somewhat contradict their previous statement indicating that "clearance of solute from the brain varies across the day independent of arousal state" (line 76). Could the authors clarify which they are claiming is more important, circadian rhythm or arousal state? Also, perhaps the authors could rephrase "the latter scenario is less likely" (line 315) to make clear that this is their interpretation.

The authors assessed the contraction rate of lymphatic vessels under different conditions, which showed no differences. We thank the authors for carrying out this additional experiment to address whether circadian changes in one component of lymphatic activity has contributed to the results observed. We suggest that the term "contractility" be replaced with "contraction rate" throughout, since "contractility" typically includes measures of both contraction rate and contraction amplitude; these measures together affect the volume of lymph transported along the lymphatic vessel. Similarly, the authors should use the term "contractions" per min instead of "beats".

7) Re: previous major comment 6- relationship between CSF influx and CSF efflux

We thank the authors for clarifying their model of CSF influx and efflux. We suggest they also rephrase line 270 ("temporally related to glymphatic influx") to clarify that this relationship is inverse.

In addition, we suggest the authors clarify the terminology in line 237 ("lymphatic efflux") e.g. to read "tracer reaching the lymph nodes", as assessing the exact route taken by the tracer to reach these lymph nodes is beyond the scope of this study.

8) Re: previous major comment 7 - AQP4 KO mice

The authors state that the lymph node imaging of the two groups (WT and AQP4 KO mice) cannot be compared to each other due to the temporal delay between the two sets of experiments. However, they do compare CSF influx between these groups of mice and refer to it in the text (line 267). It would be helpful to the reader to clarify which measures exactly are being compared (ex vivo brain slices from KX group? Which time point?) and to include a graph of the data e.g. in Fig. 6.

Previous minor comments/additional comments:

9) Graphs

We thank the authors for their explanations of their choice of data display. We apologise if our previous comment was unclear; we were not suggesting that all data in the manuscript should be shown in the style of a column graph, but rather that all data in the column graph format (box and whiskers, plunger, 95% confidence interval) could be shown using a similar type of graph, ideally including individual data points. We appreciate the authors' wish to display as many descriptors of the data as possible, and that some forms of data display are more commonly used for certain types of data.

However, we ask that the following graphs are standardized such that the y-axis starts at 0: 4c, S1f, S5b. If the authors do not wish to start the y-axis at 0 for these graphs, a clear axis break must be included.

Similarly, we ask that the authors standardize their display of the polarization index, either by starting the y-axis at 0 throughout, or choosing one value (e.g. 0.3, as well as including a clear y-axis break) for all polarization index graphs (3c, 4g, S3).

In addition, we ask that the new graphs showing phase, amplitude, and mesor (S1e) are shown in the same style as is used for these parameters in 1f (or vice versa).

Furthermore, we ask that black points showing mean \pm standard error are added to the graphs in S1E, as are shown in similar graphs in 1d.

In Fig. 6, perhaps shading of columns in boxplots could be included as done elsewhere?

The authors should also adjust the placement of significance stars/ns as in some graphs (e.g. 5d, 6b) these have been shifted.

10) Polarization index

We thank the authors for clarifying the analysis method used to obtain the polarization index. Please could the authors clarify which "highest value" was normalized to? From the reviewer's current understanding of the text, each group was normalized to the highest value within itself (so two different normalization values per graph). Is this correct?? Alternatively, was the value normalized to the highest value across both groups in each graph, which seems a more logical approach? If so, please clarify this in the methods.

11) Methods

We thank the authors for clarifying several aspects of the methods. We ask that they also clarify the following:

Line 421: "the following day", this does not seem to be applicable if the interval was up to 36h

Line 426: "once every 15 min for 90 minutes (14 images per animal)" - it is not clear from this information how the images were acquired

Line 427: were these brains also collected using the decapitation protocol?

12) Discussion

Four references are listed for the claim that two-photon microscopy has demonstrated that CSF tracer influx occurs along penetrating arteries (line 326-329). However, reference 10 does not show beads entering the penetrating arteries - in fact the discussion clearly states that this was not occurring due to the size of the particles. Reference 50 demonstrated tracers entering the PVS of penetrating arteries only after stroke.

To put their findings into context, the authors should also cite and discuss the recent Cai et al

paper on circadian regulation of CSF influx in rats. Perhaps parts of the discussion in response to Reviewer 1's comment 4 could be included?

Reviewer #3 (Remarks to the Author):

The authors have responded significantly to the prior review. The manuscript is significantly improved with changes to the text and figures.

Additional comments from Reviewer #2 in relation to lack of littermate controls:

You are referring to what is now Figure 6, and Reviewer 1's comment 11.

They are comparing the AQP4 knockout animals here to the normal wild type animals used in the other figures, and do not assess anything in the WT littermates of the AQP4 line. They also included this comparison between AQP4 KO and unrelated C57Bl6 in the text, without including a graph or making clear which time point etc they were comparing, which we pointed out in our latest comments. It would be better if they did not try to compare these groups, since the experiments were also likely done at different times, possibly by different people.

It would be strongly recommended that wild type littermates be included, since they have previous data showing there are no differences between the AQP4 KO WT littermates and normal C57BL6 during the day but do not show this during the night. Therefore, there could be differences in circadian variation in the WT littermates from the AQP4 KO line. If they don't want to do this additional experiments, then they should not try to compare these groups, and should admit that they are not certain that there are no strain-specific circadian variations.

We would like to thank the reviewers for their constructive feedback on our manuscript. For this revision, we have included a small dataset of littermate controls, a new supplemental figure expanding upon differences between WT and KO animals, expanded rationale for our analyses, along with several revisions to the abstract, introduction, methods, results, and discussion (highlighted in yellow).

Below, we have addressed each reviewer's concerns and questions to the best of our abilities. We look forward to your feedback and would appreciate consideration for publication in your journal.

Reviewer #1 (Remarks to the Author):

We disagree on one point, and that is the importance of littermate controls. Even though they have bred their mice for 20 generations, littermate controls are very important. C57b6 raised in different locations can have drastic differences in their physiology. Mating 20 generations can also lead to specific mutations within the specific line. Littermate controls are the best way to control for these.

We would like to note that we are using wildtype, C57BL/6 mice from Charles River Laboratories when breeding all our KO mouse lines to avoid inbreeding. Since 2011, we have not observed any differences between littermate and WT controls. We have during this period utilized AQP4 KO mice in at least 8-10 reports.

We do agree that the most appropriate controls necessary for WT/AQP4KO comparison are littermate controls. Because of current lab ramp-down protocols due to COVID-19, and the resulting reduction in breeding, the amount of littermate controls in our lab has been severely reduced. Yet, we were able to include a small subset of new data. The manuscript has been edited to read:

Loss of AQP4 eliminates circadian CSF distribution

We performed CM injections of CSF tracer (BSA647) into AQP4 knockout (KO) mice during the day (ZT 4-8; $n = 10$ mice) or night (ZT 14-20; $n = 10$ mice) to test whether this day/night difference in AQP4 is necessary for day/night differences in glymphatic influx. AQP4 KO animals did not have a day/night difference in glymphatic influx, as measured by average slice intensity (t -test: $t(18) = 1.511$, $p = 0.1481$; Fig. 6abc). Because lymphatic drainage shows circadian variation that was temporally antiphase to glymphatic influx, we did the same lymph node drainage assay described above in the AQP4 KO mice during the day ($n = 6$ mice) and night ($n = 7$ mice). There was no difference in lymph node drainage after 50 minutes between day and night (t -test: $t(11) = 1.103$, $p = 0.2937$; $n = 6-7$ mice), nor in the slope of drainage calculated from the first 20 min of recording (t -test: $t(11) = 0.9202$, $p = 0.3772$; Fig. 6d-h).

The reduction in day-time influx in AQP4KO animals compared to littermate (LM) controls has already been published.^{1,2} Next, we tested whether this persists during the night, and whether there are LM/KO day/night differences in drainage to the lymph nodes. Using the ex vivo tissue processing as described above, we found no significant difference in nighttime influx (t -test: $t(14) = 0.2208$, $p = 0.8284$) between LM ($n = 3$ mice) and KO animals ($n = 13$ mice; 3 mice in addition to those shown in Fig. 6b) (Fig. S6a). Previous work has shown that LM controls for AQP4KO animals have similar day-time influx compared to C57BL6 (C57) animals.¹ Because our animals are on a C57 background, we grouped the littermate controls with the nighttime C57 under KX from Fig. 1. All three LM fit within the median quartiles of the C57 data, indicating they are comparable. Though trending toward decreased influx in KOs, there was no significant

difference between WT ($n = 37$ mice) and KO influx in the brain (t -test: $t(48) = 1.962$, $p = 0.0556$).

To measure lymph node drainage, we compared fluorescence in the mandibular lymph nodes after 30 minutes of CSF circulation during the day and night in LM ($n = 3$ mice per time point) and KO (day: $n = 9$ mice, night: $n = 10$ mice; 3 additional mice per group pooled with 30 min from pump start in Fig. 6) (Fig. S6b). With the small sample size, we found no significant differences between genotypes (2-way ANOVA: $F(1,21) = 0.184$, $p = 0.672$), time ($F(1,21) = 0.191$, $p = 0.667$), or the interaction between the two factors ($F(1,21) = 0.902$, $p = 0.353$). Because our AQP4KO line is backcrossed to C57, and the majority of LM fell within the two median quartiles of drainage distribution, we next pooled LM with C57 values at 30 min post pump start (Fig. 5) (WT day: $n = 15$ mice, WT night: $n = 18$ mice). Overall, drainage to the mandibular lymph nodes was different between groups (Kruskal-Wallis test: $H(3) = 8.464$, $p = 0.037$; Fig. 6b). Prior to Bonferroni correction, WT day vs. night, and WT night vs. KO night comparisons were significantly different ($p = 0.020$ and $p = 0.011$, respectively) supporting increased drainage only during the night in WT mice, though this effect was gone after correction.

Overall, our observations support the hypothesis that daily changes in glymphatic function and drainage of CSF to the lymph nodes is supported by daily rhythms in polarization of AQP4 to the vascular end feet of astrocytes. [Pages 10-12, Lines: 269-308]

The main purpose of this paper is to provide the first report of the circadian clock controlling CSF distribution (specifically between the brain and the mandibular lymph nodes). Secondary to this is the finding that mRNA expression of the dystrophin-associated complex (DAC) is under diurnal control, potentially influencing AQP4 localization in a circadian manner across multiple regions in the brain. The last finding was that AQP4KO animals lacked rhythmicity in CSF distribution to these compartments. We hope with the inclusion of a small subset of littermate controls, our transparent approach with regards to comparing C57 animals and littermates, and

a new supplementary figure, the reviewers agree that we have supported these conclusions.

Fig. S6. Comparison of WT and AQP4KO glymphatic influx and drainage to the lymph nodes. (a) Schematic of experiment and min/max boxplot of average mean pixel intensity for WT ($n = 3$ AQP4KO littermates [blue] and 34 C57BL6 mice [gray]) and KO ($n = 13$ mice) receiving a CM injection during the night. Individual mice represented by single dots, large dots indicate those taken in a single cohort. (c) Schematic and min/max boxplot of lymph node intensity in WT ($n = 3$ AQP4KO littermates per time point [colored] and C57 [gray; day: $n = 12$ mice, night: $n = 15$ mice]) compared to KO (day: $n = 9$ mice, night: $n = 10$ mice) at 30 min from pump start. Orange: day, blue: night. Individual mice represented by colored dots, larger dots indicate animals taken in the same cohort.

Reviewer #2 (Remarks to the Author):

1) Additional major comment: Dependence of CSF influx on arousal state

In the revised manuscript, the authors included additional data assessing CSF influx in ex vivo slices from mice injected with CSF tracer while awake. We congratulate them for this feat! However, the results showing no circadian variation in CSF influx in awake mice lead to some questions/require clarification of some aspects of the manuscript.

-in the abstract (line 47) - "independent of anesthesia", please rephrase to "independent of anesthesia type", since anesthesia itself (in comparison to an awake state) does clearly play a role

*Because of the 150 word limit, this sentence has been altered to: "We show glymphatic influx and clearance exhibit **endogenous**, circadian rhythms peaking during the mid-rest phase of mice." [Page 2, Lines: 56-57]*

-in the introduction (line 77) - "varies across the day independent of arousal state", please rephrase to clarify that arousal state does play a role, since no variation is seen in awake mice. Perhaps the authors mean independent of light or dark conditions?

The full sentence reads: "Here, we test the hypothesis that glymphatic influx of CSF tracer from the cisterna magna into the brain, and clearance of solute from the brain varies across the day independent of arousal state."

We developed experiments that were independent of sleep (different anesthetic conditions). We also included a supplemental experiment with awake animals. We find a clear, reproducible rhythm under anesthesia, but, unsurprisingly, not in the awake state when glymphatic influx is overall shut-down, as previously reported³. This was a primary hypothesis in our study, which we feel we adequately tested.

-in the introduction (line 79) - "independent of sleep/wake state", please rephrase to clarify that solute clearance from the parenchyma was not assessed in awake mice, and that CSF influx was not increased during the day compared to the night in awake mice.

*This sentence has been changed to: "We show that glymphatic influx and clearance of a small tracer from the parenchyma is increased during the day compared to night in mice **independent of the type of anesthesia or light/dark cycle**, corresponding to day/night changes in AQP4 localization." [Pages 3-4, Lines: 87-89]*

2) Additional major comment: New experiments using Evans blue infusion into the brain parenchyma and signal increase in peripheral blood to assess interstitial fluid clearance. We thank the authors for the additional experiments on the impact of circadian rhythms on interstitial fluid clearance to peripheral blood, and for sending us the preliminary manuscript on their methodology. The use of a real-time in vivo assessment of brain parenchymal clearance to blood would have clear advantages over postmortem assessment of brain slices.

However, it is not clear that the data generated through this method are consistent with the authors' conclusions regarding circadian control of the glymphatic system. As acknowledged by the authors, due to its low molecular weight, the Evans blue shows diffusion within the brain tissue rather than an obvious efflux through a paravascular route. Do the authors have any evidence of how the dye is

reaching the subarachnoid space or lymphatic vessels? Is there a direct route for this dye to the bloodstream?

We have defined the glymphatic system as a network of perivascular spaces that promotes movement of cerebrospinal fluid (CSF) into and clearance of metabolic waste from the brain. We were requested to include clearance of the system, so we included bulk clearance of solute from the parenchyma to the periphery. We show that more solute is cleared from the brain parenchyma to the periphery during the sleep-phase than the wake-phase. With this definition of the glymphatic system, and strong evidence of circadian glymphatic influx and clearance of brain solute, we feel this is consistent with our conclusions. Multiple prior reports have used a clearance assay which, in principle, is identical to the new assay included here. The main difference from prior clearance assay is that we measure efflux from brain to peripheral tissue (blood), whereas prior studies have measured what is left in the brain after predetermined time. It is widely accepted, and thousands of reports have shown, that EB does NOT pass the BBB and is therefore similar to inulin cleared from the brain by diffusion/convection within the brain tissue followed by lymphatic transport. The largest advantage of this clearance assay is that it detects the ultimate result of all clearance paths.

One important observation displayed in Fig. 6 of the supplemental methodology manuscript is that EB efflux is not increased, but rather strongly suppressed in old mice. The BBB has been proposed to open in aging⁴. Our data suggest that the BBB is not open to EB in old mice as the tracer was only cleared to a minimal degree compared with young mice.

At this time, we do not have information about dye in the subarachnoid space or lymphatic vessels, both of which would be downstream of the brain parenchyma. However, in the supplemental manuscript (see Fig. 2, Fig. 3, Fig. 5) and in the data here (Fig. 2, Fig. 4) we confirmed there was no ventricular re-perfusion of EB, with signal only limited within the brain around the injection site. This methodology was not developed to study precise routes of efflux, but to quantify total efflux independently of how the solute exits the brain. The data clearly shows that clearance is regulated in a circadian manner, in phase with glymphatic influx.

The initial time point assessed (15 min) appears to indicate that the dye is already within the systemic circulation. Wouldn't this be a surprisingly rapid efflux from the brain parenchyma?

The initial time point (15 min) is consistent with previous work that shows up to 30% of radioactive amyloid β is no longer in the brain parenchyma within 15 minutes of intrastriatal injection (see Fig. 6 in Iliff et al², <https://www.ncbi.nlm.nih.gov/pmc/articles/PMC3551275/>, and Fig. 1 in Xie et al³, <https://www.ncbi.nlm.nih.gov/pmc/articles/PMC3880190/>). Also, we see influx of tracer along the MCA within 10 minutes of CM infusion (Fig. 1, S1), and lymphatic drainage on the same time scale (Fig. 5). Based on this evidence, we conclude that fluid movement is quite quick, and are therefore not surprised to measure peripheral signal at the 15 min mark.

There is also no indication of the sensitivity of this assay.

Figures 1, 5, and 6 of the supplemental manuscript we included at last submission has information about the sensitivity of this assay. In Fig. 1 we show a dose-response curve to I.V. injected EB detectible with at as little as 1 μ g, and within 5 minutes. In Fig. 5 we show sensitivity for 0.5, 1.0, and 1.5 μ L injections of EB to the striatum, with detectible signal at all concentrations. Finally, in Fig. 6 we are able to recapitulate the radioactive experiments in old animals (see Fig. 1 in Kress et al⁵, <https://www.ncbi.nlm.nih.gov/pmc/articles/PMC4245362/>), showing our sensitivity is adequate to measure already known changes in the glymphatic system. Based on this evidence, we conclude the sensitivity of this technique is known and sufficient for our analysis of circadian-regulation of intra-parenchymal solutes.

Given these concerns and the fact that the methodology has not yet been peer reviewed, it might be prudent to not include these data in the current manuscript.

Given the evidence above, the full manuscript we provided for the reviewers as supplemental material, and that Evan's blue has been used regularly in neurovascular studies, we believe that this technique is a reliable, reproducible, sensitive, and dynamic way to measure global efflux from the brain parenchyma to the periphery. The experiments in this paper clearly demonstrate parenchymal-clearance of solute is under circadian control, and are integral to our definition of the glymphatic system. In fact, the data was included in response to the reviewers' request. Future studies, beyond the scope of circadian control of fluid movement, may investigate the exact pathways EB enters the periphery.

3) Re: Previous major comment 1 - Glymphatic system. On the whole, the authors have taken a very constructive approach to responding to our comments, and seemed to take our comments seriously in an effort to strengthen the manuscript through the author-reviewer interaction. We were therefore somewhat surprised at their statement that "whatever 'controversy' surrounds the glymphatic system is not relevant to the conclusions of this paper". This manuscript describes novel and exciting new data implicating a circadian regulation of CSF circulation and outflow. If other mechanisms besides the proposed glymphatic hypothesis could be responsible for these data, then these mechanisms are highly relevant. We believe that in order to ensure an unbiased interpretation of their results, the authors must mention that other putative mechanisms of solute CSF-ISF exchange could also be contributing (diffusion, periarterial drainage, BBB, ...).

We appreciate that the reviewer agrees that our findings that CSF is redistributed based on time-of-day is highly novel. The data support increased perivascular influx of CSF, and increased intra-parenchymal waste clearance during the day, the two key features we use in our definition of the glymphatic system. While many parts of this system remain incompletely understood, we simply aim to highlight a novel interaction of the circadian system and CSF movement.

We have already included discussions about CSF outflow, lymphatic connections, how cerebrospinal fluid production may alter glymphatic influx, and how neuronal brain activity may alter fluid movement in the parenchyma. For this revision, we have also added citations for circadian-regulation of the BBB: "In flies, there is evidence of a circadian-clock in the equivalent of the blood-brain barrier^{6,7}." [Page 15, Lines: 379-380]

With the inclusion of rhythmicity in BBB permeability, we feel this is sufficient discussion for a primary research article.

4) Re: Previous major comment 2 - Influx to the brain. We thank the authors for their reworking of the manuscript to use more exact terminology regarding the location of CSF influx assessed in different experiments. However, some instances remain where this is still unclear:

line 95 "CSF tracer influx to the brain": please clarify location assessed

*This has been changed to: "movement was imaged through the skull of anesthetized mice (Fig. 1abc, Videos S1 and S2). We found approximately 53% more CSF tracer influx to the brain during the day compared to night **in vivo along the MCA**" [Page 4, Lines: 104-105]*

line 110 "difference of influx": please clarify location assessed

The paragraph already states: "Using ex vivo tissue processing, we prepared 100 μ m thick coronal sections through the brains and quantified mean pixel fluorescence intensity across six

slices collected at 600 μ m intervals starting at 1.2 mm anterior to bregma, to calculate the mean total tracer influx per entire brain (n = 10-12 mice per time point; Fig. S1b, S2). Cosinor analysis of mean pixel intensity of brains across a 24h cycle revealed a significant rhythm in glymphatic influx with a peak at ZT 6.2, around mid-day (cosinor analysis: $F(2, 67) = 5.949$, $p = 0.004$; $R^2 = 0.151$). The difference in influx between the day and night was approximately 22% (average day influx: 7.91 ± 0.4 MPI, average night influx: 6.20 ± 0.3 MPI)." [Pages 4-5, Lines: 112-120]

line 237 "influx into the brain": please clarify location assessed.

This paragraph already describes which kinetics it was comparing: "Appearance of tracer in the lymph node occurs as early as the first frame of recording (on average 8 min post tracer infusion start; Fig. 5eg), similar to first detectable tracer movement along the perivascular space of the MCA (5 min post injection; Fig. 1c), suggesting that the CSF evident in the mandibular lymph nodes has not entered the glymphatic system. If CSF had entered the glymphatic system, we would have expected a delay between influx into the brain and lymphatic **drainage we measure in the lymph nodes.**" [Pages 9-10, Lines: 241-247]

We are sure they agree that there is no certainty that nothing changes with death, even with their improved protocol. We would like to stress that other CSF pools except those at the cisterna magna could contribute to increased tracer spreading after death, including at the basal cisterns and in the spaces surrounding the MCA on the brain surface. These areas would not be removed by decapitation.

Here, we have confirmed circadian rhythmicity to glymphatic influx both via MCA macroscopic in vivo imaging and via ex vivo tissue analysis. We would like to emphasize that there has yet to be any evidence of ex vivo tissue analysis not corroborating findings from transcranial imaging.

Three independent, direct comparisons of macroscopic live imaging and ex vivo analysis of tracer distributions show a direct relationship during the day (Fig. 1, S1abc), when altering hypertonicity (See Fig. 1-3, S2, S3, and S10 in Plog et. al.⁸, <https://insight.jci.org/articles/view/120922>), and between WT and AQP4KO animals (See Fig. 5 in Mestre et. al.¹, <https://elifesciences.org/articles/40070>). These analyses show that our method is reliable, reproducible, and reduces the impact of death on CSF redistribution.

Because other pools of CSF may contribute to the death effect, we have edited our discussion of this technique in the discussion using "reduce" instead of "eliminate": "Additionally, we have used rapid decapitation followed by drop fixation to process all of our brain samples since 2017 to **reduce** the death effect⁹ and perfusion artifact¹⁰ of CSF tracer localization in the brain." [Page 14, Lines: 357-359]

5) Re: previous major comment 3 - Analysis ROI for through-skull imaging. We thank the authors for adding analysis for an ROI focused on the MCA to confirm their findings. We suggest they briefly describe the extent of the original ROI utilized in Figure 1 (line 96) to make clear to the reader how the new ROI differs.

This has already been described, and the appropriate figure clearly cited: "Because there is also increased CSF tracer in areas outside of cortex, we re-analyzed the images using an ROI more tightly confined to the MCA and confirmed similar results (Mann Whitney test: $U = 4$, $p = 0.0260$; Fig. S1ab)." [Page 4, Lines: 107-109]

6) Re: previous major comment 5 - circadian changes in CSF production and lymphatic activity. The authors believe it is unlikely that circadian changes in CSF production volume could have an effect on CSF influx, "since brain states regulate glymphatic influx independently of any potential endogenous rhythm in CSF production" across the day (line 315). By "brain states", do the authors mean

arousal states? If so, this sentence appears to somewhat contradict their previous statement indicating that "clearance of solute from the brain varies across the day independent of arousal state" (line 76). Could the authors clarify which they are claiming is more important, circadian rhythm or arousal state? Also, perhaps the authors could rephrase "the latter scenario is less likely" (line 315) to make clear that this is their interpretation.

The reviewer is correct that the argument against CSF production mechanically altering glymphatics may be unclear and hard to interpret, especially because it is completely unexplored. Those sentences have been deleted. The paragraph now reads:

"Cerebrospinal fluid is produced in the choroid plexus (CP) of the brain, an epithelial layer of cells located within the ventricles. The CP exhibits robust cycling of the molecular clock in vitro, and can tune periodicity of molecular rhythms in SCN co-cultures and in behavior¹¹. Additionally, CSF production in humans may be rhythmic with peak CSF production during the night^{12,13}, though results may be inconclusive¹⁴. This poses at least two potential paths the CP might regulate glymphatic rhythms: first, by regulating concentration of signaling molecules in the CSF and second by mechanically supporting glymphatic influx by providing a larger volume of CSF during the rest phase. When improved methodology for CSF production in rodents becomes available^{15,16}, it should be investigated whether CSF production regulates glymphatic CSF influx along the perivascular spaces of the brain." [Pages 13-14, Lines: 340-349]

The authors assessed the contraction rate of lymphatic vessels under different conditions, which showed no differences. We thank the authors for carrying out this additional experiment to address whether circadian changes in one component of lymphatic activity has contributed to the results observed. We suggest that the term "contractility" be replaced with "contraction rate" throughout, since "contractility" typically includes measures of both contraction rate and contraction amplitude; these measures together affect the volume of lymph transported along the lymphatic vessel. Similarly, the authors should use the term "contractions" per min instead of "beats".

These changes have been made within the manuscript, Fig. 5, as well as the legend for Fig. 5.

7) Re: previous major comment 6- relationship between CSF influx and CSF efflux. We thank the authors for clarifying their model of CSF influx and efflux. We suggest they also rephrase line 270 ("temporally related to glymphatic influx") to clarify that this relationship is inverse. In addition, we suggest the authors clarify the terminology in line 237 ("lymphatic efflux") e.g. to read "tracer reaching the lymph nodes", as assessing the exact route taken by the tracer to reach these lymph nodes is beyond the scope of this study.

These changes have been made.

8) Re: previous major comment 7 - AQP4 KO mice. The authors state that the lymph node imaging of the two groups (WT and AQP4 KO mice) cannot be compared to each other due to the temporal delay between the two sets of experiments. However, they do compare CSF influx between these groups of mice and refer to it in the text (line 267). It would be helpful to the reader to clarify which measures exactly are being compared (ex vivo brain slices from KX group? Which time point?) and to include a graph of the data e.g. in Fig. 6.

Additional comments to editor: You are referring to what is now Figure 6, and Reviewer 1's comment 11. They are comparing the AQP4 knockout animals here

to the normal wild type animals used in the other figures, and do not assess anything in the WT littermates of the AQP4 line. They also included this comparison between AQP4 KO and unrelated C57BL6 in the text, without including a graph or making clear which time point etc they were comparing, which we pointed out in our latest comments. It would be better if they did not try to compare these groups, since the experiments were also likely done at different times, possibly by different people.

It would be strongly recommended that wild type littermates be included, since they have previous data showing there are no differences between the AQP4 KO WT littermates and normal C57BL6 during the day but do not show this during the night. Therefore, there could be differences in circadian variation in the WT littermates from the AQP4 KO line. If they don't want to do this additional experiments, then they should not try to compare these groups, and should admit that they are not certain that there are no strain-specific circadian variations.

We would like to note that we are using wildtype, C57BL/6 mice from Charles River Laboratories when breeding all our KO mouse lines to avoid inbreeding. Since 2011, we have not observed any differences between littermate and WT controls. We have during this period utilized AQP4 KO mice in at least 8-10 reports.

We do agree that the most appropriate controls necessary for WT/AQP4KO comparison are littermate controls. Because of current lab ramp-down protocols due to COVID-19, and the resulting reduction in breeding, the amount of littermate controls in our lab has been severely reduced. Yet, we were able to include a small subset of new data. The manuscript has been edited to read:

Loss of AQP4 eliminates circadian CSF distribution

We performed CM injections of CSF tracer (BSA647) into AQP4 knockout (KO) mice during the day (ZT 4-8; n = 10 mice) or night (ZT 14-20; n = 10 mice) to test whether this day/night difference in AQP4 is necessary for day/night differences in glymphatic influx. AQP4 KO animals did not have a day/night difference in glymphatic influx, as measured by average slice intensity (t-test: $t(18) = 1.511$, $p = 0.1481$; Fig. 6abc). Because lymphatic drainage shows circadian variation that was temporally antiphase to glymphatic influx, we did the same lymph node drainage assay described above in the AQP4 KO mice during the day (n = 6 mice) and night (n = 7 mice). There was no difference in lymph node drainage after 50 minutes between day and night (t-test: $t(11) = 1.103$, $p = 0.2937$; n = 6-7 mice), nor in the slope of drainage calculated from the first 20 min of recording (t-test: $t(11) = 0.9202$, $p = 0.3772$; Fig. 6d-h).

The reduction in day-time influx in AQP4KO animals compared to littermate (LM) controls has already been published.^{1,2} Next, we tested whether this persists during the night, and whether there are LM/KO day/night differences in drainage to the lymph nodes. Using the ex vivo tissue processing as described above, we found no significant difference in nighttime influx (t-test: $t(14) = 0.2208$, $p = 0.8284$) between LM (n = 3 mice) and KO animals (n = 13 mice; 3 mice in addition to those shown in Fig. 6b) (Fig. S6a). Previous work has shown that LM controls for AQP4KO animals have similar day-time influx compared to C57BL6 (C57) animals.¹ Because our animals are on a C57 background, we grouped the littermate controls with the nighttime C57 under KX from Fig. 1. All three LM fit within the median quartiles of the C57 data, indicating they are comparable. Though trending toward decreased influx in KOs, there was no significant difference between WT (n = 37 mice) and KO influx in the brain (t-test: $t(48) = 1.962$, $p = 0.0556$).

To measure lymph node drainage, we compared fluorescence in the mandibular lymph nodes after 30 minutes of CSF circulation during the day and night in LM ($n = 3$ mice per time point) and KO (day: $n = 9$ mice, night: $n = 10$ mice; 3 additional mice per group pooled with 30 min from pump start in Fig. 6) (Fig. S6b). With the small sample size, we found no significant differences between genotypes (2-way ANOVA: $F(1,21) = 0.184$, $p = 0.672$), time ($F(1,21) = 0.191$, $p = 0.667$), or the interaction between the two factors ($F(1,21) = 0.902$, $p = 0.353$). Because our AQP4KO line is backcrossed to C57, and the majority of LM fell within the two median quartiles of drainage distribution, we next pooled LM with C57 values at 30 min post pump start (Fig. 5) (WT day: $n = 15$ mice, WT night: $n = 18$ mice). Overall, drainage to the mandibular lymph nodes was different between groups (Kruskal-Wallis test: $H(3) = 8.464$, $p = 0.037$; Fig. 6b). Prior to Bonferroni correction, WT day vs. night, and WT night vs. KO night comparisons were significantly different ($p = 0.020$ and $p = 0.011$, respectively) supporting increased drainage only during the night in WT mice, though this effect was gone after correction.

Overall, our observations support the hypothesis that daily changes in glymphatic function and drainage of CSF to the lymph nodes is supported by daily rhythms in polarization of AQP4 to the vascular end feet of astrocytes. [Pages 10-12, Lines: 269-308]

The main purpose of this paper is to provide the first report of the circadian clock controlling CSF distribution (specifically between the brain and the mandibular lymph nodes). Secondary to this is the finding that mRNA expression of the dystrophin-associated complex (DAC) is under diurnal control, potentially influencing AQP4 localization in a circadian manner across multiple regions in the brain. The last finding was that AQP4KO animals lacked rhythmicity in CSF distribution to these compartments. We hope with the inclusion of a small subset of littermate controls, our transparent approach with regards to comparing C57 animals and littermates, and a new supplementary figure, the reviewers agree that we have supported these conclusions.

Fig. S6. Comparison of WT and AQP4KO glymphatic influx and drainage to the lymph nodes. (a) Schematic of experiment and min/max boxplot of average mean pixel intensity for WT ($n = 3$ AQP4KO littermates [blue] and 34 C57BL6 mice [gray]) and KO ($n = 13$ mice) receiving a CM injection during the night. Individual mice represented by single dots, large dots indicate those taken in a single cohort. (c) Schematic and min/max boxplot of lymph node intensity in WT ($n = 3$ AQP4KO littermates per time point [colored] and C57 [gray; day: $n = 12$ mice, night: $n = 15$ mice]) compared to KO (day: $n = 9$ mice, night: $n = 10$ mice) at 30 min from pump start. Orange: day, blue: night. Individual mice represented by colored dots, larger dots indicate animals taken in the same cohort.

Previous minor comments/additional comments:

9) Graphs. We thank the authors for their explanations of their choice of data display. We apologise if our previous comment was unclear; we were not suggesting that all data in the manuscript should be shown in the style of a column graph, but rather that all data in the column graph format (box and whiskers, plunger, 95% confidence interval) could be shown using a similar type

of graph, ideally including individual data points. We appreciate the authors' wish to display as many descriptors of the data as possible, and that some forms of data display are more commonly used for certain types of data.

However, we ask that the following graphs are standardized such that the y-axis starts at 0: 4c, S1f, S5b. If the authors do not wish to start the y-axis at 0 for these graphs, a clear axis break must be included.

We would like to highlight that an abbreviated search of Nature Communications publications between March 31 and April 2 2020 (only three days) found at least 9 articles, in the Biological Sciences section alone, that included at least one graph with an axis that did not start at 0, without an axis break (hyperlinks below). This indicates that starting an axis at 0, or with a break, is not always necessary or appropriate.

Circadian periodicity data is frequently plotted without a break to improve visualization of the data (for a small subset of examples see: ¹⁷⁻²²), so we do not believe that there is a basis for changing Fig. S5b.

We would like to emphasize that Fig. S1f cannot be compared to any other figure because a different volume of BSA647 was used for the awake animals (stated both in the previous response letter, and in the methods section [Page 16, Lines: 411-415]).

Finally, starting the y axis at 0 for Fig. 4c and S1f will not provide new information. The mean intensity will never be 0, and thus starting at 0 is uninformative. However, due to concern about visualization, we confirmed that both Fig. 4c and Fig. S1f y-axes start at 2 AU.

Recent articles in the Biological Sciences section of Nature Communications:

<https://www.nature.com/articles/s41467-020-15423-5>

<https://www.nature.com/articles/s41467-020-15312-x>

<https://www.nature.com/articles/s41467-020-15413-7>

<https://www.nature.com/articles/s41467-020-15361-2>

<https://www.nature.com/articles/s41467-020-15420-8>

<https://www.nature.com/articles/s41467-020-15425-3>

<https://www.nature.com/articles/s41467-020-15361-2>

<https://www.nature.com/articles/s41467-020-15457-9>

<https://www.nature.com/articles/s41467-020-15483-7>

Similarly, we ask that the authors standardize their display of the polarization index, either by starting the y-axis at 0 throughout, or choosing one value (e.g. 0.3, as well as including a clear y-axis break) for all polarization index graphs (3c, 4g, S3).

Normalizing the y-axis between polarization index graphs in different cohorts of animals and staining, across different experimental paradigms that were analyzed individually, would not add useful information and is therefore not necessary. Adding breaks in the y-axis would not be informative as no value measured would ever be 0, intrinsic to the methodology. The purpose of these graphs are to compare two different phases of the clock in single cohorts of animals (one cohort per figure).

In addition, we ask that the new graphs showing phase, amplitude, and mesor (S1e) are shown in the same style as is used for these parameters in 1f (or vice versa).

Because the data in Fig. S1e are three distinct outputs of mesor and/or phase information, it is appropriate and intentional to have them visually distinct. The CircWave analysis does not give 95% confidence intervals, it provides a center of gravity (CoG) estimate with standard deviation

estimates for mesor and phase (to download CircWave: <https://www.euclock.org/results/item/circ-wave.html>; for applications see: ²³⁻²⁵). We use a dot plot with error bars to most accurately display these estimates. We use the same boxplot style as Fig. 1 to re-graph the Cosinor 95% confidence intervals of mesor and phase so readers can clearly delineate what data they have already seen. Finally, because the peak of the CircWave curve is not the same as the CoG, and CircWave does not give confidence interval estimates for the fitted curve, we use a single point to represent the peak.

Furthermore, we ask that black points showing mean +/- standard error are added to the graphs in S1E, as are shown in similar graphs in 1d.

The mean ± standard error time point estimates in Fig. 1d are intentionally not included in Fig. S1e to eliminate redundancy in figures, and to help the reader to better visualize the CoG estimates which is the purpose of this panel.

In Fig. 6, perhaps shading of columns in boxplots could be included as done elsewhere?

The lack of shading for boxplots in Fig. 6 is intentional, so readers do not confuse experiments. The purpose of Fig. 6 was to look at the effects of AQP4KO on rhythms in glymphatic influx and lymphatic drainage.

The authors should also adjust the placement of significance stars/ns as in some graphs (e.g. 5d, 6b) these have been shifted.

Significance indicators have been fixed.

10) Polarization index. We thank the authors for clarifying the analysis method used to obtain the polarization index. Please could the authors clarify which "highest value" was normalized to? From the reviewer's current understanding of the text, each group was normalized to the highest value within itself (so two different normalization values per graph). Is this correct?? Alternatively, was the value normalized to the highest value across both groups in each graph, which seems a more logical approach? If so, please clarify this in the methods.

We apologize for any confusion or concern. As the reviewer suggests in the alternative: the normalization was made to the highest value out of all groups compared in the statistical analysis, one value per graph. This has been clarified in the methods. [Page 19, Lines: 488-491]

11) Methods. We thank the authors for clarifying several aspects of the methods. We ask that they also clarify the following:

Line 421: "the following day", this does not seem to be applicable if the interval was up to 36h

*This section now reads: "All experiments occurred between CT 2-7 or CT 14-19 depending on if it was the rest or active phase of the animal, respectively. Cannulas (guide cannula: 26G, C315G SPC, 4.5 mm below pedestal, dummy: 33G, C315DC/SP, 0.1 mm projection; PlasticsOne, Roanoke, VA) were implanted into mice 24-36h prior to experimentation into striatum (AP: +0.6 from Bregma, LM: -2 and DV: -3.25mm). **At the appropriate experimental time points**, mice in either LD or 10 days LL were anesthetized with KX and fitted with a microdialysis probe (inner cannula: 33G, C315I/SP, 0.1 mm projection), had skin over the femoral vein resected to better visualize EB in the blood stream, and EB was injected into the brain (4% in ACSF, 1µL, 0.2 µL/min)." [Pages 17-18, Lines: 446-453]*

Line 426: "once every 15 min for 90 minutes (14 images per animal)" - it is not clear from this information how the images were acquired

The section now reads: "Images of the femoral vein were taken under a macroscope (MVX10 Research Macro Zoom Microscope, Olympus) using the Cy5 filter set, for EB visualization, and GFP, for venous structure visualization, once every 15 minutes for 90 minutes (14 images/animal) **beginning at the start of EB injection.**" [Page 18, Lines: 453-457]

Line 427: were these brains also collected using the decapitation protocol?

This was already addressed in the next sentence following line 427 (now line 446): "Brains were collected upon experiment end, drop-fixed overnight in PFA and sliced the next day in the same manner as the CSF influx experiments. Slices were used to ensure parenchymal injections did not hit the ventricle and volume of injection, as measured by % area, was similar between animals." [Page 18, Lines: 457-460]

We also included the information in the discussion: "Additionally, we have used rapid decapitation followed by drop fixation to process all of our brain samples since 2017 to reduce the death effect⁹ and perfusion artifact¹⁰ of CSF tracer localization in the brain." [Page 14, Lines: 357-359]

12) Discussion. Four references are listed for the claim that two-photon microscopy has demonstrated that CSF tracer influx occurs along penetrating arteries (line 326-329). However, reference 10 does not show beads entering the penetrating arteries - in fact the discussion clearly states that this was not occurring due to the size of the particles. Reference 50 demonstrated tracers entering the PVS of penetrating arteries only after stroke.

We have removed reference 10 and 50. The sentence now reads: "Our lab has already demonstrated CSF tracer alongside penetrating arterioles in live animals using two-photon microscopy^{2,3,26}, which has considerably higher spatial resolution than macroscopic imaging^{10,27}." [Page 14, Lines: 354-357] (Note: reference numbers here may not correspond to numbers in the manuscript text, but the articles are the same.)

To put their findings into context, the authors should also cite and discuss the recent Cai et al paper on circadian regulation of CSF influx in rats. Perhaps parts of the discussion in response to Reviewer 1's comment 4 could be included?

Although the Cai et al paper seems comparable, and our EB clearance data supports the author's hypothesis that day/night differences were due to increased clearance of tracer during the light phase, our awake-influx study and Cai et al had fundamental differences in methodology:

- The Cai et al paper functionally sleep deprived animals by training the light phase cohort for 5 days prior to experimentation. Sleep deprivation can alter cognition, metabolism, stress response, physical activity, temperature regulation, food intake, and more.²⁸ Additionally, the authors' baseline subtraction of their T1 images may have been impacted by sleep deprivation in only one group.²⁹ To our knowledge, the effects of sleep deprivation on glymphatic function has not been published.
- The idea that day/night differences could be seen in awake conditions is highly unlikely. Our original findings were that wakefulness decreased penetration of CSF tracer approximately seven-fold (See Fig. 1 in ³). The circadian variation we find under anesthesia is approximately 20%. It would be extremely difficult to find 20% changes when glymphatic influx is essentially shut down in the awake state. Additionally, they used intra-ventricular contrast agent delivery, a process known to decrease glymphatic

influx (See Fig. 7 in ¹). Indeed, the authors report no significant changes in whole-brain signal.

Because of the differences in technique, experimental paradigm, species, and the caveats included above, we feel that including a discussion of the Cai et al paper in our primary report would be cumbersome, misleading and unproductive. Such a discussion is appropriate for an opinion review.

Reviewer #3 (Remarks to the Author):

The authors have responded significantly to the prior review. The manuscript is significantly improved with changes to the text and figures.

References

- 1 Mestre, H. *et al.* Aquaporin-4-dependent glymphatic solute transport in the rodent brain. *Elife* **7**, doi:10.7554/eLife.40070 (2018).
- 2 Iliff, J. J. *et al.* A paravascular pathway facilitates CSF flow through the brain parenchyma and the clearance of interstitial solutes, including amyloid beta. *Sci Transl Med* **4**, 147ra111, doi:10.1126/scitranslmed.3003748 (2012).
- 3 Xie, L. *et al.* Sleep drives metabolite clearance from the adult brain. *Science* **342**, 373-377, doi:10.1126/science.1241224 (2013).
- 4 Montagne, A. *et al.* Blood-brain barrier breakdown in the aging human hippocampus. *Neuron* **85**, 296-302, doi:10.1016/j.neuron.2014.12.032 (2015).
- 5 Kress, B. T. *et al.* Impairment of paravascular clearance pathways in the aging brain. *Ann Neurol* **76**, 845-861, doi:10.1002/ana.24271 (2014).
- 6 Cuddapah, V. A., Zhang, S. L. & Sehgal, A. Regulation of the Blood-Brain Barrier by Circadian Rhythms and Sleep. *Trends Neurosci* **42**, 500-510, doi:10.1016/j.tins.2019.05.001 (2019).
- 7 Zhang, S. L., Yue, Z., Arnold, D. M., Artiushin, G. & Sehgal, A. A Circadian Clock in the Blood-Brain Barrier Regulates Xenobiotic Efflux. *Cell* **173**, 130-139.e110, doi:10.1016/j.cell.2018.02.017 (2018).
- 8 Plog, B. A. *et al.* Transcranial optical imaging reveals a pathway for optimizing the delivery of immunotherapeutics to the brain. *JCI Insight* **3**, doi:10.1172/jci.insight.120922 (2018).
- 9 Ma, Q. *et al.* Rapid lymphatic efflux limits cerebrospinal fluid flow to the brain. *Acta Neuropathol* **137**, 151-165, doi:10.1007/s00401-018-1916-x (2019).
- 10 Mestre, H. *et al.* Flow of cerebrospinal fluid is driven by arterial pulsations and is reduced in hypertension. (*Submitted*) (2018).
- 11 Myung, J. *et al.* The choroid plexus is an important circadian clock component. *Nat Commun* **9**, 1062, doi:10.1038/s41467-018-03507-2 (2018).
- 12 Nilsson, C., Stahlberg, F., Gideon, P., Thomsen, C. & Henriksen, O. The nocturnal increase in human cerebrospinal fluid production is inhibited by a beta 1-receptor antagonist. *Am J Physiol* **267**, R1445-1448, doi:10.1152/ajpregu.1994.267.6.R1445 (1994).
- 13 Nilsson, C. *et al.* Circadian variation in human cerebrospinal fluid production measured by magnetic resonance imaging. *Am J Physiol* **262**, R20-24, doi:10.1152/ajpregu.1992.262.1.R20 (1992).
- 14 Takahashi, H., Tanaka, H., Fujita, N., Murase, K. & Tomiyama, N. Variation in supratentorial cerebrospinal fluid production rate in one day: measurement by nontriggered phase-contrast magnetic resonance imaging. *Jpn J Radiol* **29**, 110-115, doi:10.1007/s11604-010-0525-y (2011).
- 15 Oreskovic, D., Klarica, M., Vukic, M. & Marakovic, J. Evaluation of ventriculo-cisternal perfusion model as a method to study cerebrospinal fluid formation. *Croatian medical journal* **44**, 161-164 (2003).
- 16 Oreskovic, D. & Klarica, M. Measurement of cerebrospinal fluid formation and absorption by ventriculo-cisternal perfusion: what is really measured? *Croatian medical journal* **55**, 317-327, doi:10.3325/cmj.2014.55.317 (2014).
- 17 Fernandez, D. C. *et al.* Light Affects Mood and Learning through Distinct Retina-Brain Pathways. *Cell* **175**, 71-84.e18, doi:10.1016/j.cell.2018.08.004 (2018).
- 18 Tso, C. F. *et al.* Astrocytes Regulate Daily Rhythms in the Suprachiasmatic Nucleus and Behavior. *Curr Biol*, doi:10.1016/j.cub.2017.02.037 (2017).
- 19 Brancaccio, M. *et al.* Cell-autonomous clock of astrocytes drives circadian behavior in mammals. *Science* **363**, 187-192, doi:10.1126/science.aat4104 (2019).

- 20 Corty, R. W., Kumar, V., Tarantino, L. M., Takahashi, J. S. & Valdar, W. Mean-Variance QTL Mapping Identifies Novel QTL for Circadian Activity and Exploratory Behavior in Mice. *G3 (Bethesda)* **8**, 3783-3790, doi:10.1534/g3.118.200194 (2018).
- 21 Rijo-Ferreira, F. *et al.* Sleeping sickness is a circadian disorder. *Nat Commun* **9**, 62, doi:10.1038/s41467-017-02484-2 (2018).
- 22 Musiek, E. S. *et al.* Circadian clock proteins regulate neuronal redox homeostasis and neurodegeneration. *J Clin Invest* **123**, 5389-5400, doi:10.1172/jci70317 (2013).
- 23 Oster, H., Damerow, S., Hut, R. A. & Eichele, G. Transcriptional profiling in the adrenal gland reveals circadian regulation of hormone biosynthesis genes and nucleosome assembly genes. *J Biol Rhythms* **21**, 350-361, doi:10.1177/0748730406293053 (2006).
- 24 Keller, M. *et al.* A circadian clock in macrophages controls inflammatory immune responses. *Proc Natl Acad Sci U S A* **106**, 21407-21412, doi:10.1073/pnas.0906361106 (2009).
- 25 van der Spek, R., Fliers, E., la Fleur, S. E. & Kalsbeek, A. Daily Gene Expression Rhythms in Rat White Adipose Tissue Do Not Differ Between Subcutaneous and Intra-Abdominal Depots. *Front Endocrinol (Lausanne)* **9**, 206, doi:10.3389/fendo.2018.00206 (2018).
- 26 Iliff, J. J. *et al.* Cerebral arterial pulsation drives paravascular CSF-interstitial fluid exchange in the murine brain. *J Neurosci* **33**, 18190-18199, doi:10.1523/jneurosci.1592-13.2013 (2013).
- 27 Mestre, H. *et al.* Cerebrospinal fluid influx drives acute ischemic tissue swelling. *Science*, doi:10.1126/science.aax7171 (2020).
- 28 Hirotsu, C., Tufik, S. & Andersen, M. L. Interactions between sleep, stress, and metabolism: From physiological to pathological conditions. *Sleep Sci* **8**, 143-152, doi:10.1016/j.slsci.2015.09.002 (2015).
- 29 Shan, Z. Y. *et al.* Medial prefrontal cortex deficits correlate with unrefreshing sleep in patients with chronic fatigue syndrome. *NMR Biomed* **30**, doi:10.1002/nbm.3757 (2017).

REVIEWER COMMENTS

Reviewer #1 (Remarks to the Author):

Authors addressed my concern.

Reviewer #2 (Remarks to the Author):

The authors have addressed the majority of our comments. However, we urge strongly that the authors remove the data from the manuscript related to the EB clearance assay, as this method remains unpublished and unvalidated at this stage. We have significant concerns regarding the interpretation of the data from this assay, first and foremost, being that EB in such a percentage (4%) directly into the brain may cause toxic effects that could alter the mechanisms of clearance. Please see discussion in Saunders et al, *Frontiers in Neuroscience*, 2015 regarding potential toxic effects of this dye and its unsuitability for assessments of BBB permeability. Please note that the discussion in this review refers to systemic application of the dye in which the concentrations affecting the blood vessels of the brain would be even far lower than used in the current manuscript. The authors explanation for the rapid efflux to blood observed at similar time frames as seen for amyloid beta is not valid, as amyloid beta has known transporters for this process.

Previous minor comment: "once every 15 minutes for 90 minutes (14 images/animal) beginning at the start of EB injection." [Page 18, Lines: 453-457] – It is still not clear how the math was worked out here.

Previous minor comment: Although the Cai et al report does have fundamental differences in the study design, this published paper should still be mentioned in the manuscript as it is directly relevant to the topic of this study.

Reviewer #3 (Remarks to the Author):

The authors have responded well to prior concerns.

We would like to thank the reviewers for their constructive feedback on our manuscript. For this revision, we have included more information on the lack of toxicity of Evans Blue (EB) in our methodology to measure solute clearance from the brain. Specifically, we have included: (1) an exhaustive literature search for toxicity of EB, (2) data demonstrating that EB injection is not neurotoxic, (3) new data demonstrating intrastriatal injection of fluid (and EB specifically) does not change glymphatic influx, and, finally, (4) data demonstrating that EB injection does not alter brain EEG activity. We have also incorporated the Cai et al¹ reference in our discussion section.

Below, we have addressed each reviewer's concerns and questions to the best of our abilities. We look forward to your feedback and would appreciate consideration for publication in your journal.

Reviewer #1 (Remarks to the Author):

Authors addressed my concern.

We thank the reviewer for the constructive critiques.

Reviewer #2 (Remarks to the Author):

The authors have addressed the majority of our comments. However, we urge strongly that the authors remove the data from the manuscript related to the EB clearance assay, as this method remains unpublished and unvalidated at this stage. We have significant concerns regarding the interpretation of the data from this assay, first and foremost, being that EB in such a percentage (4%) directly into the brain may cause toxic effects that could alter the mechanisms of clearance. Please see discussion in Saunders et al, *Frontiers in Neuroscience*, 2015 regarding potential toxic effects of this dye and its unsuitability for assessments of BBB permeability. Please note that the discussion in this review refers to systemic application of the dye in which the concentrations affecting the blood vessels of the brain would be even far lower than used in the current manuscript.

We have approached the question of toxicity of EB in multiple ways.

*First, we did a comprehensive literature search, including Saunders et al, *Frontiers in Neuroscience*, 2015, for toxicity of EB in neuroscience research.*

Saunders et al provides unconvincing arguments for the acute toxicity of EB. The primary evidence of EB toxicity is cited from two studies. First, a 1944 study tested different doses of intravenous EB in different animal models, and showed that months after administering 25mg/kg (5 rats, 1 died at 1 month) or 50mg/kg (5 rats, 2 died at 3 months) the majority of rats survived for 5-6 months (sacrificed and analyzed at this time).² In fact, this 1944 study shows EB is less toxic to rats compared to dogs, cats and rabbits, with rats able to handle much higher doses of EB. It is misleading when Saunders et al cites the 1944 study a lower concentration than modern BBB assays, because even though it is 0.5% EB compared to modern 2%-4% EB, the effective mg/kg dosage is the same (approximately 50 mg/kg for both). The second study from 1962, used varying doses of intravenous EB given to monkeys (25, 50, 100, and 200mg/kg) and then observed for several weeks. Higher doses were lethal (50mg/kg-11 days, 100mg/kg-7 days, 200mg/kg-4 days)³. Saunders et al do not specify this information and state "died within

days of injection". It is highly suspect to apply this 1962 monkey study to rodent models due to the extremely high doses used, and the 1944 study showing rats survive higher doses of EB than larger animals (rabbits, cats or dogs).

Both studies discussed above examine very long term outcomes (weeks to months) at high doses (50mg/kg or more) compared to our acute assay with relatively small doseage (90 minutes, 1.6mg/kg). Saunders et al merely speculates on acute toxicity of EB:

"The duration of the experiments [BBB assays] is generally much less than in Hueper and Ichniowski (1944), in the range of 10 min to 24 h. So it is unclear whether there would have been histologically detectable lesions within that time span. There appear to be no reports on this."

We have collected new data, to be paired with the supplemental manuscript provided, that demonstrates there is in no toxicity 2h after intra-striatal injected EB (**Auxiliary Fig. 1; below**).

EB is extensively used for vascular permeability assays. We searched pubmed for "evans blue" and sort by publications in the last 4 years and found over 1,200 references (<https://pubmed.ncbi.nlm.nih.gov/?term=evans+blue&filter=years.2016-2020&timeline=expanded>), and have included several of these references in the attached table illustrating recent publications utilizing EB (**below**), examining effects it has on vesicular transporters, and examining the toxicity of EB. Note that all the papers that are indeterminate or report "toxicity" use higher doses, longer incubation times, and/or slice electrophysiology/cell culture models where perfusion is nonexistent. We would like to highlight Yin et al in 2019⁴, which demonstrated intrathecal infusion of EB (2mg/kg - 50µg), a similar dose to our assay (1.6mg/kg - 40µg), improved recovery from spinal nerve ligation in rats. This study represents the closest route of administration and dose to our assay (intra-striatal infusion) and showed no toxicity from EB. We believe this exhaustive literature search supports our conclusions that acute administration of EB at the low dose used in our clearance assay is not toxic to the mouse, and investigating EB's long term effect on mice is well outside the scope of our acute (90 minute) assay.

Additionally, we have expanded upon data in Pla et al, the supplemental manuscript provided in the past two reviews. We included a full analysis of glymphatic influx (**Auxiliary Fig. 2; below**) where we demonstrate that EB does not alter CSF inflow. Then, we used immunohistochemistry for markers of neuroinflammation, specifically markers of gliosis, and found no change between artificial cerebrospinal fluid (aCSF) and EB infusion at the cannula site, indicating EB does not cause significant damage to the tissue independently of cannula placement (**Auxiliary Fig. 1; below**). Finally, we have included a new data set to the supplemental manuscript where we used dual placement of EEG recordings and cannula infusion to demonstrate there is no definable difference in brain EEG activity upon administration of EB compared to aCSF (**Auxiliary Fig. 3; below**).

Based on the literature search (**See summary table; below**), unaltered CSF influx with EB administration (**Auxiliary Fig. 2; below**), no reactive gliosis due to brain injury (**Auxiliary Fig. 1; below**), and the lack of change in cortical activity upon EB administration (**Auxiliary Fig. 3; below**), we conclude that the acute, non-recovery in vivo imaging of EB clearance we use in this paper is not confounded by any potential EB toxicity in the brain. Methodology is described in the supplemental manuscript, EEG was done as published previously^{5,6}.

[REDACTED]

[REDACTED]

[REDACTED]

[REDACTED]

The author's explanation for the rapid efflux to blood observed at similar time frames as seen for amyloid beta is not valid, as amyloid beta has known transporters for this process.

We were using data both from previous literature, and measurements from this study specifically, to show we were not surprised that there is rapid efflux from the brain parenchyma.

In the last response, we explicitly gave A β as an example, but both papers cited used more than one radioactive molecule to determine whether there was a conserved waste clearance pathway. Specifically: data from previous reports has demonstrated AQP4 is necessary for clearance of both radioactive A β and radioactive mannitol (see see Fig. 5 and Fig. 6 in Iliff et al⁷, <https://www.ncbi.nlm.nih.gov/pmc/articles/PMC3551275/>), and sleep increases clearance of both radioactive A β and radioactive inulin (see Fig. 1 in Xie et al⁶, <https://www.ncbi.nlm.nih.gov/pmc/articles/PMC3880190/>). All three of these molecules are very different, yet are highly correlated with CSF influx measurements and consistent with our glymphatic hypothesis. Combined with influx of tracer along the MCA within 10 minutes of CM infusion (Fig. 1, S1), and lymphatic drainage on the same time scale (Fig. 5), we conclude that fluid movement in the brain and body is quite quick, and are therefore not surprised to measure a small, but significant peripheral signal at the 15 min mark.

Previous minor comment: “once every 15 minutes for 90 minutes (14 images/animal) beginning at the start of EB injection.” [Page 18, Lines: 453-457] – It is still not clear how the math was worked out here.

The full section reads: “Images of the femoral vein were taken under a macroscope (MVX10 Research Macro Zoom Microscope, Olympus) using the Cy5 filter set, for EB visualization, and GFP, for venous structure visualization, once every 15 minutes for 90 minutes (14 images/animal) beginning at the start of EB injection.” (Page 18, Lines: 444 – 448)

There were two images taken at each time point. There were 7 time points. Therefore, total, there were 14 images/animal.

Previous minor comment: Although the Cai et al report does have fundamental differences in the study design, this published paper should still be mentioned in the manuscript as it is directly relevant to the topic of this study.

This reference has been included in the discussion. “Recent work hypothesized increased glymphatic clearance during the sleep phase, driving rhythmic fluid flow in the brain¹. “ (Page 15, Lines: 366-367)

Reviewer #3 (Remarks to the Author):

The authors have responded well to prior concerns.

We thank the reviewer for the constructive critiques.

Uses Evans Blue as an assay, as a therapy, or any otherwise nontoxic use.	Evidence for toxicity of Evans blue	Studies showing Evans Blue interactions and activity.
---	-------------------------------------	---

Year	Title	Purpose of study	EB dose	EB administration	Circulation Time	Model organism	Outcome
2020	In vivo evaluation of glymphatic efflux by imaging efflux to the vascular compartment	Measurement of interstitial solute clearance from the brain.	40ug - 1.6mg/kg - 1uL 4%EB	Intrastriatal infusion	2 hours	C57BL/6 Mouse	EB clearance from brain in real time in vivo. No lesions in brains after EB. No change in EEG after 2 hours under K/X anesthesia. No change in glymphatic influx.
2020	Autophagy-mediated occludin degradation contributes to blood-brain barrier disruption during ischemia in bEnd.3 brain endothelial cells and rat ischemic stroke models	Use MCAO in rats, measure levels of autophagy and BBB disruption.	80mg/kg - 2% 4mL/kg	I.V. tail vein. 20 hours after MCAO	4 hours	Sprague-Dawley Rats	EB used to evaluate BBB disruption
2019	Matrix metalloproteinase-9 and p53 involved in chronic fluorosis induced blood-brain barrier damage and neurocyte changes	Evaluate effects of high fluoride water on blood-spinal cord barrier. Use EB assay	60mg/kg - 2% 3mL/kg	I.V. (exact route not specified)	Perfused acutely (time not specified)	Wistar Rats	Disruption of blood-spine-barrier as measured by EB.
2019	Permeability of Intestinal and Blood-Tissue Barriers in Rats for Evans Blue Dye under Conditions of Acute Intoxication with Cyclophosphamide.	Cyclophosphamide effects on intestinal and blood barriers measured by EB	50mg/kg	I.V. tail vein, or by gavage	3 hour circulation	Outbred Albino Rats	Measured EB extravasation in blood, brain, lung, liver, kidney, and intestine.
2019	Evans Blue Reduces Neuropathic Pain Behavior by Inhibiting Spinal ATP Release	Rats given spinal nerve ligation (SNL). EB given as VNUT inhibitor	1 week after SNL 5,15,50,100ug	Intrathecal infusion of EB	7 days	Sprague-Dawley Rats	50ug EB attenuated pain behavior, improved gait, reduced activated microglia, decreased IL-1B IL-6.
2018	Evans Blue Dye: A Revisit of Its Applications in Biomedicine	Review of EB use	N/A	N/A	N/A	N/A	Cites recent papers that use EB, disputes EB toxicity in small doses.
2018	An Optimized Evans Blue Protocol to Assess Vascular Leak in the Mouse	Technical JOVE paper	60mg/kg - 50uL 3%	I.V. Jugular Vein catheter	20 minutes	Mouse	EB extravasation in bladder, duodenum, kidney, lung, tail
2017	Regardless of etiology, progressive renal disease causes ultrastructural and functional alterations of peritubular capillaries	2-photon microscopy to measure extravasation of EB in kidney disease models	5mg/kg - 10uL 1%	I.V. Tail Vein	35 minutes	C57, 129SvJ Mouse	Increased extravasation of EB in fibrotic kidneys
2016	Hypertonic saline alleviates experimentally induced cerebral	Test hypertonic saline to reduce edema in	80mg/kg - 2% 4mL/kg	I.V. Tail vein, right after reperfusion	24 hours	Sprague-Dawley Rats	10% Hypertonic saline reduced edema, reduced EB

	oedema through suppression of vascular endothelial growth factor and its receptor VEGFR2 expression in astrocytes.	MCAO model in rats. Measured brain water content and BBB permeability					extravasation.
2013	An in vivo Assay to Test Blood Vessel Permeability	Technical JOVE paper	40mg/kg - 200uL 0.5%	I.V. Lateral tail vein	30 minutes	Mouse	EB extravasation in tissues: (Liver, Lung, Kidney, skin)
2011	Comparison Evans Blue Injection Routes: Intravenous vs. Intraperitoneal, for Measurement of Blood-Brain Barrier in a Mice Hemorrhage Model.	Evaluate efficiency of EB administration methods in mice given Intracerebral Hemorrhage.	80mg/kg - 2% 4mL/kg	I.V. Jugular Vein or Intraperitoneal	3 hours	CD-1 Mice	Evans Blue accumulates in brains of mice after ICH. No statistical difference between IP and IV
2001	Cellular resistance to Evans blue toxicity involves an up-regulation of a phosphate transporter implicated in vesicular glutamate storage	Evaluate adaptation of cultured neurons to prolonged EB exposure	10nM	put in culture media	4 days, most cells died, 10 days+	NG108-15 cultured neurons	Only toxic to neurons in culture after prolonged EB exposure
1962	Toxicity of Evans Blue Dye in the Monkey and Tracing of It in the Tooth Pulp	To determine what happens when injecting various amounts of EB into a monkey	25,50,100,200 mg/kg	I.V.	Until death or 5 weeks	Monkey	50mg died 11 days. 100mg died 7 days. 200mg died 4 days. 25mg/kg survived until sacrifice.
1944	TOXICOPATHOLOGIC STUDIES ON THE DYE T-1824	To determine what happens over time when injecting various amounts of Evans blue intravenously.	Dogs: 5,15,25,50,125 mg/kg Cats: 5,15,25,50,125 Rabbits: 5,15,25,50 Rats: 5,15,25,50	I.V. in jugular vein or femoral vein. Cohort of rats injected with 10mg/kg I.P.	5-6 months or premature death at high doses	Dog, Cat, Rabbit, Rat	In rats: 5, 15 mg/kg had no premature deaths. 25mg/kg had 1 rat of 5 die at 1 month, 4 lived 5-6months. 50mg/kg: 2 of 5 rats died 3 months. 3 lived 5-6 months
2013	Vesicular Nucleotide Transporter-Mediated ATP Release Regulates Insulin Secretion	Measure ATP release/ insulin secretion in pancreas cells treated with EB. EB blocks VNUT.	10nM-5uM dose	bathed over cells	30 min, then glucose treatment for 60 min w/EB in solution	MIN6 cells (clonal pancreatic B cell) islet isolate from c57male mice	EB blocks VNUT from releasing ATP, which in turn blocks insulin secretion.
2013	Inhibitors of ATP Release Inhibit Vesicular Nucleotide Transporter	Analyze different inhibitors of ATP release via VNUT	40nM	put in extracellular solution	2 minutes	E.coli expressing human VNUT	IC50:40nM for EB to inhibit VNUT release of ATP
2009	Differential modulation of AMPA receptor mediated currents by Evans Blue in postnatal rat hippocampal neurones	Investigate non-NMDA receptor mediated currents by EB treatment.	10uM EB	bathed over cultured cells	20s pre-equilibration	Hippocampal cultures from P0-P5 Wistar Rat	No effect <1uM, little effect at 2uM, IC50 at 10uM. EB modulates AMPA receptor gating in different neuron subsets. Could be useful in electrophysiology.
2004	Endocannabinoid-Independent Retrograde Signaling at Inhibitory	Investigate endocannabinoid	500nM-5uM	dissolved in pipet solution		Slices from p14-18	EB blocks retrograde signaling by blocking

	Synapses in Layer 2/3 of Neocortex: Involvement of Vesicular Glutamate Transporter	signaling. Use Evans Blue as a means to block VGLUT3				Sprague Dawley Rat	VGLUT3
1996	Evans blue antagonizes both alpha-amino-3-hydroxy-5-methyl-4-isoxazolepropionate and kainate receptors and modulates receptor desensitization	Investigate EB receptor interactions	1uM EB. Also did a dose response curve	Bathed over cells	unclear	Human Embryonic kidney 293 cells (HEK cells)	EB blocks glutamate evoked current from GluR6 (IC50:120nM) and GluR1(IC50:220nM).

References:

- 1 Cai, X. *et al.* Imaging the effect of the circadian light-dark cycle on the glymphatic system in awake rats. *Proc Natl Acad Sci U S A* **117**, 668-676, doi:10.1073/pnas.1914017117 (2020).
- 2 HUEPER, W. C. & ICHNIOWSKI, C. T. TOXICOPATHOLOGIC STUDIES ON THE DYE T-1824. *Archives of Surgery* **48**, 17-26, doi:10.1001/archsurg.1944.01230010020002 (2020).
- 3 AM, M. & C, F. Toxicity of Evans Blue Dye in the Monkey and Tracing of It in the Tooth Pulp. *Oral surgery, oral medicine, and oral pathology* **15**, doi:10.1016/0030-4220(62)90162-7 (1962).
- 4 Yin, Y. *et al.* in *Int J Mol Sci* Vol. 20 (2019).
- 5 Hablitz, L. M. *et al.* Increased glymphatic influx is correlated with high EEG delta power and low heart rate in mice under anesthesia. *Sci Adv* **5**, eaav5447, doi:10.1126/sciadv.aav5447 (2019).
- 6 Xie, L. *et al.* Sleep drives metabolite clearance from the adult brain. *Science* **342**, 373-377, doi:10.1126/science.1241224 (2013).
- 7 Iliff, J. J. *et al.* A paravascular pathway facilitates CSF flow through the brain parenchyma and the clearance of interstitial solutes, including amyloid beta. *Sci Transl Med* **4**, 147ra111, doi:10.1126/scitranslmed.3003748 (2012).

REVIEWERS' COMMENTS:

Reviewer #2 (Remarks to the Author):

The authors have supplied a significant amount of data and a thorough literature search to demonstrate that the acute administration of Evans blue does not cause any detectable toxicity. This are important data that will alleviate one of the major concerns of their new assay, that the dye might be causing inflammation that will disrupt the integrity of the BBB.

I remain unconvinced about the authors arguments regarding the rapid efflux dynamics to the periphery and the use of amyloid beta as a valid molecule for comparison. The authors cite Iliff et al, 2012 in the rebuttal but it states clearly in this manuscript that efflux directly to blood most likely plays a role compared to inert tracers:

"Soluble Ab1-40 was cleared from the brain interstitium more rapidly than a comparably sized dextran molecule, suggesting that interaction between specific BBB Ab efflux receptors with bulk flow-dependent clearance may occur."

As the authors cannot at this point rule out that unbound Evans blue (i.e. not bound to albumin as after intravenous administration) can (at least partially) directly clear through blood vessels to the periphery, I would highly recommend that they state this as a potential caveat of the assay.

We would like to thank the reviewer for their constructive feedback on our manuscript. For this revision, we have added the explicit caveat to our manuscript. Below, we have addressed the reviewer's concerns and questions to the best of our abilities. We look forward to your feedback and would appreciate consideration for publication in your journal.

Reviewer #2 (Remarks to the Author):

The authors have supplied a significant amount of data and a thorough literature search to demonstrate that the acute administration of Evans blue does not cause any detectable toxicity. This are important data that will alleviate one of the major concerns of their new assay, that the dye might be causing inflammation that will disrupt the integrity of the BBB.

I remain unconvinced about the authors arguments regarding the rapid efflux dynamics to the periphery and the use of amyloid beta as a valid molecule for comparison. The authors cite Iliff et al, 2012 in the rebuttal but it states clearly in this manuscript that efflux directly to blood most likely plays a role compared to inert tracers:

"Soluble Ab1–40 was cleared from the brain interstitium more rapidly than a comparably sized dextran molecule, suggesting that interaction between specific BBB Ab efflux receptors with bulk flow–dependent clearance may occur."

We still maintain that it is not surprising to measure a small, but significant peripheral signal from Evans blue at the 15 min mark (our first time point). We provide two lines of evidence:

- 1) We are in this study documenting that CSF tracer drainage to the lymph nodes occurs within 10 minutes after injection. Similar to the literature, our data shows that CSF can move quickly. Therefore, we are not surprised to measure a small amount of EB in blood at the 15 min mark.
- 2) Previous literature^{1,2} uses three separate radioactive tracers: A β , mannitol, and inulin, to measure clearance from the brain tissue. For both A β ^{1,2} and inulin², up to 25% is out of the brain as early as 15 minutes. In the case of mannitol, 25% is cleared as early as 1h.¹ The slower timescale may reflect differences in sensitivity of the assay, only sampled with 1h resolution¹. Therefore, we are not surprised to measure a small, but significant amount of EB cleared to the periphery in 15 min.

Based on these two lines of evidence, it is not surprising that we can measure EB, initially injected into the brain, in the periphery at the 15 min mark. Though we would like to, once again, emphasize that this is a small signal (less than 12% of total signal in every group measured), and do not yet know what this amount means in a more physiological setting. We hope this clarifies our argument.

As the authors cannot at this point rule out that unbound Evans blue (i.e. not bound to albumin as after intravenous administration) can (at least partially) directly clear through blood vessels to the periphery, I would highly recommend that they state this as a potential caveat of the assay.

As the reviewer has suggested, we have included this in the results section while introducing the technique:

"Glymphatic influx does not reflect function of the whole glymphatic system, therefore we measured clearance of Evans blue (EB), a small molecule (960 Da) that freely diffuses through the brain and binds tightly to albumin in the blood, from the brain to the femoral vein in anesthetized animals during the day (n = 9 mice) and night (n = 6 mice). We chose this method because although we cannot measure where in the vasculature EB enters and binds to albumin, any fluorescence detected in the femoral vein had to be cleared from the brain parenchyma. Thus, this approach enabled us to test the hypothesis that net clearance of EB from the brain is under diurnal control."[Page 6, Lines: 127-134]

References:

- 1 Iliff, J. J. et al. A paravascular pathway facilitates CSF flow through the brain parenchyma and the clearance of interstitial solutes, including amyloid beta. *Sci Transl Med* 4, 147ra111, doi:10.1126/scitranslmed.3003748 (2012).
- 2 Xie, L. et al. Sleep drives metabolite clearance from the adult brain. *Science* 342, 373-377, doi:10.1126/science.1241224 (2013).